# Ribosomal modifications are associated with mesenchymal fate selection in the neural crest lineage

Irina Poverennaya[1], Aliia Murtazina[2], Lei Li[3], Lorena Maili[4], Lukas Sourada[5,6], Luis Fernando Montano-Gutierrez[1], Rozalina Galimullina[1], Tobias Steinschaden[1], Marketa Kaiser[7], Tomas Zikmund[7], Adna Goralija[5,6], Teng Gao[8,24], Aurore Attina[9], Ornella Clara[10], Christoph Bartenhagen[11,12], Alek Erickson[2,13], Yaakov Gershtein[1], Shiyuan Chen[4], Kristyna Polaskova[5,14], Jaroslav Sterba[14], Bettina Semasch[15], Emma R. Anderson[15], Varsha Prakash[16], Theresa Vincent[16,17], Maria Arceo[18], Per Kogner[19], Susanne Schlisio[18], Peter V. Kharchenko[20], Alexandre David[9,21], Jozef Kaiser[7], Matthias Fischer[11,12], Jan Skoda[5,6], Paul A. Trainor[4,22], Andrei S. Chagin[3,23] ✉ & Igor Adameyko[1,2] ✉

Neural crest cells contribute to craniofacial formation by differentiating into skeletogenic mesenchyme and neuro-glial lineages. Using Smart-seq2 single-cell transcriptomics, we show that mesenchymal fate commitment correlates specifically with the expression of rRNA-modifying and ribosome assembly factors, rather than structural ribosomal proteins. Notably, EMG1 and NHP2 introduce key post-transcriptional modifications into 18S rRNA, including $m^1acp^3\psi$ at U1248, which requires TSR3 for final maturation. Disrupting NHP2 or TSR3 in vitro and in vivo perturbs cranial neural crest differentiation; post-migratory temporal knockout of *Polr1a* or *Polr1c* also causes craniofacial malformations. These findings align with cell type-specific $m^1acp^3\psi$ levels during neural crest differentiation. Given the neural crest contribution to neuroblastoma, we analyze patient data to find that elevated ribosomal control and rRNA-modifying proteins predict poorer outcomes. Complementary experiments in neuroblastoma cell lines reveal functional roles for TSR3 and WDR74 in mesenchymal-like tumor states. Together, our results link rRNA modifications and ribosome assembly to fate decisions, suggesting ribosomal heterogeneity shapes both normal development and tumor progression.

Neural crest cells are transient multipotent progenitors often considered to be the 4th germ layer[1], as they give rise to most of the craniofacial skeleton, sensory systems, autonomic nervous system, peripheral glial cells, melanocytes and cardiac cell populations[2,3]. These cells originate from the dorsolateral neural ectoderm during neurulation, and they delaminate via epithelial-to-mesenchymal transition (EMT) into the surrounding mesenchyme to migrate to numerous final destinations[4–8]. During their migration, neural crest cells experience diverse environmental signals and commit to a variety of fates depending on their position[2]. However, the mechanisms of their fate selection and multipotency control are not entirely understood, especially when it comes to the separation of mesenchymal

skeletogenic and neuro-glial progenies. Recently, by taking advantage of single-cell data, we revealed that competing gene expression programs bias neural crest cells prior to fate selection and drive them into different fates[9,10]. The analysis of the composition of such programs shed light on the molecular mechanisms regulating their downstream fates[11,12]. This issue is specifically important for understanding the differentiation bias of neural crest cells towards skeletogenic fates, which remains poorly understood, and is key to comprehending craniofacial pathology.

Given the degree of neural crest multipotency and its major role in vertebrate embryonic development[13], it is of no surprise that diverse pathologies stem from defects in neural crest cell development, ranging from congenital craniofacial and cardiac anomalies[14] to tumors such as neuroblastoma, pheochromocytoma, melanoma and Schwannoma[15]. Anomalies in neural crest cell fate selection mechanisms might underpin the heterogeneity and malignancy of neural crest-derived tumors, which often exhibit signs of reiterative use of developmental programs and mesenchymal malignant subpopulations[9,16,17].

The generation of different proportions of distinct cell types as a result of fate selection implies that prior to fate selection, progenitor cells represent a micro-heterogeneous population, with individual cells biased towards different specific future fates or states with varying strength[10,18,19]. With the aid of robust single-cell data, it is possible to examine this micro-heterogeneity and determine the specific, coordinated transcripts responsible for regulating the proportions of resulting cell types prior to fate bifurcation[9,10,20].

Importantly, the nature of fate-biasing mechanisms and their competition is not well understood, as they likely consist of complex combinations of extrinsic signals and intrinsic states or processes[13]. Examples of biasing factors range from signals from other cells[21] to autonomous epigenetic priming[22,23]. In theory, any significant intracellular process can contribute to fate biases[19]. For instance, different fate choices might require activation of divergent metabolic states or sets of efficiently translated proteins[24]. From this point of view, the presence of specialized ribosomes preferentially translating specific mRNAs[25] might introduce a bias into fate-related or other decision-making processes.

Ribosome biogenesis (assembly), ranging from the transcription of ribosomal DNA (rDNA) to fully assembled mature and translation-competent ribosomes in the cytoplasm, is the most energy-consuming process in the cell. It was recently reported that the drastic changes in cell phenotype, in particular during neural crest delamination and tumor progression and metastasis, are driven by de novo rRNA biogenesis to accommodate renewal and potentially modification of the cellular ribosomal pool[26,27]. Such changes in the ribosome pool may enable ribosome specialization[28,29]. The existence of specialized ribosomes, albeit still debated, postulates that differences in ribosome composition either at the rRNA[30–32] or at the ribosomal protein level[33] will result in preferential translation of specific subgroups of mRNAs[34–37] underpinning cell plasticity in normal physiology and in disease. Specialization and functional diversity of ribosomes can be achieved by numerous means. At the rRNA level, it can be caused by rRNA variant allele expression[38,39] or by chemical modifications at the level of individual nucleotides[40,41]. At the ribosomal protein level, heterogeneity can arise from the expression of specific ribosomal protein paralogues[33,37,42], post-translational modification of ribosomal proteins[43,44], by differences in ribosomal protein composition as some ribosomal proteins are found to be substoichiometric[36,45] or by distinct proteins associating with the mature ribosomes[46,47].

In this study, we investigated the system of fate choices in cranial and trunk neural crest cells (NCCs) to understand the factors associated with their transition into skeletogenic ectomesenchyme of the future face. As a result, we discovered that post-transcriptional nucleotide modifications at position 1248 of 18S rRNA, corresponding to the ribosomal P-site (which hosts the peptidyl-tRNAs during translation elongation as well as ribosomal assembly processes), correlate not only with specific fate selection in neural crest, but also show a connection with nervous system-derived tumors. Overall, we extended our knowledge of the potential mechanisms regulating fate selection and cell states towards post-transcriptional control involving sequence modification of rRNA transcripts and assembly of new, potentially specialized, ribosomes.

## Results

### Ectomesenchymal and neuro-glial fate split in CNCC development

A prior analysis of fate selection in neural crest cells (NCC) revealed the opposing programs that operate right before fate decisions are made in the trunk and cervical (vagal and cardiac) NCCs[10]. However, the fate selection of ectomesenchymal vs neurogenic fates has never been addressed for HOX-negative cranial neural crest populations which contribute to the formation of the mammalian face.

Therefore, we set out to explore how fate outcomes such as skeletogenic ectomesenchyme and neuro-glial cell types of the head arise due to decision-making processes from migratory cranial NCCs (CNCCs). To understand the fate biasing of ectomesenchymal and neuro-glial programs in mouse CNCCs at E8.5, E9.5 and E10.5, we sequenced FACS-isolated individual *Wnt1*-traced cells from the head of *Wnt1-Cre/R26Tomato* mouse embryos using the Smart-seq2 protocol[48], which allows for recovery of the expression of 7000-9000 genes per individual cell on average (Fig. 1a). As expected, the single cell analysis revealed two super-clusters representing the neural tube (*Sox2*+*Sox3*+) and the neural crest (*Sox10*+) cell populations connected via the bridge of delaminating CNCCs (Fig. 1b, c). The data revealed that migratory *Sox10*+*Twist1*+ CNCCs underwent the fate selection, giving rise to two distinct branches of *Twist1*+*Prrx1*+ mesenchymal cells (future skeletogenic ectomesenchyme of the face), and *Sox10*+*Plp1*+*Neurog1*+ neuroglial progenitors, which give rise to Schwann cell precursors (SCPs), as well as cranial sensory and autonomic neurons.

To validate these transitions within the CNCC lineage, we took advantage of RNA velocity (Fig. 1d), which predicts the "future" state of a cell based on its current mRNA splicing dynamics. Results of RNA velocity agreed with our computed and biological knowledge of the CNCC developmental trajectory (Fig. 1e), which was built using the SimplePPT approach for tree construction, implemented by scFates[20].

Already at the beginning of their trajectory, in the EMT phase, the CNCCs exhibit increased levels of the mesenchyme gene expression program in comparison to the trunk NCCs (Supplementary Fig. 1), i.e., CNCCs express *Twist1* - a key driver of mesenchymal fate[10,49], whereas trunk NCCs proceed through EMT only with *Snai* genes and without *Twist1* expression[10]. Even though *Twist1* is expressed early in migratory CNCC, its transactivation activity, predicted by coordinated up-regulation of possible targets, turns on closer to the first bifurcation point leading to stable mesenchymal fate acquisition, which is accompanied by the transcriptional activation by another key mesenchymal gene – *Prrx1* (Supplementary Fig. 2a). Contrary to this pattern, the transactivation activity of another EMT driver, *Snai1*, fades away when the bifurcation point towards ectomesenchymal fate is reached (Supplementary Fig. 2b).

Overall, the mesenchymal bias of delaminating and migrating CNCCs precedes the first fate selection point leading to facial ectomesenchyme, neurons and glia. This observation suggests that the CNCC-derived neurons emerge from the progenitor cells with an initially strong mesenchymal bias, which then becomes down-regulated in some cells around the fate selection/bifurcation point. This raises questions about the molecular mechanisms underpinning fate selection in CNCCs, which we attempted to address below.

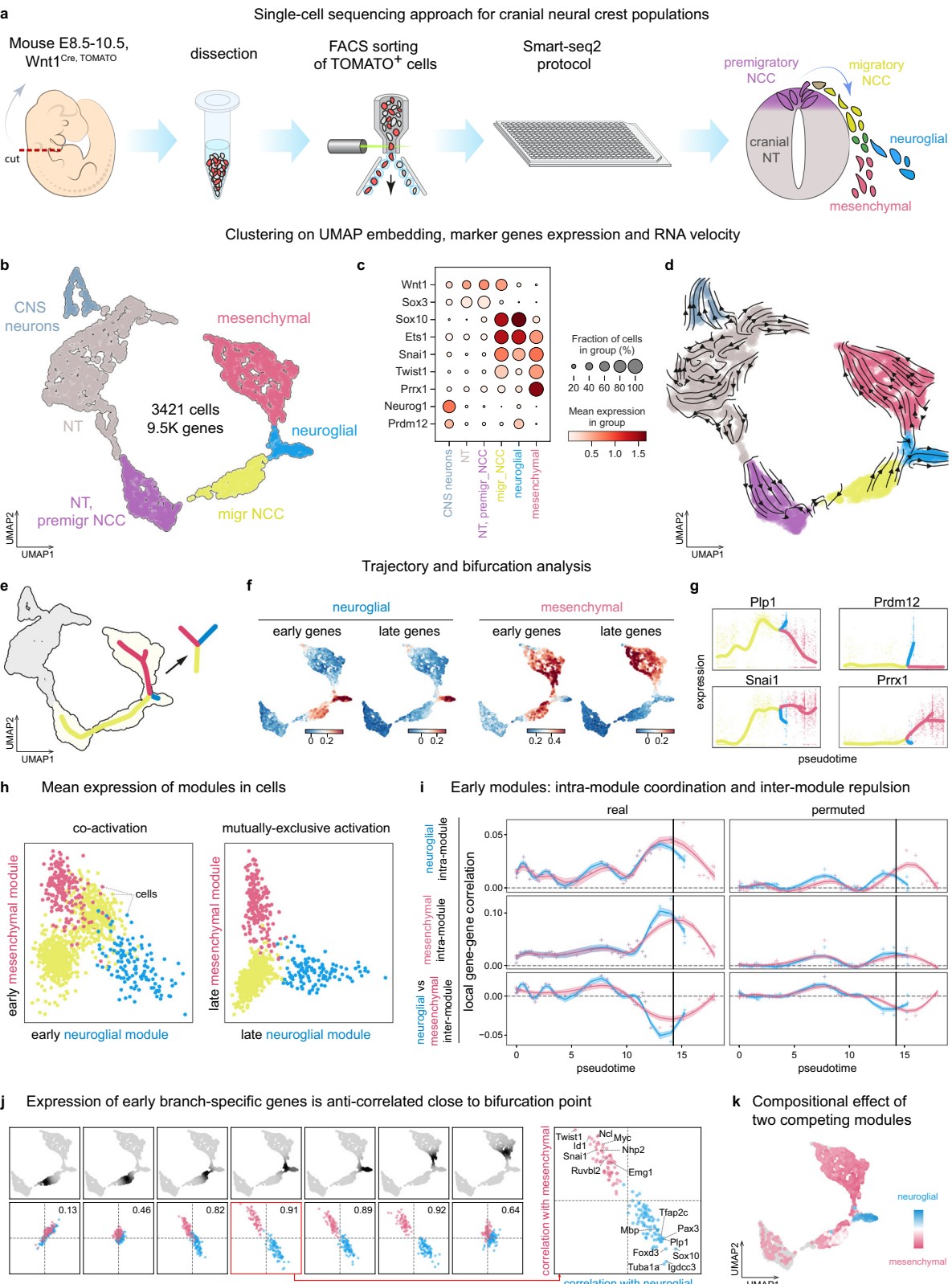

**a** Single-cell sequencing approach for cranial neural crest populations

Clustering on UMAP embedding, marker genes expression and RNA velocity

Trajectory and bifurcation analysis

**h** Mean expression of modules in cells

**i** Early modules: intra-module coordination and inter-module repulsion

**j** Expression of early branch-specific genes is anti-correlated close to bifurcation point

**k** Compositional effect of two competing modules

## Ribosomal control and rRNA modifications program in CNCC fates

We therefore analyzed the presence of pro-ectomesenchymal and pro-neuroglial fate-biasing programs prior to the first bifurcation point using scFates[20]. This approach benefited from Smart-seq2 deep sequencing (8 K genes per cell on average), enabling the identification

of fate-associated early pre-bifurcation genes expressed in migratory CNCCs before their commitment to terminal fates (Fig. 1e, f).

This analysis revealed that individual pre-bifurcation CNCCs shared the gradually emerging expression of ectomesenchymal- and neuro-glial- biasing programs with increasing negative cross-correlation in the same cells (Fig. 1g–i). This negative correlation

**Fig. 1 | Competing gene programs in early cranial neural crest development revealed by single-cell analysis. a** Overview of cranial neural crest single-cell sampling and sequencing. **b** UMAP embedding of mouse cranial dataset (NC and NT lineages), color-coded by cell type (3421 cells). **c**, Expression of key marker genes. **d** RNA velocity vectors mapped on UMAP embedding to show the differentiation directionality. **e** Overview of the bifurcation between ectomesenchymal and neuroglial fates on UMAP embedding. Colors encode cell type (as in (**a**)). **f** Average expression pattern of early and late branch-specific modules: neuroglial in blue and mesenchymal in pink. Early and late modules are sets of branch-specific genes that are activated before or after the bifurcation, respectively. **g** Example of an early and a late gene of neuroglial module (*Plp1* and *Prdm12*) and mesenchymal one (*Snai1* and *Prrx1*) and their expression trends on pseudotime. **h** Scatter plot of average expression of early (left) and late genes (right) from mesenchymal and neuroglial modules in each cell. **i** Mean local gene-gene correlations of all possible gene pairs inside one module, or between the two modules along the trajectory. Early modules exhibit increasing intra-module coordination and inter-module repulsion. Average local correlations within neuroglial and mesenchymal modules progressively increase before the bifurcation point, indicating enhanced intra-module coordination. In parallel, inter-module correlations decline, reflecting growing repulsion between modules. Permuted controls show much weaker trends thus supporting the biological specificity of the observed patterns. Shaded bands represent the magnitude of correlation expected by chance (permutation-based null model). Plus signs (+) indicate the onset of significant gene synchronization. **j** Correlation plot of early branch-specific genes on a non-intersecting window of cells (highlighted in black on UMAP). Close to the bifurcation point the repulsion of the early genes from different modules could be observed. **k** Compositional effect of neuroglial and mesenchymal modules that are co-activated and competing before the bifurcation point.

suggested direct or indirect competition of opposing biasing programs prior to commitment to a fate (Fig. 1j, k). After the bifurcation, this early biasing mesenchymal program turned into a "late mesenchymal signature" (Supplementary Data 1) of the differentiating facial ectomesenchyme (Fig. 1f).

Next, we selected genes representing the early pre-bifurcation biasing program associated with the ectomesenchymal fate outcome ("early mesenchymal signature" in Supplementary Data 1) and probed their functional relationships and potential interactions using the bioinformatics tool STRING[50]. We observed two large, interconnected clusters of functionally interacting proteins within the network graph generated by STRING (Fig. 2a). One of the interaction-enriched protein sub-networks corresponded to EMT and mesenchymal cell state based on the expression of *Snai1* and *Twist1*, as well as other genes involved in extracellular matrix remodeling and cell adhesion (Fig. 2a). However, the largest functional sub-network of interacting partners included genes associated with different ribosomal processes such as *Ncl*, *Nhp2*, *Wdr75*, *Ddx31*, *Ruvbl1* and others (Supplementary Data 1; Fig. 2a and Supplementary Fig. 3). Among those, *Ncl*, *Nhp2*, *Emg1*, *Wdr74/75* and *Rrp8* are related to rRNA processing. In particular, nucleolin (NCL), in addition to its role in rRNA transcription and ribosomal assembly, appears to play a crucial role in the early cleavage of pre-rRNA[51]. Furthermore, the loss of NCL function has been linked to craniofacial anomalies during embryonic development[52]. Similarly, the product of the *Wdr74* gene is implicated in early pre-rRNA cleavage[53], whereas WDR75 is required for optimal pre-rRNA transcription[54]. *Rrp8* encodes the methyltransferase that participates in the methylation of rRNA and silencing of rDNA loci[55,56]. Both *Nhp2* and *Emg1* gene products are involved in the posttranscriptional hypermodification of uridine in 18S rRNA[32]. Specifically, NHP2 is an essential component of the SNORA13 complex (snR35 in yeast), which catalyzes the transformation of uridine into pseudouridine (the first step)[57,58]. Pseudouridine is subsequently methylated by the EMG1 methyltransferase (the second step), which is mutated in association with Bowen Conradi syndrome, a pleiotropic condition characterized by craniofacial anomalies[59,60]. The assembly of such C/D box and H/ACA box snoRNPs as mentioned for the SNORA13 complex is facilitated by a chaperone complex that includes other proteins in our network encoded by *Ruvbl1* and *Ruvbl2* genes[61]. Therefore, we named this coordinated sub-network "ribosomal control and rRNA modifications signature" (Supplementary Data 1) implying that the corresponding proteins are involved in the control of rRNA and ribosomal assembly and modification.

To experimentally test if the expression of *Nhp2* indeed correlates with transition into facial ectomesenchyme, we performed whole-mount in situ hybridization for *Sox10* and *Nhp2* at E9.5 (Fig. 2b, c). *Nhp2* appeared to be more strongly expressed in the branchial arches and anterior face as compared to the coalescing trigeminal ganglion (*Sox10*⁺ cells). A closer examination of the branchial arch region revealed that *Nhp2* is more prominently expressed in NC-derived cells (outer layer in Fig. 2c), as compared to centrally-positioned mesoderm-derived skeletal muscle progenitors[62] (highlighted inner region in Fig. 2c).

To further explore the correlation of the mesenchymal early signature with transcriptional cell states in the entire neural crest lineage, we analyzed a previously published single-cell RNA-seq dataset from Kastriti *et al.*[9]. We found that the expression of individual genes from the mesenchymal early signature as expected appears in cranial ectomesenchyme and correlates with the overall plotted consensus of pro-mesenchymal signature, albeit with very few exceptions. For example, *Myc* (here and below, *c-Myc*) was ubiquitously distributed throughout multiple neural crest derivatives, except in sensory neurons (Fig. 2f).

## Ribosomal biogenesis is critical for craniofacial development

EMT requires significant cytoskeletal modifications and global changes of protein pools[63,64]. Thus, it might either demand elevated levels of new protein synthesis in general, or require an increase in the ratio of specialized ribosomes driving EMT-related translation. Both cases require new rRNA biosynthesis executed by polymerase 1 (POLR1).

Recently, two studies revealed that the time-controlled knockout of polymerase 1 subunit alpha (*Polr1a*) at the time of CNCC delamination (E8.5) causes craniofacial, neural, and cardiac anomalies[26,65]. To explore the importance of rRNA synthesis in fate choice towards ectomesenchyme of the face, we performed a similar experiment at a later migratory and post-migratory stages, E9.0-E9.5, which corresponds to the first bifurcation timepoint in the single cell CNCC dataset.

The compositional analysis of the single cell transcriptomics data from E12.5 control and temporal *Polr1a* knockout mutant mice (based on R26-CreERT2), injected with tamoxifen at E9.0-E9.5, revealed a significant reduction of several ectomesenchymal clusters, without neuro-glial progeny being affected (Fig. 3a, b and Supplementary Fig. 4a, b). To validate this observation, we performed HCR in situ hybridization for *Sox9*, a master regulator of chondrogenesis; *Runx2*, a master regulator of osteoblast differentiation; and *Sox10*, a marker of the developing peripheral nervous system (Fig. 3c, d, left panel). We also carried out immunostaining for SOX9 and TUJ1 (TUBB3A; neuron-specific class III β-tubulin) to corroborate the changes at a protein level (Fig. 3d, right panel). As expected, mutant embryos exhibited reduced SOX9 signal in the mandible, consistent with diminished cartilage development. This was evident most notably for Meckel's cartilage, which manifested as two discrete non-contiguous proximal and distal elements (Fig. 3c, d). In contrast, the branches of the neural crest-derived trigeminal ganglion were comparatively less affected, and the mandibular nerve now traversed the gap between the non-contiguous Meckel's cartilage elements (Fig. 3c, d). Although craniofacial malformations are evident in *Polr1a* mutants in the bright field images (Supplementary Fig. 4c, d); we opted for higher resolution microCT

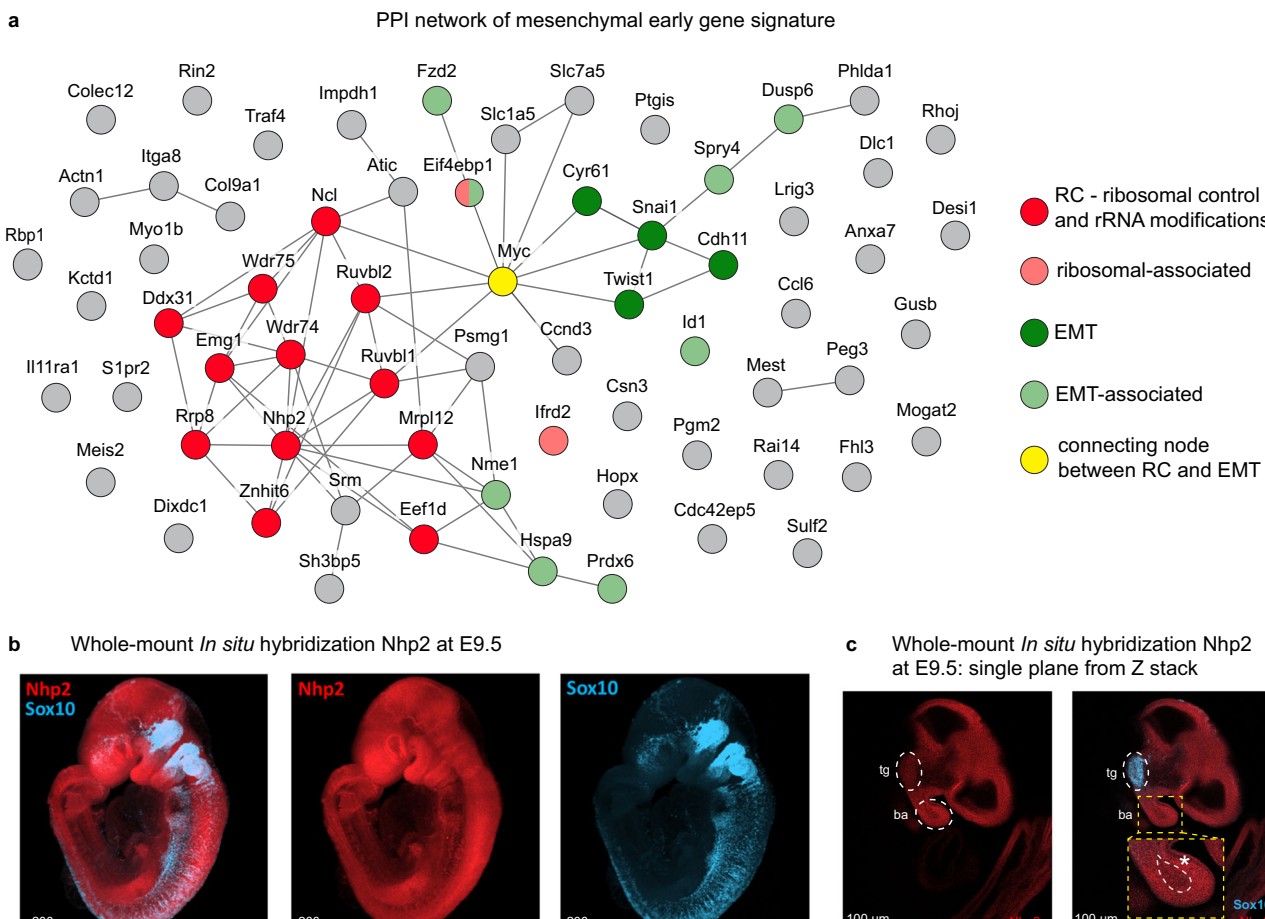

**b** Whole-mount *In situ* hybridization Nhp2 at E9.5

**c** Whole-mount *In situ* hybridization Nhp2 at E9.5: single plane from Z stack

**d** Expression of genes from mesenchymal early signature in neural crest development dataset from Kastriti, Faure and von Ahsen et al.

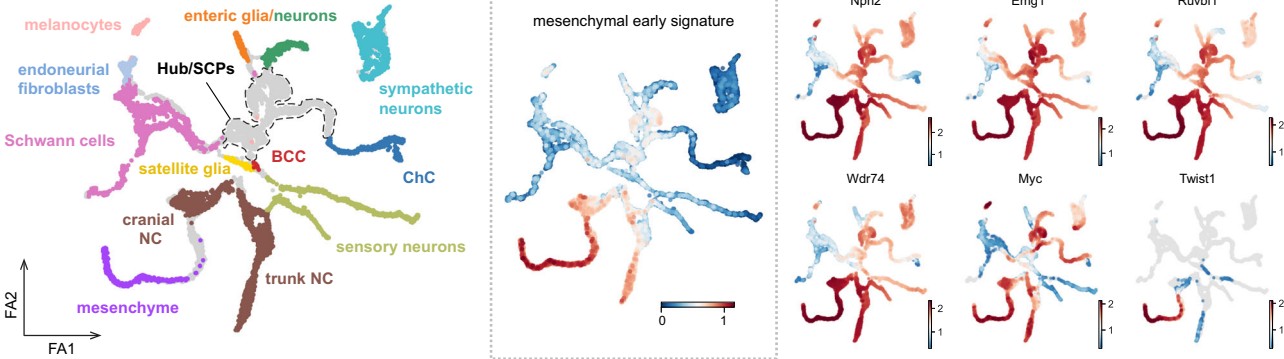

**Fig. 2 | The core of the mesenchymal early signature consists of genes associated with EMT and genes involved in ribosomal control and rRNA modifications. a** Protein-protein interaction network achieved by STRING for a mesenchymal early signature that was predicted by scFates as a set of genes associated with mesenchymal bias in CNCCs. The color code of the nodes reflects their association with the process of the epithelial-mesenchymal transition (green for direct participants and light green for associated ones) or with ribosome-related processes (red for the ribosomal control and rRNA modifications signature, light red for any other ribosomal-associated genes). **b** Whole-mount in situ hybridization for Nhp2 and Sox10 at E9.5 mouse embryo (n = 2). **c** Single plane from Z stack for whole-mount in situ hybridization of Nhp2 at E9.5 shows that Nhp2 is more expressed at the neural crest-derived branchial arch cells (ba) than in mesodermal-derived branchial arch cells and trigeminal ganglion (tg) cells (n = 2). **d** Expression of mesenchymal early signature and its individual genes in Smart-seq2 neural crest lineage dataset from (Kastriti et al., 2022). Expression of genes was denoised with MAGIC package[157].

scanning of PTA-contrasted embryos[66] and rendered the 3D volumes of trigeminal ganglion and ectomesenchymal chondrogenic condensations (Fig. 3e and Supplementary Fig. 4e, Supplementary Data 2). This approach revealed that whereas *Polr1a^{flx/flx};R26-Cre-ERT2* embryos exhibited reduced facial mesenchymal condensations, the trigeminal ganglia were not significantly affected, indicating that facial skeletogenesis was specifically disrupted as a consequence of diminished rRNA transcription and ribosome biogenesis (Fig. 3e and Supplementary Fig. 4e).

Similarly, temporal knockout of another RNA Polymerase I subunit, *Polr1c*, at E9.0-E9.5 also resulted in critical craniofacial abnormalities at E12.5, with reduced neural crest derivatives and

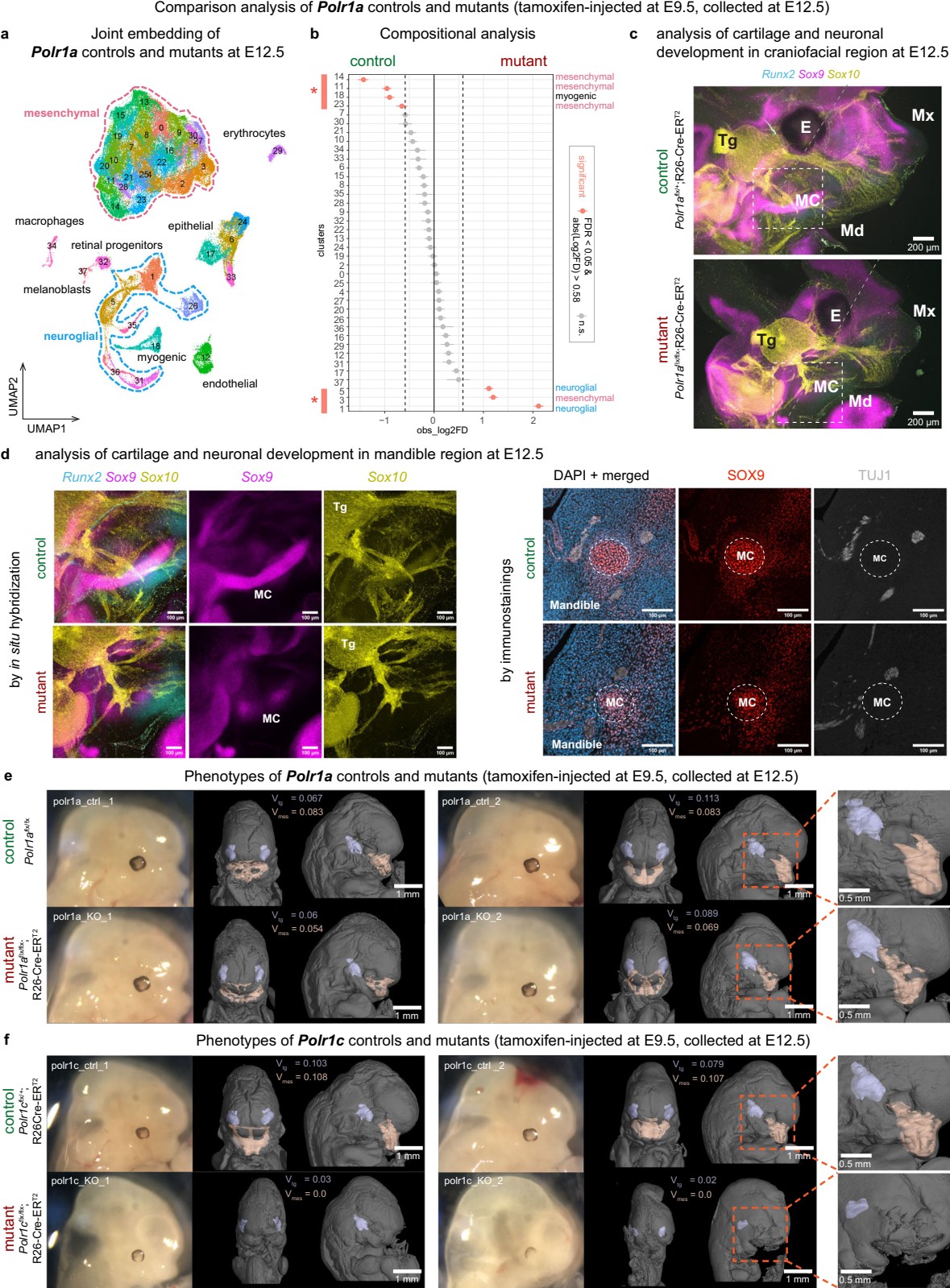

Comparison analysis of *Polr1a* controls and mutants (tamoxifen-injected at E9.5, collected at E12.5)

complete absence of skeletal condensations in the face (Fig. 3f, Supplementary Fig. 4e). In contrast, the trigeminal ganglion was persisting in expected position. Together, these observations support a mechanism in which the dynamics of rRNA synthesis at the time of fate bifurcation influences later ectomesenchymal development in CNCCs.

## m¹acp³ψ modification of 18S rRNA correlates with CNCC fates

As we did not find any correlation between CNCC lineage fates and structural ribosomal proteins known to mark specialized ribosomes[36,37,45], we hypothesized that rRNA is post-transcriptionally modified, which in turn impacts ribosome function and possibly their re-assembly.

**Fig. 3 | Disruption of rRNA transcription via knockout of Polymerase 1 results in craniofacial abnormalities. a** Sequencing results from eight control and *Polr1a* mutant craniofacial tissues (4 biological replicates per condition) were analyzed together, and 38 clusters, representing 9 major cell types, were identified. **b** Compositional analysis results identified altered cell proportions in Polr1a mutants (sample size and number of replicates are described in **a**). Mesenchymal and myoblast cells were reduced, while 2 neuroglial cell clusters and one mesenchymal cluster were proportionally increased in mutants compared to controls (scProportion test, marked with *: FDR < 0.05, and abs(Log2FD)> 0.58). Dots represent the observed log2 fold difference, and error bars represent the 95% confidence intervals (CI). **c** Comparison of craniofacial development in *Polr1a* control and mutant mouse embryos at E12.5 by in situ hybridization for Sox9 (a master regulator of chondrogenesis) and Sox10 (here a marker of developing peripheral nervous system). Abbreviations: E - eye, MC - Meckel's cartilage, Md - mandibular prominence, Mx - maxillary prominence, Tg - trigeminal ganglion. Scale bar: 200 μm. **d** In situ hybridization for *Sox9* and *Sox10* (left) and immunostainings for Sox9 and Tuj1 (alias name is Tubb3a; a neuron-specific class III β-tubulin) (right) in the mandibular region of *Polr1a* control and mutant mouse embryos at E12.5 (*n* = 2). Scale bar: 100 μm. **e** Brightfield image and corresponding 3D visualizations based on microCT data of two *Polr1a* control and two *Polr1a* mutant mouse embryos at E12.5. Segmentation of mesenchymal and trigeminal cell condensations (with associated volumes in mm$^3$) are highlighted in light orange and violet, respectively. **f** Brightfield image and corresponding 3D visualizations based on microCT data of two *Polr1c* control and two *Polr1c* mutant mouse embryos at E12.5. Segmentations of mesenchymal and trigeminal cell condensations (with associated volumes in mm$^3$) are highlighted in light orange and violet, respectively.

Indeed, our analysis of the Smart-seq2 data revealed the systematic appearance of an "incorrect" nucleotide at position U1248 in 18S rRNA (Fig. 4a). This position in the structure of 18S rRNA does not reside in variable regions of pre-rRNA genes that are known for accumulating rRNA genomic variants[39,67]. Instead, the U1248 position maps to one of the most highly conserved regions of 18S rRNA - the P-site of the ribosome, where peptide-bond formation is catalyzed (Fig. 4b)[68]. Interestingly, the sequencing error in position U1248 appeared over-represented in cells of premigratory and migratory CNCCs (Fig. 4c–f).

The canonical reference nucleotide at position 1248 of 18S rRNA is uridine (U), which is known to be hypermodified via a three-step reaction catalyzed by the SNORA13 complex (including NHP2 protein), EMG1 and TSR3 proteins (Fig. 4g)[57,69]. Importantly, the expression of the corresponding genes *Nhp2* and *Emg1* have been associated with the transition of migratory neural crest to ectomesenchymal cell types −as a part of the ribosomal control and rRNA modifications signature (see Fig. 2a, red color-code; Supplementary Data 1). Though the non-canonical nucleotides at position 1248 were enriched in the reads from premigratory and migratory CNCCs (Fig. 4c, e), their distribution did not correlate with expression of ribosomal structural protein genes (Fig. 4d). This indicates that CNCC fate selection may be influenced by specialized ribosomes harboring specifically modified 18S rRNA, rather than by ribosome abundance per cell.

Our results correspond with the previously shown increase of rRNA synthesis in delaminating NCCs[26,27], and also to the ubiquitously present expression of transcripts of ribosomal structural proteins in all neural crest-derived populations in our Smart-seq2 data (see Fig. 4d). Altogether, these data suggest that fate selection might depend on the ratio of specifically modified rRNA structures and relevant specialized ribosomal pools.

To confirm the reason behind the sequencing errors in the evolutionary conserved U1248 position of 18S rRNA, and to eliminate genomic variants of 18S rRNA, we examined human genomic and transcriptomic data obtained from the GTEx project[70] (Fig. 5a). Our analysis of genomic data from 10 different individuals showed no nucleotide substitutions in 18S rDNA (Fig. 5b). However, at the transcriptomic level, the distribution of nucleotides at position 1248 of 18S rRNA varies from individual to individual (see Fig. 5c) and even within different tissues of the same individual (Fig. 5d). These data indicate that variations at position 1248 in 18S rRNA are introduced at the mRNA level.

Indeed, it has already been shown that heavy nucleotide modifications introduce sequencing errors[71–73], and we reasoned that the observed variability at position U1248 arises during the reverse transcription step of sequencing, due to heavy modifications of this specific uridine.

### Proportions of rRNA modifications are tissue-dependent

Uridines may harbor various modifications, including pseudouridine (ψ), N1-methyl pseudouridine (m$^1$ψ), or N1-methyl-N3-(3-amino-3-carboxypropyl) pseudouridine (m$^1$acp$^3$ψ) (Fig. 4g). Pseudouridine can be excluded as a cause of sequencing error as it is known to be widely present in 18S rRNA[74,75], and yet no other repetitive pseudouridine-associated sequencing error was detected in our single-cell data. At the same time, an extensive m$^1$acp$^3$ψ modification was reported for U1248 in yeast and mammalian cells[74,76,77].

To confirm the presence of the m$^1$acp$^3$ψ modification in murine embryonic 18S rRNA, we performed mass spectrometry (Fig. 5e). Direct analysis at earlier stages, such as cranial neural crest EMT (E8.5) and subsequent migration and fate selection (E9-9.5), was not feasible due to the limited number of available cells and the sensitivity constraints of mass spectrometry. We therefore chose a slightly later developmental window (E9.5-E10), when cell numbers are substantially higher and allow for reliable detection. To contrast the predominant tissue types at this stage, we dissected the heads of twenty E9.5-E10 mouse embryos from multiple independent litters and surgically separated brain regions from facial structures, collecting them into two distinct sample sets for quantitative analysis. After isolating 18S rRNA from pooled developing brain areas and advanced facial structures, we subjected it to mass spectrometry to investigate the relative abundance of 30 different nucleotide modifications, including pseudouridines and m$^1$acp$^3$ψ. We quantified the relative abundance of each modification by a ratio of modified/unmodified nucleoside peak area. Mass spectrometry results confirmed the differential presence of m$^1$acp$^3$ψ in 18S RNA from the brain area and facial samples (tissues from multiple embryos were pooled together for this analysis) (Fig. 5f) at developmental stages after the neuro-mesenchymal bifurcation in CNCCs. Also, as expected, the levels of m$^1$acp$^3$ψ were much lower compared to other modifications such as pseudouridines (ψ), 2′-O-methyladenosine (Am), 3-methyluridine (m$^3$U) or even 3-(3-amino-3-carboxypropyl)uridine (acp$^3$U) (Supplementary Fig. 5, Supplementary Data 3).

Though the mechanism and functional relevance of the hyper-modification at position U1248 should be established in different contexts, previous experiments in yeast showed that loss of the SNORA13 (snR35) complex (containing NHP2), which initiates the hypermodification process, reduced the speed of amino acid incorporation and cell growth[68]. Considering the low proportions of observed U1248 modifications in trunk neural crest cells (Supplementary Fig. 6), we reasoned that pro-ectomesenchymal CNCCs may require a specialized pool of ribosomes with the modified U1248 site in 18S rRNA.

### m$^1$acp$^3$ψ rRNA-modifying enzymes support CNCC development

The key steps of enzymatic post-transcriptional modification of U1248 are catalyzed by NHP2 (as a part of SNORA13 complex) and EMG1, which are a part of the signature correlating with mesenchymal vs neuro-glial fate choice in NCCs (Fig. 2a). The TSR3 accomplishes the final step of the m$^1$acp$^3$ψ modification.

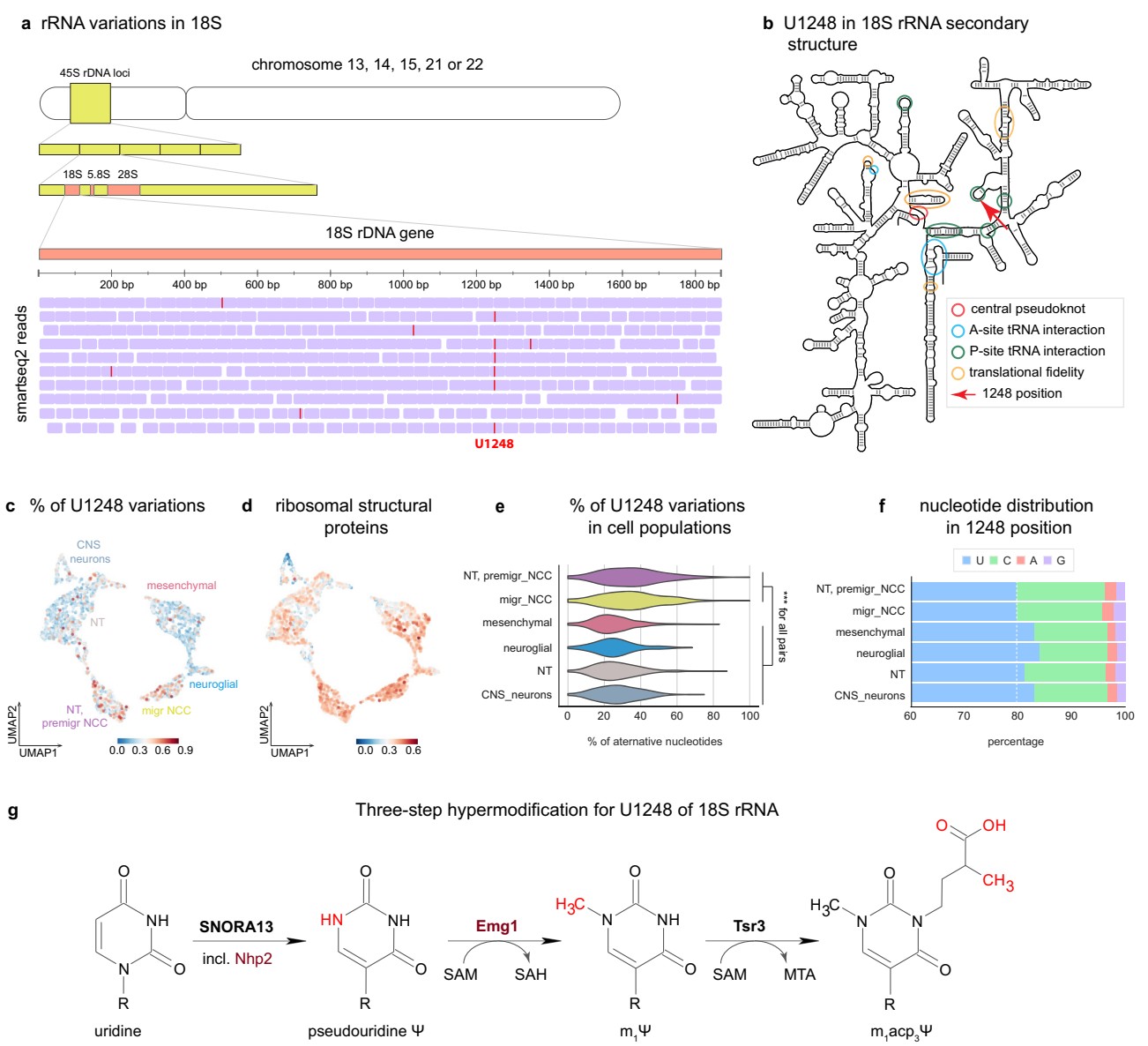

**Fig. 4 | The search for significant 18S rRNA variations in the mouse cranial NC lineage. a** Schematic representation of rDNA organization on mouse chromosomes with highlighted 45S rDNA loci. The 45S precursor transcript encodes the 18S, 5.8S, and 28S rRNAs. A zoomed-in view of the 18S rDNA gene is shown, with aligned Smart-seq2 reads visualized in a genome browser-style layout (similar to IGV). Each violet rectangle represents an individual read, with mismatches - corresponding to putative sequence variations or post-transcriptional modifications - highlighted in red. **b** A secondary structure of human 18S rRNA with highlighted functional regions of rRNA. **c** UMAP embedding of mouse cranial neural crest dataset colored by the percentage of U1248 modification per cell. **d** Average expression pattern of ribosomal structural protein genes. **e** Violin plots showing the percentage distribution of U1248 modification in different cell populations. Asterisks (***) indicate statistical significance (FDR < 0.001) for all pairwise comparisons determined by two-sided Mann-Whitney U tests with Benjamini-Hochberg FDR correction for multiple comparisons. **f** Nucleotide composition in 1248 position of 18S rRNA in different cell populations. **g** Three-step hypermodification pathway for uridine in 1248 position of 18S rRNA. ψ − pseudouridine; m$^1$ψ - N1-methyl pseudouridine; m$^1$acp$^3$ψ - N1-methyl-N3-(3-amino-3-carboxypropyl) pseudouridine; SAM - S-adenosyl-L-methionine; SAH - S-adenosyl-L-homocysteine; MTA - 5′-methylthioadenosine. Source data are provided as a Source Data file.

To explore the functional importance of the m$^1$acp$^3$ψ modification in embryonic neural crest cells, we utilized lentiviral vectors to overexpress *Nhp2*, *Emg1*, or *Tsr3* in iPSC-derived neural crest (NC) cells in vitro according to a published protocol (see Methods). We chose an overexpression strategy because previous studies reported early embryonic lethality for *Nhp2* and *Emg1* gene knockout mutants[78,79]. As NCC emerged in vitro, we collected the cultured cells and analyzed them with the help of 10X Chromium single cell transcriptomics. Different ratios of neural tube−like (premigratory progenitors) and NCC-like cells (progenies) were observed under NHP2 and TSR3 overexpression conditions, supporting the conclusion that

these proteins influence the formation of NCCs (Supplementary Fig. 7).

To assess the role of these rRNA-modifying enzymes in vivo, we overexpressed *Nhp2* and *Tsr3* in the neural ectoderm of mouse embryos immediately before the NCC formation. We employed the NEPTUNE system of high-frequency ultrasound-guided nano-injections of lentiviral particles expressing the proteins of interest, together with a fluorescent protein marker[80]. The amniotic cavity of E7.5-E8.0 embryos was injected with *Nhp2*- and *Tsr3*-carrying lentiviruses, and the embryos were harvested at E9.5 after the CNCC cells emigrated to begin to build the cranioskeletal and neuro-glial

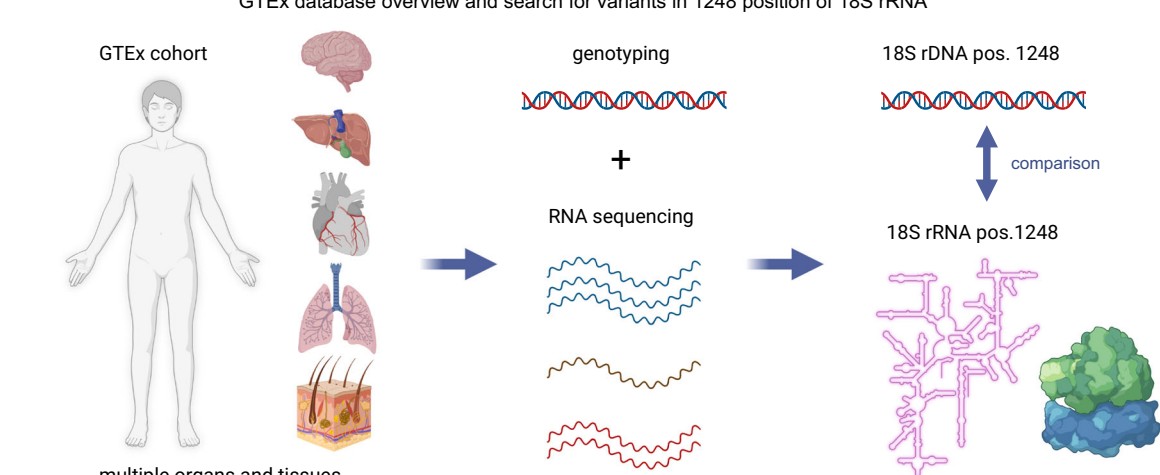

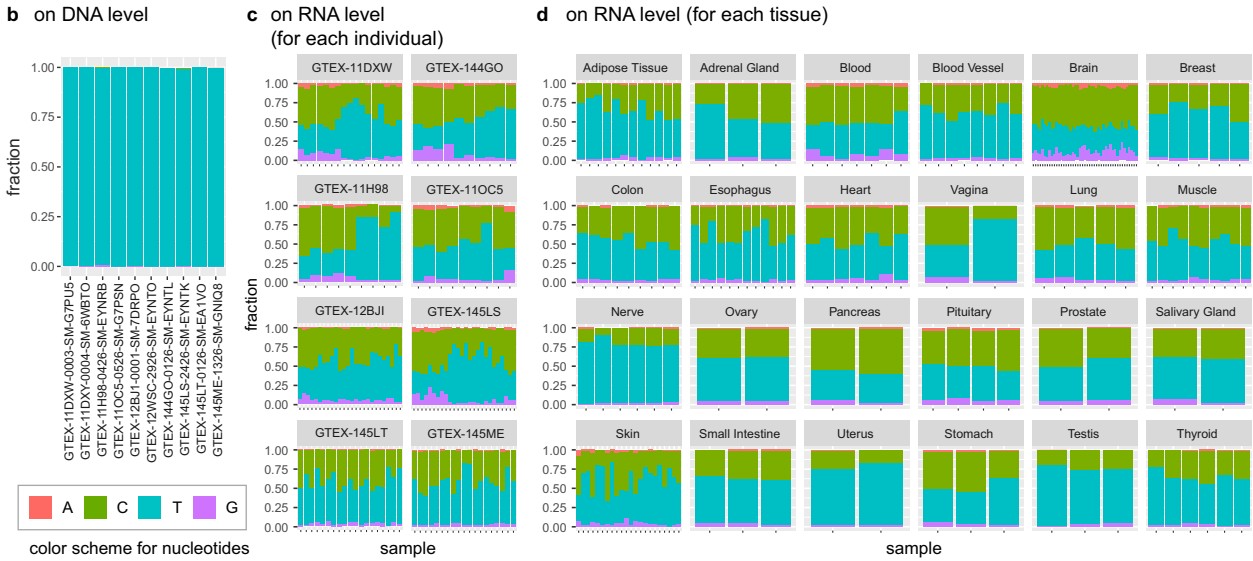

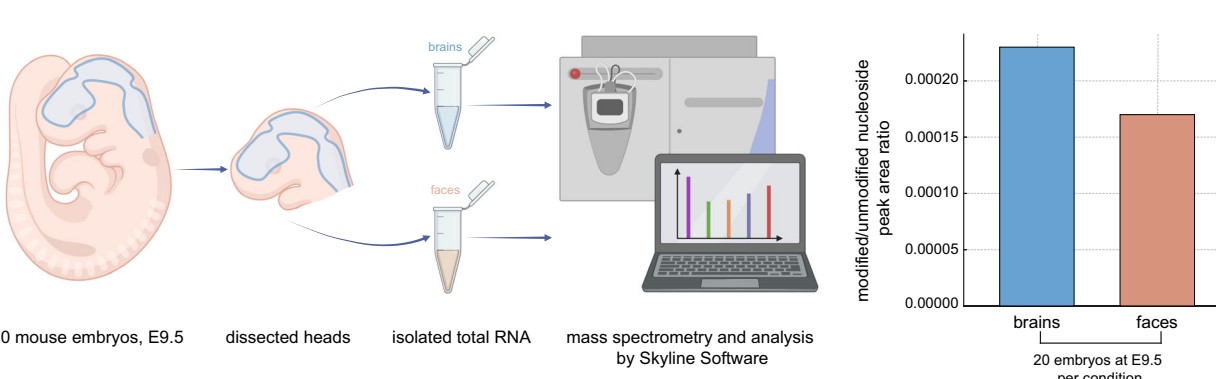

**Fig. 5 | U1248 modification was identified only on the transcriptomic level.**
**a** Search for rRNA variants in the GTEx database. **b** Nucleotide composition at position 1248 of 18S rDNA sequence in 10 individual human genomes (genomics data). **c** Nucleotide composition at position 1248 of 18S rRNA sequence in 8 individual human transcriptomes (bulk RNA-seq data). **d** Nucleotide composition at position 1248 of 18S rRNA sequence in different tissues (bulk RNA-seq data). Search

for modified nucleotides by mass-spectrometry: sample preparation (**e**) and histogram of measured modified/nonmodified peak area nucleotide ratio for m¹acp³ψ (**f**). **a**, **e** Were created using BioRender.com. Adameyko, I. (2026) https://BioRender.com/pvtj8ya (**a**). Adameyko, I. (2026) https://BioRender.com/2cryor7 (**e**). Source data are provided as a Source Data file.

structures (Supplementary Fig. 8). While control viral injections did not affect embryo morphology, both *Nhp2* and *Tsr3* overexpression resulted in gross morphological abnormalities throughout the whole embryo including the craniofacial complex (Supplementary Fig. 8). These abnormalities were severe, thus, suggesting that the levels of NHP2 and TSR3 must be tightly controlled during embryogenesis for correct and constructive craniofacial development. Although NHP2 and EMG1 are involved in several cellular processes in addition to catalyzing the initial steps of m¹acp³ψ modification, the only known role of TSR3 is catalyzing the last step of m¹acp³ψ modification in 18S rRNA[81]. Together, this supports the importance of m¹acp³ψ in control of developmental processes such as CNCC formation and differentiation.

To further validate this idea, we used the iGonad Cas9/CRISPR based strategy[82,83] to knock out exons 2–5 of the *Tsr3* gene in murine embryos. This resulted in a previously undescribed phenotype (Fig. 6a, b). The *Tsr3*−/− mutant embryos appeared dysmorphic at E10.5 compared to littermate controls (Fig. 6c and Supplementary Fig. 9), with delayed development of mesodermal and NCC lineages. Overall, the *Tsr3*−/− embryos appeared smaller in size compared to control littermates, with a notable exception of the heart, which remained comparable to the size of a normal heart for that age (Fig. 6c). Immunostaining for SOX10, SOX2, and 2H3 (also known as NEFM, a neurofilament marker for peripheral neurons) revealed that although the trigeminal ganglia were affected in the mutant embryos, they still formed successfully in expected locations (Fig. 6d). However, the facial structures were severely underdeveloped, with almost a total absence of anterior ectomesenchyme (Supplementary Fig. 9).

## Ribosomal control and rRNA modifications signature in neuroblastoma

We next questioned whether the m¹acp³ψ modification of U1248 in 18S rRNA, as well as the ribosomal control and rRNA modifications signature, correlated not only with the ectomesenchymal fate choice in CNCCs, but also with possible mesenchymal phenotypes emerging in neuroglial tumors from human patients. Recently, phenotypic plasticity of malignant cells has attracted substantial attention due to the onco-fetal transformation and drug resistance/metastatic properties of minor tumor cell populations[84,85]. The mesenchymal transcriptional state is often associated with the increased malignancy of cells that are specifically resistant to therapy and, thus, can be associated with a poor prognosis[16,86]. The neuroblast-to-mesenchymal transition might be an example of how tumor cells re-enact the embryonic programs, e.g., in neuroblastoma[10,87].

To address this question, we sequenced several neuroblastoma tumors using Smart-seq2 (Supplementary Data 4 for tumor classification details) and also re-analyzed published single cell datasets of glioblastoma and medulloblastoma tumors[88,89] (Supplementary Fig. 10a–f). We annotated the cell types in a neuroblastoma dataset based on established markers (Supplementary Fig. 10c, g–i) and kept the original cell annotation for re-investigated published datasets. Then we searched for the enrichment of our CNCC signatures, including ribosomal control and rRNA modifications in tumor cell subtypes (Supplementary Fig. 10a–c).

These analyses revealed that the investigated tumors contained diverse populations of malignant cells with increased and decreased mesenchymal gene expression modules, as determined via assessing the presence of "mesenchyme early" and "mesenchyme late" CNCCs programs (Supplementary Fig. 10a–c). This correlated with previous reports of such mesenchymal-like populations in neuroblastoma and glioblastoma[16,90,91]. Furthermore, the ribosomal control and rRNA modifications signature pattern was differentially distributed in relation to the pattern of "mesenchyme early" and "mesenchyme late" signatures (Supplementary Fig. 10a–c, Supplementary Data 1). Also, the analysis revealed a significant proportion of glioblastoma cells with

high levels of both "mesenchyme early" and "ribosomal control and rRNA modifications" signatures, which were expressed more strongly compared to stromal cell types (Supplementary Fig. 10d). Similarly, the Group 3 medulloblastoma subtype was significantly enriched for these signatures compared to other subtypes (SHH, WNT, Group 4) (Supplementary Fig. 10e). In the case of neuroblastoma, the tumor populations expressed varying degrees of ribosomal control and rRNA modifications signature (Supplementary Fig. 10f). Interestingly, a small proportion of neuroblastoma tumor cells exhibited a higher mesenchymal early signature (including ribosomal control and rRNA modifications signature part), though they never reached the values of truly mesenchymal stromal cells such as the adipocyte-like cell type (Supplementary Fig. 10f). Together, these results support a hypothesis that neuro-glial tumors are comprised of a spectrum of cell phenotypes with different degrees of mesenchymal state that correlate with ribosome modifications.

We next took advantage of Smart-seq2 coverage of sequenced neuroblastoma tumors to search for the U1248 modification and its distribution across cell types. The integration of tumor data derived from 7 patients resulted in six clusters of tumor cells as well as several stromal cell subtypes (Supplementary Fig. 10g–i). We calculated the ratio of reads with U1248 modifications for each cell in each sample and compared the distribution of these scores in different tumor cell and stromal clusters. Enrichment of the misread U1248 nucleotide in 18S rRNA was observed in two tumor clusters, of which cluster 5 predominantly belonged to a single patient, suggesting that this phenotype might not be universal. In contrast, the second enriched cluster comprised cells from four different patients and appeared to represent adrenergic tumor "bridge-like" cells[17] (Supplementary Fig. 10k, l). The uneven distribution of the U1248 modification across patients and tumor cell subtypes may suggest a correlation with the biological features and malignancy of tumors.

Therefore, we next investigated how the "early mesenchymal" (corresponding to the early CNCC ectomesenchymal bias), "ribosomal control and rRNA modifications", "ribosomal structural proteins", and "late mesenchymal" (corresponding to ectomesenchymal fate established after the fate split with neuro-glial fates in CNCCs) (Supplementary Data 1 and Figs. 1–2 for clarification) signatures correlated with survival of neuroblastoma patients from published cohorts (Fig. 7).

To this end, we estimated the strength of all signatures in tumor transcriptomic data from six neuroblastoma cohorts (Supplementary Data 5): Oberthuer et al., 2006 (*n* = 251)[92]; Versteeg cohort (*n* = 88)[93], TARGET-NBL (*n* = 133)[94], Westermann-ALT dataset (*n* = 144), which comprises also neuroblastomas with ATRX and TERT mutations[95], Kocak et al, 2013 (*n* = 476)[96] and SEQC (*n* = 498)[97,98]. The last cohort was analysed by both RNAseq (SEQC-GSE49711) and microarray (SEQC-GSE49710), both of which we included in our analyses.

Neuroblastoma is a heterogeneous tumor, and as such, many known covariates influence survival[99], including MYCN amplification status (i.e., the acquisition of several copies of this gene), tumor stage according to the International Neuroblastoma Staging System (INSS), and risk (usually integrating histopathology). To isolate the effect of the mesenchymal signatures on survival from that of established covariates, we first performed survival analysis on non-MYCN amplified subcohorts from later stages 3–4, or 4 alone (metastatic tumors). Initial survival analysis of the SEQC dataset (Fig. 7a) revealed that high levels of the mesenchymal early signature and particularly its sub-signature—the ribosomal control and rRNA modifications signature—were significantly associated with poorer prognosis in the subcohort. Deeper inspection of these signatures across other subdivisions of the SEQC dataset (Supplementary Fig. 11) revealed that the ribosomal control and rRNA modifications signature consistently predicted poorer prognosis with the exception of the MYCN-amplified subcohorts in this dataset. Ten out of twelve genes of the signature, on

**a**  Schematic of *Tsr3⁻ᐟ⁻* knockout and breeding strategy.

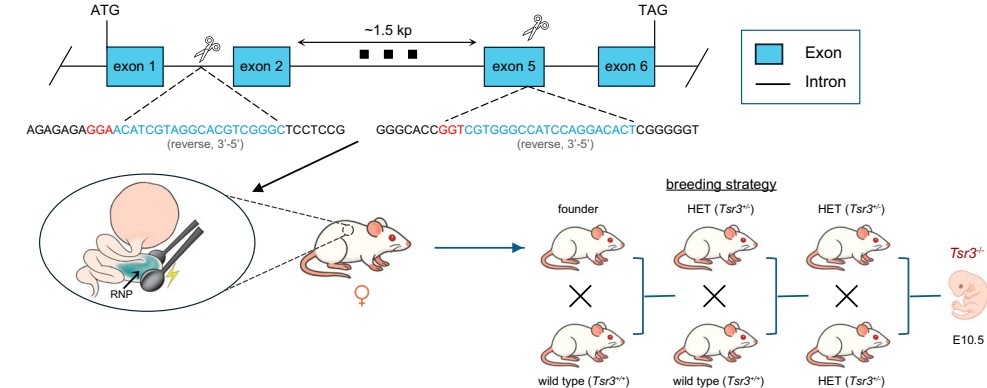

**b**  Confirmation of *Tsr3⁻ᐟ⁻* genotype by Sanger sequencing

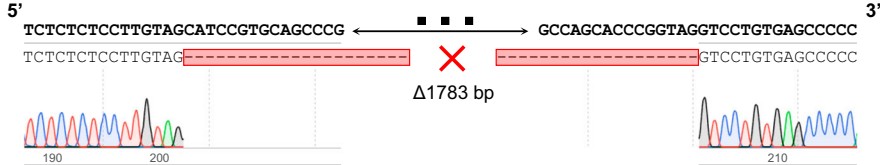

**c**  Phenotypes of *Tsr3* control and mutant mouse embryos at E10.5

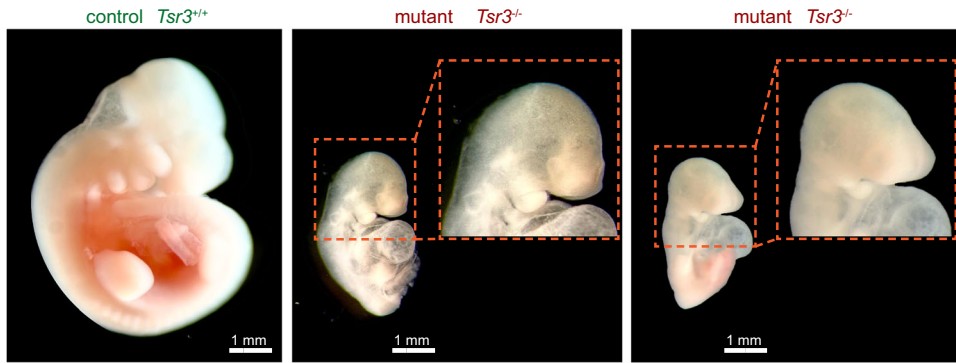

**d**  Whole-mount immunostaining of SOX2, SOX10, and 2H3 in *Tsr3* control and mutant mouse embryos at E10.5

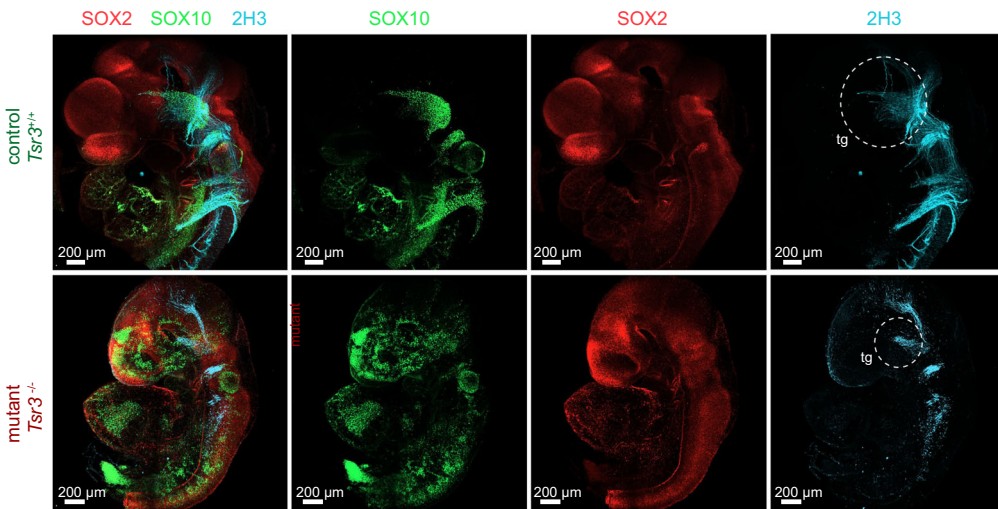

**Fig. 6 | *Tsr3⁻ᐟ⁻* knockout leads to regression of ectomesenchymal facial derivatives and body size. a** Schematic representation of the knockout and breeding strategy. A 1783 bp fragment between exon 2 and exon 5 was deleted using CRISPR/Cas9 via the i-GONAD method. Left and right homology arms flank the deleted region. PAM sequences and sgRNA target sites are indicated. **b** Sanger sequencing of the 5′ and 3′ junctions confirms correct recombination of flanking sequences.

Representative chromatograms show clean junction formation at both deletion breakpoints. **c** Brightfield images of *Tsr3* control and mutant mouse embryos at E10.5. **d** Whole-mount immunostainings for SOX2 (marker of developing central neural system), SOX10 (NC and glial marker) and 2H3 (marker of the peripheral nervous system) at E10.5 control and mutant mouse embryos. tg trigeminal ganglion.

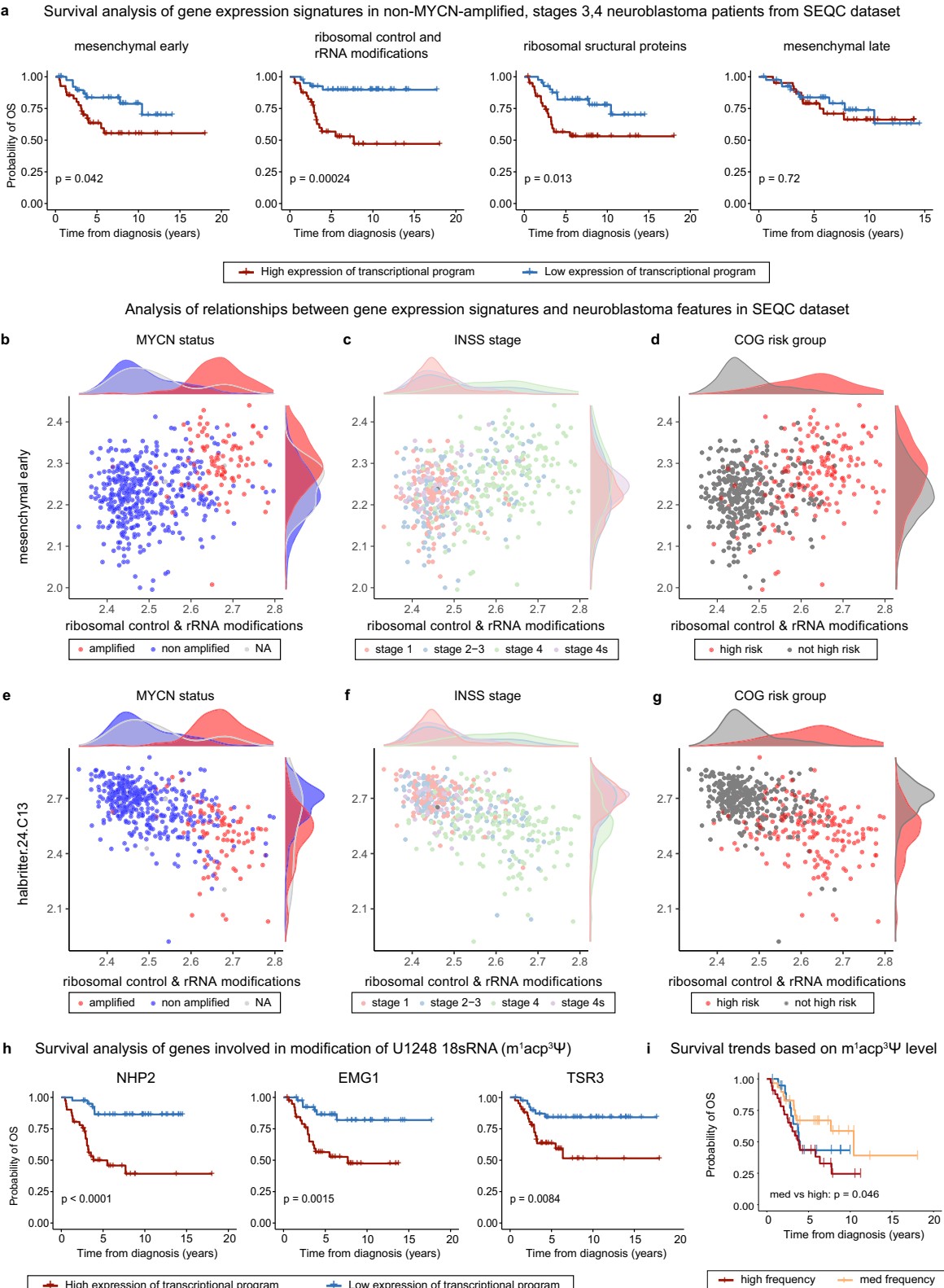

their own, were predictive of poorer outcomes in non-MYCN-amplified stage 3–4 patients (Supplementary Fig. 12).

We also observed that high levels of ribosomal control and rRNA modifications signature were associated with progressively later stages and with MYCN amplification (Fig. 7b, c), making it highly predictive of MYCN status, stage, and even other covariates (Fig. 7b–d, ROC AUC > 0.94 for MYCN status in SEQC database, Supplementary

Data 6). Furthermore, this signature was anticorrelated with a previously published signature, Halbritter C13, which was identified in an in vitro neuroblastoma model and was associated with favorable prognosis (Saldana-Guerrero et al., 2024) (Fig. 7e–g).

We then extended the analysis to five additional neuroblastoma patient cohorts (Supplementary Fig. 13). The ribosomal control and rRNA modifications signature again demonstrated strong predictive

**Fig. 7 | Ribosomal control and rRNA modification signature as a predictor of a poor outcome in neuroblastoma. a** Survival analysis of different gene signatures in non-*MYCN*-amplified, stages 3–4 (*n* = 163 samples) neuroblastoma subtypes from SEQC dataset (RNA-seq version SEQC-GSE49711 here and below). *P*-values were calculated by performing the two-sided log-rank test between Kaplan–Meier curves of overall survival (OS) for the top and the bottom quarter of tumors (i.e., 75% and 25% quartiles, 41–42 samples in each quartile) sorted by a gene signature expression. **b–d** Scatterplots of the ribosomal control & RNA modification signature score (X-axis) versus the mesenchymal early signature (Y-axis). Each point represent a tumor sample and is colored based on MYCN status (**b**), International Neuroblastoma Staging System - INSS stage (**c**), and COG risk group annotation (**d**). **e–g** Scatterplots showing the ribosomal control & RNA modification signature score (X-axis) versus the 'cluster 13' (C13) developmental signature (Y-axis) as a positive prognosis marker reported by (Saldana-Guerrero et al., 2024). Each point represents a tumor sample and is colored based on MYCN status (**e**), INSS stage (**f**), and COG risk group annotation (**g**). **h** Survival analysis of genes involved in modifications of U1248 18S rRNA in non-*MYCN*-amplified, stages 3–4 (*n* = 163 samples) neuroblastoma subtypes from SEQC dataset. **i** Survival analysis of U1248 post-modification level in non-*MYCN*-amplified, high-risk (*n* = 83 samples) neuroblastoma subtypes from SEQC dataset.

power for MYCN status across datasets (Supplementary Fig. 13a, b) and for stage 4 (except for the Westermann-ALT dataset). The signature also showed general opposition to the Halbritter C13 signature (Supplementary Fig. 13c). Moreover, it exhibited a consistent, statistically significant association with poor prognosis in non–MYCN-amplified stage 4 patients from these cohorts (Supplementary Fig. 13d). As in the SEQC dataset, the mesenchymal signatures showed no discriminatory power for survival within MYCN-amplified patients. This result suggests that small differences in the already high levels of the signature – particularly in MYCN-amplified patients, where signature expression is highest – may not contribute further to the strong effect of MYCN on negative prognosis. To extend these conclusions quantitatively, we estimated the proportional effects of different variables by applying a Cox proportional hazards model to stage 4 patient subcohorts. This analysis revealed consistently higher hazard associated with a high ribosomal control and rRNA modifications signature in almost all subcohorts, whereas other covariates were less consistent (Supplementary Fig. 15). Together, these findings suggest that the ribosomal control and rRNA modifications signature inform about unifying features of neuroblastoma biology concomitant with disease progression.

In contrast to the results of the ribosomal control and rRNA modifications signature, the late mesenchymal signature (Fig. 7a) and the signature containing transcripts encoding ribosomal structural proteins (see Supplementary Data 1) showed substantially less correlation with a poor survival rate of neuroblastoma patients and did not discriminate MYCN status as well (Fig. 7a and Supplementary Fig. 16a, Supplementary Data 6). This suggests the absolute numbers of ribosomes might not be as essential as the modification of the ribosomes for cancer prognosis and for the potential presence of the most malignant cell subpopulations within individual tumors. Furthermore, the ribosomal control and rRNA modifications signature is not strongly correlated to the S phase signature (Supplementary Fig. 16b), which seems to be generally associated with poor outcomes[100,101]. Interestingly, we also found that the ribosomal control and rRNA modifications signature predicts MYCN status more accurately than the cell cycle signature (Supplementary Fig. 16b).

Altogether, the ribosomal control and rRNA modifications signature was pervasively associated with poor outcomes across diverse patient cohorts. The signature provided additional survival information on top of established covariates and thus may help as an additional risk stratifier across neuroblastomas with diverse characteristics, such as mutational diversity (Hartlieb et al., 2021). This signature was particularly informative in non-*MYCN*-amplified cases, where risk is more variable and harder to elucidate.

Next we explored the impact of the U1248 modification on neuroblastoma patient survival. Although the expression of genes involved in introducing this modification correlated with poor outcomes (Fig. 7h and Supplementary Fig. 12), this does not directly confirm the impact of the modification itself. To resolve this issue, we estimated the frequency of modification using raw sequencing data and compared it with survival data of high risk neuroblastoma tumors in the SEQC cohort (high risk, non-MYCN-amplified). Notably, the tumors with medium levels of the modification had the best survival

outcomes compared to those with either low or high levels of the U1248 modification (Fig. 7i). This suggests that the ratios of $m^1acp^3\psi$ modification in tumors might reflect different programs in tumor cells and potentially can be linked to specific aspects of tumor cell behavior. This requires future studies and independent validations across a spectrum of tumor models.

## In vitro study of mesenchymal early genes in neuroblastoma cell lines

We next asked whether the ribosomal control and rRNA modifications signature also associates with aggressive neuroblastoma traits at the protein level. We focused on two representative ribosomal assembly proteins (based on high quality antibody availability), WDR74 and NHP2, together with markers of aggressive neuroblastoma phenotypes, mesenchymal neuroblastoma-associated prominin-1 (CD133)[102] and serotonin receptor HTR3A, a marker of bridge cells with potential implications for neuroblastoma etiology[103]. Intriguingly, screening a panel of 20 neuroblastoma cell lines revealed distinct expression patterns of the selected proteins (Fig. 8a, Supplementary Fig. 17a). By analyzing protein levels in detail, a strong positive correlation was observed between WDR74 and each of the three remaining proteins of interest (Fig. 8b). We therefore used the levels of WDR74 as a classifier, defining two distinct groups of cell lines, WDR74-high and WDR74-low (Fig. 8c). Notably, the WDR74-high group predominantly comprised cell lines derived from high-risk (HR) tumors (5/6 cases), including both MYCN/MYC-amplified and -non-amplified cases, whereas models from non-high-risk cases (Supplementary Fig. 18) were prevalently found in the WDR74-low group (Fig. 8c).

We therefore hypothesized that protein abundance patterns might correlate with specific aggressive characteristics of neuroblastoma cells. Indeed, as assessed by sphere formation assay, WDR74-high cells showed enhanced cancer stem-like traits in vitro (Fig. 8d). These data suggested that WDR74 alone could serve as a marker of at least a subset of aggressive neuroblastomas. To further explore this potential prognostic significance, we performed survival analysis using a publicly available bulk transcriptomic dataset, which confirmed that *WDR74* expression is significantly associated with poor outcomes (Supplementary Fig. 17b). Remarkably, in non-MYCN-amplified neuroblastoma patients - thus controlling for the potential confounding effects of MYCN amplification as a canonical marker of high-risk disease[104,105] - elevated WDR74 levels remained strongly associated with poor prognosis (Fig. 8e), underscoring the predictive power of a single gene from the ribosomal control and rRNA modifications signature.

Next we hypothesized that the elevated expression of ribosomal assembly proteins may reflect the reliance of neuroblastoma cells on high ribosomal production and processivity, making WDR74-high cells more vulnerable to overloading by ribosomal inhibition. Cycloheximide blocks ribosome translocation, effectively locking ribosomes onto mRNA, which leads to rapid depletion of the nuclear pools of ribosomal proteins, arrest of rRNA processing, and causes an overall increase in cellular demand for ribosomal assembly[106,107]. In line with this notion, treatment with cycloheximide dramatically limited

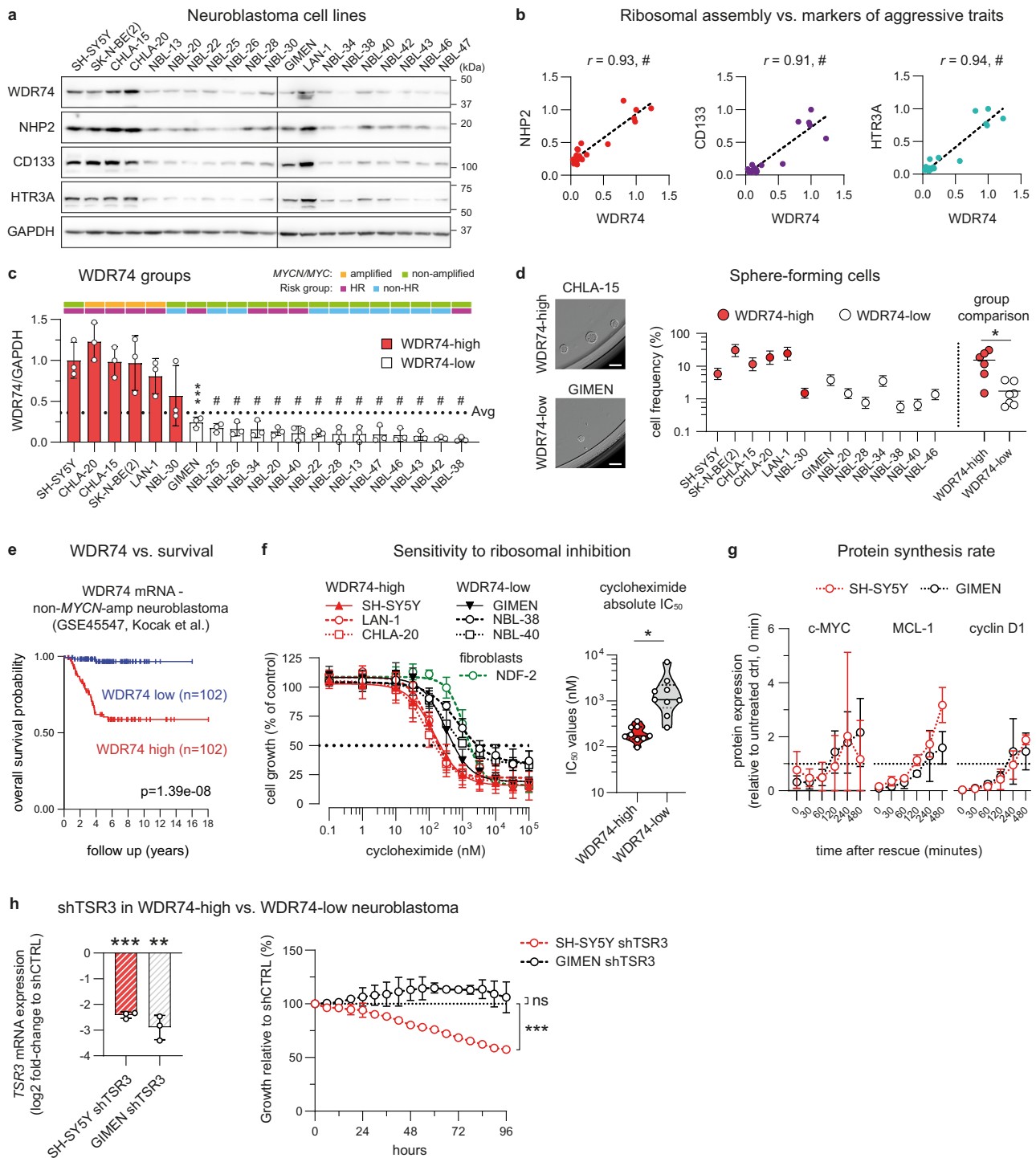

proliferation of WDR74-high cell lines, which were significantly more sensitive (2.5–20×) than WDR74-low counterparts (Fig. 8f and Supplementary Fig. 17c,d). Importantly, the different sensitivity to ribosomal inhibition could not be explained by differences in basal protein synthesis rates between WDR74-high and -low cell lines. Comparison of the time-dependent increase of short-turnover proteins after their depletion by cycloheximide pre-treatment revealed no consistent trend in the expression dynamics between WDR74-high SH-SY5Y (mesenchymal) and WDR74-low GIMEN (neuroblastic) cell lines (Fig. 8g). These results demonstrate that expression of genes from the ribosomal control and rRNA modifications signature does not directly determine protein synthesis rates in neuroblastoma cells. Instead, these genes are likely to be involved in other processes, such as RNA

metabolism and events associated with differential efficiency of peptide chain elongation, including the possibility of specialized ribosomal pools containing rRNAs with post-transcriptional modifications.

To further elucidate the importance of RNA metabolism in neuroblastoma cell physiology, we disrupted the expression of *TSR3* (see Fig. 4g). Because neuroblastoma cell lines with *TSR3* gene knockout did not survive in our conditions, we instead established neuroblastoma models with stable *TSR3* knockdown (shTSR3) derived from SH-SY5Y (WDR74-high; mesenchymal) and GIMEN (WDR74-low; neuroblastic) cell lines (Fig. 8h). Although both WDR74-high and WDR74-low shTSR3 models exhibited a comparable >4-fold decrease in *TSR3* expression, only the WDR74-high cells exhibited markedly reduced growth (Fig. 8h). These experiments revealed that adequate levels of

**Fig. 8 | High levels of ribosomal assembly proteins associate with aggressive neuroblastoma cell phenotypes. a** Protein expression analysis by western blotting in a panel of human neuroblastoma cell lines. Note the apparent expression patterns among the cell lines. Blots are representative of three experiments. **b** Correlation of ribosomal assembly protein WDR74 with other three examined proteins - ribosomal assembly NHP2, and markers of aggressive stem-like neuroblastoma cells, CD133 and HTR3A. Data points represent the mean (biological $n = 3$) of relative protein expression in individual cell lines (densitometric analyses provided in Supplementary Fig. 17a). For each plot, Pearson's correlation coefficient ($r$) is shown, #$p < 0.0001$. **c** Densitometric analysis of the WDR74 protein levels. Two distinct groups of cell lines can be defined based on the average level of WDR74 (dotted line) across the neuroblastoma panel (risk group and MYCN/MYC status for the respective original tumors are indicated). Data presented as mean ± SD, biological $n = 3$. Statistically significant differences compared with SH-SY5Y cells are determined by one-way ANOVA followed by Tukey's post hoc test, ***$p < 0.001$, #$p < 0.0001$. **d** Limiting dilution sphere formation assay revealed significant differences in the frequencies of sphere-forming cells between WDR74-high and WDR74-low cell lines. Left panel, representative images of spheres formed by WDR74-high and -low neuroblastoma cells; bar, 100 μm. Right panel, data are shown as mean ± 95% confidence interval, biological $n = 3$. Cell frequencies and probability were computed using ELDA software. Statistical significance was validated by group comparison using an unpaired two-tailed Welch's $t$-test, *$p < 0.05$. **e** Survival analysis of WDR74 mRNA expression in non-*MYCN*-amplified neuroblastoma tumors ($n = 405$ samples) using R2 platform. Kaplan–Meier curves were plotted for the top (WDR74 high) and bottom (WDR74 low) quartiles, and the $p$ value was calculated using a two-sided log-rank test. **f** Cell growth after 3-day treatment with cycloheximide. Left panel, dose-response curves shown as mean ± SD relative to untreated control, biological $n = 3$, technical $n = 5$. Note the marked difference in sensitivity of WDR74-high and WDR74-low cell lines (detailed analysis in Supplementary Fig. 17c). Normal, healthy fibroblasts NDF-2 served as an internal control to exclude the potential general cytotoxicity of effective cycloheximide concentrations. Right panel, violin plots of the absolute half-maximal inhibitory concentration ($IC_{50}$) values; *$p < 0.05$, unpaired two-tailed Welch's $t$-test. Points represent calculated $IC_{50}$ values of individual biological replicates of each WDR74-high or WDR74-low cell line analyzed. **g** Synthesis rate of proteins with short turnover. The capacity of neuroblastoma cells to restore levels of the examined proteins after the withdrawal of translation elongation inhibitor cycloheximide is not significantly different between WDR74-high (red) and WDR74-low (black) cells. Data presented as mean ± SD, biological $n = 3$. **h** Effects of stable *TSR3* knockdown (shTSR3) in WDR74-high SH-SY5Y and WDR74-low GIMEN cells. Left, qRT-PCR analysis of the mean normalized *TSR3* mRNA levels in shTSR3 models plotted relative to the scramble shRNA (shCTRL) controls. Data shown as mean ± SD, biological $n = 3$, technical $n = 3$; **$p < 0.01$, ***$p < 0.001$, unpaired two-tailed Welch's $t$-test. Right, live-cell imaging growth rate analysis of shTSR3 models compared with their shCTRL controls over 96 h following the day after seeding. Data plotted as mean ± SD; biological $n = 3$, technical $n \geq 10$. Statistical significance at the 96-h time point was determined by unpaired two-tailed Welch's $t$-test, ***$p < 0.001$, ns not significant. Source data are provided as a Source Data file.

TSR3 contribute to neuroblastoma cell cycle progression, which in turn accentuates ribosome biogenesis roles beyond protein synthesis.

## Discussion

NCCs comprise the key multipotent progenitor cell population that builds the face, parts of the heart, sensory systems, enteric and other autonomic neurons, and establishes pigmentation during vertebrate development[2]. The questions of how NCCs execute their multipotency and decide their future fates[9,13,14] is important for understanding the mechanisms of congenital disorders, including numerous craniofacial syndromes[14,59,65,108]. Furthermore, in some circumstances, NCC lineage also gives rise to pediatric tumors such as neuroblastoma, which exhibits cellular and molecular characteristics reminiscent of developing NCC-derived subpopulations[17,109,110].

Fate selection in trunk NCCs has previously been extensively analyzed[9,10], but how anterior, face-forming CNCCs choose their fates is unclear. Using deep Smart-seq2 profiling[48] of HOX-negative CNCCs, we mapped an early bifurcation between skeletogenic ectomesenchyme and neuro-glial lineages during cranial migration. Delaminating and immediate post-delaminating cranial NCCs are strongly ectomesenchymal-biased, in contrast to trunk NCCs, which predominantly generate neuro-glial progeny.

Transcriptional bias toward ectomesenchyme resolved into two major clusters of functionally integrated protein-coding genes: EMT-associated factors (*Snai1, Twist1, Mmp2*, etc) and ribosomal control and rRNA modifications genes (*Nhp2, Rrp8, Emg1, Wdr74, Ddx31*, others). This fits the known CNCC sensitivity to ribosome dysfunction in craniofacial ribosomopathies such as Shwachman-Diamond and Treacher Collins syndromes, validated in mouse and zebrafish models[26,108,111–114], but these effects had not been linked to fate decisions in multipotent CNCCs[13]. Our data indicate that rRNA synthesis and maturation, including post-transcriptional modification, are critical for fate selection, cell-type proportioning and expansion of CNCC progeny during craniofacial development.

Our analysis of pro-ectomesenchymal proteins highlighted multiple regulators of rRNA synthesis and processing, including RRP8, a component of the energy-dependent nucleolar silencing complex that couples rRNA production to cellular energy status (Murayama et al., 2008). To test whether rRNA levels influence CNCC fate and ectomesenchymal skeletogenic potential, we temporally knocked out Polr1a and Polr1c in murine embryos using R26-CreERT2 at the stage when ectomesenchymal and neuro-glial fates diverge. 3D craniofacial reconstructions and single-cell analyses revealed disrupted ectomesenchyme and reduced skeletogenic condensations, with comparatively mild effects on neuro-glial populations, indicating a localized fate imbalance distinct from the severe defects seen with earlier CNCC knockouts.

This led us to ask how ribosome-related processes might modulate CNCC bias towards ectomesenchyme. We considered (i) specialized ribosomes generated by specific rRNA processing events[40] versus (ii) increased ribosome quantity in rapidly dividing pro-mesenchymal CNCCs. Structural ribosomal protein genes were not significantly upregulated in ectomesenchyme-biased CNCCs, and their expression was comparable across proliferating CNCCs and derivatives, including mature ectomesenchyme, arguing against a simple increase in ribosome numbers. Instead, our findings support a central role for ribosome/rRNA modification, consistent with evidence for specialized ribosome pools that preferentially translate specific mRNA classes[29,36,42,45,115,116] and with rRNA modification–dependent control of neuronal differentiation[117].

Within our ribosomal control and rRNA modifications signature, we identified two proteins NHP2[61] and EMG1[60], which act sequentially to introduce pseudouridine-based rRNA modifications and regulate 40S subunit assembly. Such modifications can generate specialized ribosomes, as illustrated by EMG1 methyltransferase activity supporting high translation of Kaposi's sarcoma-associated herpesvirus ORFs[118]. EMG1 mutations in Bowen-Conradi syndrome, characterized by facial anomalies including a high-bridged nose and micrognathia[119], further link the levels of EMG1 to craniofacial ribosomopathies. Exploiting the full-length coverage of Smart-seq2, we directly probed uridine rRNA modification in CNCCs and detected a stereotypical nucleotide mis-read at position U1248 in 18S rRNA, significantly enriched in individual CNCC transcriptomes.

This specific position is known to be heavily post-transcriptionally modified, with uridine converted into $m^1acp^3\psi$ through the concerted action of NHP2 and EMG1, thereby altering ribosomal dynamics[68,71]. We ruled out the potential existence of rDNA genomic polymorphisms with nucleotide variants in position U1248 by cross-analyzing the GTEx cohort data with matched genomes and transcriptomes from different tissues and organs. In all cases, the genomes of investigated subjects did not contain any variants in the U1248 position, whereas the transcriptional data revealed systematic mis-readings in the same position.

This confirmed that such complex modification affects Watson-Crick nucleotide base pairing[73]. Different proportions of the post-transcriptionally modified U1248 corresponded to the specific organ or tissue types, thus implying that specific cell types require exact proportions of the modified ribosomes. This is in line with a recent study showing that modifications of rRNA are always present to some degree in tissues and organs[120]. Importantly, the GTEx cohort has been sequenced with a different bulk sequencing approach as compared to single cell-based Smart-seq2, which adds confidence to our conclusions about the posttranscriptional nature of the U1248 modification. Our results also align with a recent study[39] showing that, unlike 28S rRNA, the molecular structure of eukaryotic 18S rRNA does not easily tolerate nucleotide polymorphisms encoded at the level of the genome. Finally, we validated the presence of the diverse proportions of m1acp3ψ modification in U1248 of 18S rRNA in murine embryonic faces and developing brains via mass spectrometry.

The functions of U-modifications in rRNA include modulating the codon-decoding processes during translation, as they can influence the fidelity of codon-anticodon interactions by affecting the local conformation and flexibility of 18S and 25S rRNA[71,121,122]. This modulation of the decoding process helps to ensure accurate translation of the genetic code of different mRNA molecules and prevents errors in protein synthesis[68,123].

Thus, we demonstrate that the proportions of m1acp3ψ modification correlate with the CNCCs developmental trajectory and transitions between subpopulations, underpinning the potential biasing of this modification in fate selection towards ectomesenchyme. To functionally validate the importance of the corresponding 18S rRNA modification process, we overexpressed *Emg1*, *Nhp2* and *Tsr3* (the third enzyme introducing m1acp3ψ in U1248) in vitro and also in vivo in CNCCs. This resulted in distorted proportions of CNCC-derived cell types and major facial abnormalities, and was consistent with the pathologically under-developed mesenchymal part of the faces in *Tsr3* gene knockout murine embryos, which we generated to test the importance of the very last step of m1acp3ψ synthesis of U1248 in 18S rRNA. Together, these observations demonstrate that the complex m1acp3ψ pseudouridine rRNA modification is important for correct CNCC development.

We therefore hypothesized that this specific rRNA modification might also play a functional role in NCC-derived neuroblastoma tumors, driving the plasticity of intra-tumoral heterogeneity and development-like transitions between neuroblastic and mesenchyme-like phenotypes[16,102,124], much like during the fate selection of cranial neural crest cells[10]. To address this question, we tested the correlation between the ribosomal control and rRNA modifications gene expression signature with specific subpopulations in neuroblastoma and other neuro-glial tumors. Indeed, human neuroblastoma, medulloblastoma and glioblastoma tumors exhibited the differential presence of the rRNA modification and ribosomal control signature in different cell subtypes. This is supported by previous findings connecting the loss of m1acp3ψ modification in U1248 18S rRNA with the features of colon cancer, and the characteristics of more than 22 other types of tumors[71].

In further support of these ideas, we performed the correlation analysis of neuroblastoma patient survival and the ribosomal control and rRNA modifications gene signature. Across six independent cohorts, this signature consistently correlated with adverse clinical outcomes, particularly in non-MYCN-amplified tumors. Unlike a structural ribosomal protein signature with limited prognostic value, it robustly predicted high-risk features (INSS stage, MYCN amplification) and added prognostic power beyond established clinical covariates, remaining a strong predictor within stratified subgroups and across mutational contexts.

To explore a potential role for 18S rRNA U1248 modification, we re-analyzed raw reads from a neuroblastoma cohort (SEQC-GSE49711)

and found that deviations from the average U1248 modification level correlated with poor outcome in non-MYCN-amplified high-risk patients. This aligns with TSR3 knockdown in mesenchymal and neuroblastic neuroblastoma lines, where only mesenchymal-like cells showed reduced proliferation. Together, this points to a biologically meaningful role of TSR3 and m1acp3ψ modification of U1248 in 18S rRNA in cells acquiring a mesenchymal state phenotype in embryos and possibly in neuroblastoma, although more direct evidence is needed to link this modification to malignant traits.

It is important to note that the ribosomal control and rRNA modification signature may exert additional effects beyond post-transcriptional modifications of rRNA. Indeed, in multiple neuroblastoma cell lines, WDR74 expression (a factor in long rRNA precursor cleavage) correlated with aggressiveness and mesenchymal status of cultured tumor cells.

Moreover, our results from reversible protein synthesis inhibition add weight to the emerging roles of the ribosomal assembly control proteins beyond general assembly of ribosomes, such as playing a role in shaping specialized ribosomal pools, which might require both protein and rRNA variants[37,38], as well as modifications[32,40].

Although this study sheds light on the role of ribosomal modifications in embryonic development and highlights their potential importance in neuroblastoma, there are limitations, which must be acknowledged. For example, the rRNA and ribosome control signature includes proteins that might have additional, as yet unknown, functions, consistent with evidence emerging for non-canonical roles of ribosomal proteins[125,126] or even snoRNAs such as SNORA13[57] that beside guiding RNA modification, also promote p53-mediated senescence by directly binding to RPL23 and decreasing its incorporation into maturing 60S subunits.

These other roles might also contribute to mesenchymal biasing of CNCC, as well as to the biology of neuroblastoma. Furthermore, ribosome maintenance in vivo involves both the exchange of individual ribosomal proteins and complete ribosome turnover[127–129]. More specifically, peripheral structural ribosomal proteins, as well as those directly involved in peptide bond formation, are dynamically replaced multiple times throughout the lifespan of the ribosome - likely through exchange with freely available cytoplasmic protein pools[128]. In contrast, most core ribosomal proteins remain stably integrated and persist for the duration of the ribosome's functional life[128]. The exchange of ribosomal proteins seems to be an important mechanism for ribosome repair[130,131]. How this translates to the dynamics of possible rRNA modifications during the lifetime of a single ribosome currently remains unknown. Our results might be relevant to such questions, as we do not observe changes in the expression of genes encoding structural ribosomal proteins in cases where, for example, rRNA modifying enzymes appear to be differentially expressed.

## Methods

### Ethics declarations

All ethical permits for animal work and human studies were approved by the Austrian, Swedish, US, and Czech responsible agencies and appropriate institutional committees as applicable (see below for details).

Animal research as part of this study was approved by the Ethical Committee on Animal Experiments (Stockholm North committee) and performed in compliance with the recommendations outlined in The Swedish Animal Agency's Provisions and Guidelines for Animal Experimentation. Laboratory animals were kept in standardized conditions (24 °C, 12:12 h light−dark cycle, 40−60% humidity, food, and water ad libitum).

Animal research as part of this study (*Tsr3*-deficient mice) was conducted in accordance with the Swedish Animal Welfare Act and was approved by the Regional Animal Ethics Committee in Gothenburg (Göteborgs djurförsöksetiska nämnd), with

regulatory oversight by the County Administrative Board of Västra Götaland County (Länsstyrelsen i Västra Götalands län) and, where applicable, authorization for contained use of genetically modified animals by the Swedish Board of Agriculture (Jordbruksverket).

Animal research as part of this study (*Polr1a*$^{flx/flx}$ mice and *Polr1c*$^{flx/flx}$ mice) was performed in accordance with a Stowers Institute for Medical Research Institutional Animal Care and Use Committee (IACUC) approved protocol (#2022-143). All mice were housed in a 14-hour light/10-hour dark/light cycle.

Ethical permits for neuroblastoma samples (009/1369-31/1 and 03-736) were issued to Professor P. Kogner. Neuroblastoma tumor samples from all patients were collected after obtaining verbal or written consent from parents or guardians depending on IRB requirements at the time of sampling. All samples were collected and analyzed according to permits approved by the Karolinska Institutet and the Karolinska University Hospital ethics committees (reference numbers 03-736 and 2009/1369-31/1) in agreement with the Declaration of Helsinki. Sampling was performed during clinical routine procedures, treatment and management were performed according to established national and international protocols, and hospital records were used to retrieve clinical data.

Neuroblastoma cell lines were established in Dr Jan Skoda's laboratory from tumor tissues of neuroblastoma patients treated at the Department of Pediatric Oncology, University Hospital Brno. Written informed consents were obtained under research projects (IGA MZCR NR/9125-4, AZV MZCR 15-34621A, and MU 0829/2018) approved by Masaryk University's research ethics bodies (approval no. 23/2005, 26/08/2014, and EKV-2018-075, respectively).

## Mouse cranial dataset analysis

**Single-cell dataset overview.** The mouse cranial neural crest datasets at E8.5, E9.5, and E10.5, along with mouse trunk postotic neural crest datasets at E9.5 and E10.5 were previously generated by our laboratory using Smart-seq2 protocol and published in[9] and[10]. Their GEO IDs are GSE201257 and GSE129114, respectively.

**Count matrices generation, QC and filtering.** The gene expression matrices for mouse cranial and trunk postotic datasets were processed and analyzed using the *Seurat* package[132]. To filter out low-quality cells, each dataset was processed separately. The cells were filtered to remove those that have fewer than 5500 detected genes. The cells with more than 9500 and 10000 genes in cranial and trunk postotic datasets, respectively, were also excluded as putative doublets. We also removed the cells with fewer than 7500 or more than 75000 total counts as well as the cells with more than 10% of mitochondrial counts 25% of ERCC. In addition, we filtered out the genes detected in fewer than 10 cells. To predict the potential doublet cells, Scrublet[133] was used before the filtration and log-normalization steps.

**Clustering and cluster annotation.** To combine the data for the same lineage at different timepoints, the mouse cranial datasets were merged into one "mouse cranial" dataset by using the merge function from the Seurat package, while the mouse trunk postotic NC at E9.5 and E10.5 datasets were integrated into the joint mouse trunk postotic dataset. Then PCA for each dataset was estimated on top 2000 variable genes obtained after normalization and log-transformation. A graph-based clustering with visualization by UMAP was performed on the first 10 principal components and 25 nearest neighbors for mouse cranial, and on 15 PC and 30 nearest neighbors for the mouse trunk postotic dataset. For cell-type annotation, the lists of well-established marker genes were used for each cluster in each dataset. To visualize the results, the python packages Scanpy[134] and scvelo[135] were used.

**RNA velocity estimation.** To calculate spliced and unspliced transcript counts, a command-line tool Velocyto[136] with run10x and run_smartseq2 commands was used for human and mouse datasets, respectively. The obtained loom files were processed by python package scvelo[135] to predict the directions of transcriptional changes. Gene selection for RNA velocity estimation was done automatically with the default parameters for mouse cranial and mouse trunk postotic. The first and second-order moments (i.e., means and uncentered variances) required for velocity estimation were computed among 30 nearest neighbors in PCA space, which was performed on the spliced matrix, keeping 20 and 15 principal components for the mouse cranial and trunk postotic datasets, respectively. Default parameters and dynamical model were used for computing velocity vectors and velocity graph on previously obtained UMAP embeddings.

**Regulon analysis.** The analysis of regulatory activities of transcription factors (TF) was performed by the SCENIC pipeline (the python package pySCENIC 0.11.2)[137,138] as well as DOROTHEA[139]. For SCENIC that consists of three steps, GRNBoost method was applied on the normalized expression matrix of each dataset for the identification of co-expressed gene sets that could be potential TF targets. Search for candidate TF for each such gene set required the enrichment of TF-binding motifs in the corresponding promoter regions of the genes which was done by using cisTarget (mm10 500bpUp100Dw and TSS + /-10kbp) as motif databases. Lastly, the AUC scores reflecting the regulon activity in each cell were calculated and applied to the corresponding UMAP embeddings.

**Trajectory and bifurcation analysis.** The developmental trajectory was analyzed only on early NCCs in mouse datasets that originally contain also NT lineage. The tree (trajectory) construction was performed with a python package scFates v0.2[20]. Tree fitting was done on UMAP embedding by scFates function tl.tree using the SimplePPT approach[140] (parameters for mouse cranial: method = "ppt", sigma = 0.75, lambda = 1000, metric='euclidean'). The tree root was selected at the tip in the delaminating or pre-migratory NC. Functions tl.test_association and tl.fit with default parameters were applied to identify and fit the genes that significantly change their expression along the trajectory.

The analysis of bifurcation point between ectomesenchymal and neuroglial cell fates in the mouse cranial dataset (see Fig. 1) was performed following the corresponding scFates pipeline that mainly consists of the following steps: detecting the differentially upregulated genes after bifurcation point, assigning them to a specific post bifurcation branch, and further classifying them as early (activated before the bifurcation) or late (activated after the bifurcation) genes. The threshold between early and late genes was used as the pseudotime point at the bifurcation node of the trajectory with the offset value equal to 1. To check whether the branch specific molecular programs are competing prior to the bifurcation, the local intra- and inter-module correlations using the early genes were calculated with scFates functions tl.slide_cells and tl.slide_cors.

**STRING.** The graph of the predicted protein-protein interactions among the gene signature was obtained from the STRING database, version 11.5[50]. The visualization of the graph was manually edited with python scripts and Adobe Illustrator.

## Whole mount in situ hybridization for *Nhp2* and *Sox10* in mouse embryos

The procedure was conducted in accordance with the Molecular Instruments protocol. (https://files.molecularinstruments.com/MI-Protocol-RNAFISH-Mouse-Rev9.pdf). For embryo collection, euthanasia was performed via isoflurane overdose followed by cervical

dislocation. Embryos were fixed in 4% paraformaldehyde at +4 °C for 3 h. After being washed with PBST, they were dehydrated with graded MeOH/PBST washes and incubated at −20 °C until use. Subsequently, after rehydration, embryos were digested with 10 µg/ml proteinase K for 15 min at room temperature, postfixed with 4% PFA, and washed with PBST. For the Multiplexed HCR™ RNA-FISH protocol, embryos were incubated in probe hybridization buffer (digoxigenin-labeled riboprobes transcribed from plasmids containing Nhp2, and Sox10 cDNAs), pre-hybridized, and hybridized with probe solutions overnight at 37 °C. Excess probes were removed by washing with probe wash buffer and 5× SSCT. The amplification stage involved pre-amplifying embryos and incubating them with hairpin solutions overnight at room temperature. After washing with 5× SSCT, samples were imaged using a confocal LSM800 microscope.

## Microscopy
Images were acquired using LSM 800, 900 confocal microscopes (Carl Zeiss, Germany) equipped with ×20 and ×40 objectives.

## In vivo conditional knockout of Polr1a and Polr1c
**Mice and animal husbandry.** The animal protocol for this work was approved by the Stowers Institute for Medical Research Institutional Animal Care and Use Committee (IACUC)- #2022-143.

*Polr1a*flx/flx mice and *Polr1c* flx/flx mice were generated as previously described in (Falcon et al., 2022). These mice were crossed to *R26R-EYFP* (Jax strain #006148) reporter mice to generate *Polr1a*flx/flx*;YFP* and *Polr1c* flx/flx*;YFP* mice. In this reporter line, removal of the loxP site flanked stop cassette by Cre recombinase activates EYFP expression (Soriano, 1999; Srinivas et al., 2001).

Tamoxifen-inducible temporal knockout *Cre-ERT2* (B6.129 – Gt(ROSA)26Sor™ 1(Cre−ERT2)Tyj /J, Jax strain# 008463) mice were crossed to *Polr1a*flx/flx and *Polr1c*flx/flx mice to generate *Polr1a*flx/+*;Cre-ER*T2 and *Polr1c*flx/+*;Cre-ER*T2 mice which were subsequently bred to homozygous *floxed YFP*+ mice to generate experimental embryos. *Cre Recombinase* was activated via oral gavage of pregnant dams with 100 µL of 2 mg Tamoxifen/0.4 mg progesterone (Sigma Aldrich) for *Polr1a* or 100 uL of 5 mg Tamoxifen/1 mg progesterone for *Porl1c*, prepared in corn oil at E9.0–9.5. Tamoxifen treatment induced high eYFP expression in the whole embryo, indicating near complete recombination and no toxicity. To confirm recombination, *Polr1a* and *Porl1c* mice were genotyped using real-time PCR assays with specific Taqman probes designed for each strain (Transnetyx, Inc, Cordova, TN).

The day a vaginal plug was observed in a time mated female was designated as embryonic day (E) 0.5. All mice were housed in a 14-h light/10-h dark/light cycle.

Availability of these mouse lines and any additional protocol details should be requested from the Prof. Paul Trainor lab.

**Single-cell RNA sequencing.** On the day of embryo harvest, pregnant female mice were euthanized by CO2 inhalation overdose and/or cervical dislocation. Eight *Polr1a*flx; Cre-ERT2 embryos were collected from 2 different litters, and after isolating the embryos, craniofacial tissue was micro-dissected, weighed and placed in 10x Genomics fixation solution while genotyping was performed to include 4 controls and 4 mutants from both litters. The tissue samples (~25 mg) were then fixed according to 10x Genomics demonstrated protocol CG000553 (revision B) and incubated for 20 h at 4 °C. Following fixation, the tissue was dissociated into a single cell suspension. Cell concentration and viability were assessed using a Luna-FL cell counter (Logos Biosystems) prior to storage at −80 °C for two weeks.

After storage, cell concentrations and viability were assessed again using the Luna-FL cell counter, and 1–1.5 million cells per sample were processed using the Chromium Fixed RNA Profiling for Multiplexed Samples user guide (CG000527, revision D). 10x Genomics Mouse Transcriptome Probes (PN-1000491) were hybridized to each sample for 20hrs at 42c. Samples with unique probes were equally pooled based on cell concentrations in sets of four then loaded on the Chromium iX Series instrument (10x Genomics) according to manufacturer's directions. Resulting libraries were checked for quality and quantity using a 2100 Bioanalyzer (Agilent Technologies) and Qubit 2.0 Fluorometer (Thermo Fisher Scientific). Samples were multiplexed into 2 pools and randomized to control for litter and sample (pool 1: wt1-litter1, mutant1-litter1, wt1-litter2, mutant1-litter2, pool 2: wt2-litter1, mutant2-litter1, wt2-litter2, mutant2-litter2).

Libraries were pooled and converted using the Element Biosciences Adept Rapid PCR-Free Protocol for use with the Element Adept Library Compatibility Kit v1.1 before sequencing on the Element AVITI instrument. The pool was sequenced to a depth necessary to achieve at least 13,000 mean reads per cell on an Element Biosciences Cloudbreak flow cell utilizing instrument software versions current at the time of processing with the following paired read lengths: 10*10*29*91 bp. Cell capture was estimated at ~9,300-14,300 cells per sample.

Raw sequencing data was processed using CellRanger (v4.0.0)[141] for demultiplexing, alignment, filtering, barcode and UMI counting, and after mitochondria and other feature thresholding, the final dataset used for analysis consisted of 113,062 total cells. R (v4.2.0) was used for downstream analysis and the Seurat package was used to normalize data and perform clustering. Compositional analysis of single-cell data was performed using scProportionTest[142] and scCODA[143].

**Immunohistochemistry.** Embryos were dissected and fixed in 4% PFA (in PBS) overnight, then transferred to 70% ethanol and embedded in paraffin and sectioned in frontal/coronal sections at 10 um thickness. Antigen retrieval was performed in citrate buffer for 15 min and sections were permeabilized three times for 15 min in 3% PBS-Triton-X100 (PBT) and blocked in 5% goat serum + 1% AffiniPure Fab Fragment Donkey Anti-Mouse (Jackson ImmunoResearch, #715-007-003) for 2 h at room temperature. Sections were then incubated in primary antibody overnight at 4 °C and the following primary antibodies were used: Sox9 (1:500, MilliporeSigma # AB5535) and Tuj1 (anti-Tubulin β 3 (TUBB3) 1:2000, BioLegend #801201). Sections were then washed three times for 15 min in PBT at room temperature, before diluted secondary antibody solution was added and sections were incubated for 1 h in the dark at room temperature. Sections were then washed three times for 15 min in PBT and treated with TrueBlack Lipofuscin Autofluorescence Quencher (Biotium #23007) before mounting in VECTASHIELD Antifade Mounting Medium (Vector Laboratories # H-1200-10). Secondary antibodies included Alexa Fluor 546 Goat anti-Rabbit (1:500, Invitrogen, # A-11035), 633 Goat anti-Mouse (1:500, Invitrogen, # A-21052) and DAPI (1:1200, Sigma-Aldrich, #D9564).

**Whole mount HCR in situ.** Whole mount fluorescent in situ hybridization was conducted using HCR v3.0 reagents (Molecular Instruments). Embryos were fixed overnight at 4 °C in 4% PFA and dehydrated and rehydrated with MeOH/PBST (0.1% Tween-20). Embryos were then treated with 10 µg/mL proteinase K for 20 min without rocking, postfixed in PFA and incubated in 4 nM probe overnight at 37 °C. Subsequently, samples were incubated in 30 pmol hairpin overnight at room temperature, washed and stained with 1:500 DAPI solution, embedded in ultra low-melt agarose and cleared in CeD3 + [144]. Imaging was performed on a Zeiss LSM 980 confocal microscope.

**MicroCT measurement.** MicroCT experiment was performed using the GE phoenix v|tome|x L 240 system equipped with a nanofocus X-ray tube (180 kV/15 W maximal power) and flat panel detector (dynamic 41 | 100 with 4000 × 4000 pixels, each 100 × 100 µm in size). Eight E12.5 embryos were measured in two CT scans. Scanning

parameters were same for both measurements: Setting of the X-ray tube was 80 kV and 200 μA. Exposition time was 600 ms and three projections were captured in each position to reduce the noise in the data; 1800 projections were acquired over 360°. The isotropic voxel size was 6 μm for all samples. The tomographic reconstructions were performed in the GE phoenix datos|x 2.0 3D computed tomography software. Segmentation of mesenchymal condensation and ganglia was performed manually using a combination of the Avizo (Thermo Fisher Scientific, USA) and VG Studio MAX 3.2 software (Volume Graphics GmbH, Germany)[66,145]. Quantitative analysis was performed in VG Studio MAX 3.2 software; For analysis of the dimensions, 15 uniform points were determined on each embryo and 8 distances were measured in 3D in software VG Studio.

### Search for variants in 18S ribosomal reads in mouse datasets

The raw Fastq files were additionally trimmed with bbduk.sh from BBMap aligner tool[146] to remove any potential technical sequences. The reference genome was built by concatenating mouse genome from GENCODE (release M30) and 18S ribosomal sequence. The trimmed reads for each cell were mapped to reference genome by using STAR. Search of SNPs in 18S ribosomal reads was done manually by using IGV[147] on merged bam alignment files.

### Mass-spectrometry

Twenty mouse embryos were collected at age E9.5 with further dissection of their brains and facial structures as two separate samples.

Total RNA was extracted from cell pellets using the Trizol reagent (1 mL per sample), followed by the addition of 200 μL chloroform and vigorous manual shaking for 15 s. After incubation for 5 min at room temperature, samples were centrifuged at $12{,}000 \times g$ for 15 min at 4 °C. The aqueous phase (~400 μL) was transferred to a new tube, mixed with 500 μL isopropanol, and incubated for 10 min at room temperature to precipitate RNA. Following centrifugation at $12{,}000 \times g$ for 30 min at 4 °C, the RNA pellet was washed with 1 mL of 75% ethanol, vortexed, and centrifuged again at $7500 \times g$ for 10 min at 4 °C. Supernatants were discarded, and RNA pellets were air-dried at room temperature for at least 1 h before resuspension in 21 μL of RNase-free water. Samples were incubated at 55 °C for 10 min and quantified using a Nanodrop spectrophotometer.

For enzymatic digestion, 20 μL of total RNA was treated with 3 μL of 100 mM ammonium acetate (pH 5.3) and 1 μL of Nuclease P1 (1 U/μL), followed by incubation at 42 °C for 2 h. Subsequently, 3 μL of 1 M ammonium acetate and 1 μL of *E. coli* alkaline phosphatase (1 U/μL) were added, and samples were incubated at 37 °C for an additional 2 h. The reaction was diluted with 60 μL of phase A buffer (5 mM ammonium acetate, pH 5.3), filtered through 0.22 μm filters (SLGVR04NL, Millipore), and the filtrates were transferred to HPLC vials for subsequent analysis.

The nucleosides were separated by reverse-phase ultra-performance liquid chromatography (Nexera LC-40 system, Shimadzu) on a C18 column (Synergi Fusion-RP; 4 μm particle size, 250 mm × 2 mm, 80 Å, Phenomenex). The nucleoside detection was performed using a Shimadzu TripleQuad NX8060 in the positive ion mode. Injections of 5 μL and 20 μL were used for MRM (Multiple Reaction Monitoring) analysis.

For the downstream analysis Skyline Software (v23.1.0.268)[148] was used. Raw data representing the total area under the curve was extracted and then the ratio of the total area of each nucleoside relative to the total area of uridine was calculated.

### In vitro over-expression of *Nhp2*, *Emg1* and *Tsr3* in NC development cell lines

**Experiment protocol.** hiPSCs STAN line purchased from WiCell (#STAN061i-164-1) were cultured on matrigel (Corning, cat. No. 354277 in DMEM/F12 (Corning, cat. no. COR10-090-CV)) coated plates

(Sarstedt, cat no. 83.3920) in mTESRplus media (STEMCELL Technologies, cat no. 100-0276). hiPSCs at passage 27(P27) were transfected with a lentiviral vector containing tTS/rtTA(ns):T2A:EGFP and blasticidin antibiotic resistance (pLV[Exp]-Bsd-EF1A>tTS/rtTA(ns):T2A:EGFP) at MOI 50 in the mTESRplus media containing 5 μg/ml Polybrene (VectorBuilder, vector ID VB220929-1208sva) following established protocol (VectorBuilder, Lentivirus for in vitro applications, V2.1). Afterwards, the resulting cell line was cultured in the media containing Blasticidin 10 ug/mL (Sigma-Aldrich Handels GmbH, cat no. SBR00022) for 10 days, to enrich for transfected cells. The resulting hiPSCs STAN tTS/rtTA-Bsd cell line was used for 3 separate rounds of transfection.

hiPSCs STAN tTS/rtTA-Bsd cell line at P30 was transfected using lentiviral vector containing the gene of interest and puromycin antibiotic resistance: vector pLV[Exp]-Puro-TRE>hEMG1 for EMG1 gene, vector pLV[Exp]-Puro-TRE>hNHP2 for NHP2 gene, and vector pLV[Exp]-Puro-TRE>hTSR3 for TSR3 genes (VectorBuilder, VectorIDs: VB240408-1194zuq, VB240408-1195ftw, VB240408-1191wdj).

Further, the resulting STAN EMG1, STAN NHP2 and STAN TSR3 hiPSCs cell lines were enriched for transfected cells by adding puromycin 1 μg/ml (Sigma-Aldrich Handels GmbH, cat no. P9620-10ML) and blasticidin 10 ug/mL (Sigma-Aldrich Handels GmbH, cat no. SBR00022) to the culture media. All hiPSCs were differentiated into neural crest cells using STEMdiff™ Neural Crest Differentiation Kit (STEMCELL Technologies, cat no. 08610). Overexpression of genes EMG1, NHP2 and TSR3 was done using doxycycline hyclate 1ug/ml (Sigma-Aldrich Handels GmbH, cat no. D5207-5G) starting at day 1 of differentiation (day 0 is a seeding day) and collected at day 6. All samples were FACS sorted for EGFP before sequencing.

**Single-cell analysis.** The samples were sequenced by Biomedical Sequencing Facility at CeMM, Vienna, Austria using 10X Single Cell 3' v3 protocol. Raw sequencing data were processed using CellRanger (v8.0.1) for demultiplexing, alignment (to GRCh38-2020-A), filtering, barcode, and UMI counting. The resulting count matrices were analyzed using Scanpy (v1.9.3)[134]. Low-quality cells and genes were filtered based on minimum and maximum gene counts (from 1000 to 5000 genes per cell) and mitochondrial transcript proportion (<10%). Doublets were predicted with Scrublet[133], and the cells with a doublet score less than 0.2 were filtered out. Datasets were concatenated together, followed by normalization, log-transformation, scaling and selection of highly variable genes. Dimensionality reduction was performed by principal component analysis, and the obtained principal components were later adjusted for integration with Harmony algorithm[149]. First 20 adjusted principal components were used for 30 nearest neighborhood graph construction, followed by UMAP visualization, and clustering with the Leiden algorithm. Cell types were assigned based on canonical marker gene expression, and differential gene expressions were tested using non-parametric methods implemented in Scanpy.

**Ultrasound guided virus injections.** Injections for lentiviral gene delivery were performed according to previously published protocols[80]. A glass capillary was pulled and ground to create an ultra-sharp needle. At E7.5 females were anesthetized with isoflurane, placed on a heated plate, and their depth of anesthesia was checked. Eye gel was applied to prevent dehydration. Pain relief, Buprenorphine 0.05–0.1 mg/kg was given via subcutaneous injection before surgery. Instruments were sterilized, the skin disinfected, and the abdomen prepared. The uterus was carefully placed into the petri dish through the elastic bottom with PBS. The embryos were viewed using a Vevo2100 ultrasound imaging system. Using a glass capillary, lentivirus (46 nL) was injected through the uterine wall into the amniotic cavity. Following the injection, the uterus was repositioned into the abdomen, and the muscle wall incisions were closed with absorbable

suture thread, while the skin was sealed with either suture thread or staples. The mice were taken off isoflurane and allowed to wake up in their cages on a heating pad under supervision. Postoperative pain relief was administered via a subcutaneous injection of Buprenorphine (0.05–0.1 mg/kg). To retrieve the embryos, the mother was humanely euthanized using a CO2 overdose and cervical dislocation, and the embryos were collected at E9.5 before being euthanized through decapitation.

**Immunohistochemistry.** Whole embryos were dissected and preserved in 4% paraformaldehyde in PBS at 4 °C for 2–4 h. After rinsing in PBS, the samples were cryoprotected in 30% sucrose in PBS at +4 °C for 24 h. They were embedded in OCT compound, frozen at −20 °C, and sliced into 14 μm serial sections. These sections were placed on SuperFrost microscopy slides and stored at −20 °C until staining.

The cryosections were brought to room temperature (air-dried for a minimum of 2 h). For antigen retrieval, the slides were immersed in 1× Target Retrieval Solution (Dako, S1699) diluted in water. The solution was heated to boiling and then cooled for a duration of 40 to 60 min. Following antigen retrieval, the sections were washed three times for 10 min each, using PBS containing 0.1% Tween-20 (PBST). To prevent antibody overflow, the sections were carefully encircled using a Super PAP Pen (Invitrogen, 008899). Subsequently, the sections were incubated overnight at room temperature in a humidified chamber, using primary antibodies diluted in PBST. After the incubation, the sections were washed three times for 10 min each in PBST. Next, the sections were incubated at room temperature for 90 min with secondary antibodies and Hoechst (1 μg/mL), both diluted in PBST. Following this incubation, the sections were washed again three times in PBST and finally mounted using Mowiol (Merck, 81381) mounting medium. The mounting medium was prepared in accordance with the instructions provided by the manufacturer.

### In vivo knockout of *Tsr3* in mouse embryos
**CRISPR/Cas9.** To generate Tsr3 knockout mice, two guide RNAs (gRNAs) targeting conserved regions of exon 2 and exon 5 of the Tsr3 gene were designed using Benchling (www.benchling.com) and CRISPRroots v1.3. The selected 20-bp target sequences were: exon 2 – 5′-CGGGCTGCACGGATGCTACA-3′ (guideRNA_1), and exon 5 – 5′-TCACAGGACCTACCGGGTGC-3′ (guideRNA_2). Corresponding CRISPR RNAs (crRNAs) were synthesized by Integrated DNA Technologies (IDT, Coralville, IA, USA), along with the Alt-R® CRISPR-Cas9 tracrRNA.

A 190 bp homology-directed repair (HDR) template was also synthesized, comprising a 95 bp left homology arm upstream of exon 2 and a 95 bp right homology arm downstream of exon 5. The exact sequence of HDR template (HDR_donor) was 5′-CTGAA-GAGGTGGGCGCTGCGCTGCGTGGTGAGTTTGGAGCAGTCGGAG-CAGCGGGTCGGGCCTGGGAGCAGGCGCTGAG-GACCCGGCCTTCTCTCTGTGAGCCCCCAGGTGCTTCTGAAACCTT-GACCCAGTTCCTCCGGCCTTGCGGGTGTGGCCTTTTGAA-CAAGGCCTGGTGGTGATTACAAAGTGT-3′. The HDR template was designed to facilitate precise deletion of the genomic region spanning exon 2 to exon 5, and direct ligation of the flanking intronic sequences. The targeted region, encompassing exons 2 to 5, represents a highly conserved segment critical for Tsr3 function.

The CRISPR/Cas9 ribonucleoprotein (RNP) complex, composed of Cas9 protein, the two crRNAs, tracrRNA, and the HDR template, was delivered in vivo using the improved Genome-editing via Oviductal Nucleic Acid Delivery (i-GONAD) method.

**i-GONAD.** All animal procedures were conducted in accordance with the Swedish Animal Welfare Act and were approved by the appropriate regulatory authorities in Gothenburg, Sweden. Genome editing was performed via the i-GONAD method, as previously described by

Gurumurthy et al.[82,83]. Briefly, the estrous cycle of CD1 female mice was assessed by visual inspection of the vaginal opening[150], and selected females were paired for mating between 16:00 and 17:00 on the day prior to the procedure. The following morning, the presence of a vaginal plug was used to confirm successful mating, and i-GONAD was performed at approximately 13:00 the same day. A genome-editing mixture was prepared containing 30 μM gRNA (equimolar crRNAs and tracrRNA), 6.1 μM Cas9 protein, 1 μg/μL HDR donor (ssODN) template, and 0.005% (w/v) Fast Green FCF (Carl Roth), diluted in Opti-MEM (Gibco). This mixture was injected into the ampullary region of the oviduct, followed by in situ electroporation to facilitate uptake of the components.

**Genotyping and Sequence Verification.** Founder animals were screened by PCR using primer pairs specific for wild-type and mutant alleles. The For_primer_mut/wt (5′-CTCGCGGATCCAGGAAAGAG-3′) and Rev_primer_mut/wt (5′-AAACCTTGACCCAGTTCCTCC-3′) primers generated an expected amplicon of 2002 bp for the wild-type allele and 204 bp for the mutant allele. To detect the deletion event, PCR was additionally carried out with For_primer_Del (5′-GCCTTCT GCATCGTAGGTGA-3′) and Rev_primer_Del (5′-AGCCCCAGAGAAGC AATCAAG-3′), yielding an expected product size of 663 bp.

Sanger sequencing confirmation of the mutant allele was performed using Sequencing primer_F (5′-TGAACAAACGCTGGAGCTGA-3′) and Sequencing primer_R (5′-GCCTCTTGTTCTGGGGTCTC-3′), producing an expected product size of 602 bp (mutant). Sequencing was performed using a single-primer bidirectional strategy, sequencing each strand separately. For the wild-type allele (2002 bp), full-length sequencing was carried out using Oxford Nanopore Technologies (ONT) Lite Clonal Amplicons. A founder carrying the desired 204 bp deletion was identified and backcrossed to CD1 for two generations prior to use in experiments.

**Whole mount immunostaining of mouse embryos.** For embryo collection, mice were anesthetized with isoflurane prior to euthanasia. Euthanasia was performed by CO₂ inhalation for 10 min, followed by cervical dislocation to ensure death. The embryos were initially fixed in 4% paraformaldehyde at +4 °C for 3 h. Following this, they were dehydrated with a methanol gradient and bleached overnight at 4 °C using Dent's Bleach (a mixture of 20% dimethyl sulfoxide in methanol, combined in a 2:1 ratio with 30% hydrogen peroxide). Subsequently, the embryos were incubated overnight in Dent's fix (20% dimethyl sulfoxide in methanol) at 4 °C and then stored at −20 °C. The embryos were incubated with the primary antibody in blocking buffer containing 5% normal donkey serum and 20% dimethyl sulfoxide for 5–6 days and then with the secondary antibody solution for 2–3 days. After incubations the embryos were washed in PBST and cleared in BABB solution (a mix of 1 part benzyl alcohol and 2 parts benzyl benzoate) for 30 min before being imaged using a confocal LSM800 microscope. The imaged embryos were analyzed using IMARIS software (version 9.5, Bitplane).

### Tumor single-cell analysis
**Materials.** To visualize the distribution of different gene signatures on tumor data, we loaded Smart-seq2 single-cell datasets of glioblastoma and medulloblastoma tumors from GEO database (GEO IDs: GSE131928 and GSE119926). The cell type annotation was inherited from the original works.

Human neuroblastoma tumors were collected following surgical resection at the Karolinska University Hospital and processed following the Nuc-Seq protocol as described previously[151]. Briefly, nuclei are obtained from deep-frozen tissue after homogenization and filtration, and further FACs sorted in 384-wells plates where cDNA synthesis was conducted with Smart-seq2. Libraries were prepared using the Tn5 transposase tagmentation (Nextera XT), and their quality was assessed

with fragment analyzer. High-quality libraries were sequenced using Illumina HiSeq 2500, and further de-multiplexed with deindexer (https://github.com/ws6/deindexer) using the Nextera index adapters and the 384-well layout.

**Single cell analysis and search for U1248 change in neuroblastoma dataset**. Search for the U1248 variant in Smart-seq2 neuroblastoma datasets was performed as described for the mouse datasets, but using the human genome.

Smart-seq2 neuroblastoma datasets were then preprocessed by the Scanpy python package separately for each patient. The low-quality cells were filtered out by removing the cells that have more than 20% of mitochondrial counts or are in the lower and upper 5% quantile for the detected genes and total counts. The genes detected in fewer than 3 cells were also removed. The filtered datasets were then merged and normalized to a target sum of 10e4 UMIs, followed by log10 transformation. Highly variable genes were identified using default parameters and used for PCA on scaled data. To obtain adjusted principal components for integration, Harmony was applied. A kNN graph was constructed using 25 nearest neighbors on the first 10 adjusted PCs, followed by UMAP embedding and Leiden clustering (resolution = 0.5). Clusters were annotated manually based on marker genes from[151].

**Statistical analysis**. To estimate the enrichment of the different signatures in cancer datasets, we compared the distribution of signature scores in each subtype (or cell type) against all other subtypes (cell types) using a one-sided Mann–Whitney U test (alternative = "greater"), recording the mean score, p value, and number of cells per subtype (cell type). Multiple testing was controlled with Benjamini–Hochberg FDR (statsmodels; $\alpha = 0.05$). Subtypes with FDR-adjusted $p < 0.05$ were considered significantly enriched for the given signature and are reported sorted by mean score.

To assess differential presence of the U1248 modification in the neuroblastoma dataset, "U1248:score" values were calculated for each cell as log-odds ratios of modified to non-modified nucleotides, with pseudocounts added to avoid zeros. Cells with coverage < 20 reads were excluded, and only cell types with ≥ 10 cells were retained. Pairwise differences in U1248:score were tested using two-sided Mann–Whitney U tests with Benjamini–Hochberg FDR correction ($\alpha = 0.05$). For significant pairs, directionality was determined using one-sided Mann–Whitney U tests. Results were summarized as ±log10(FDR-corrected p-values), with the sign indicating the higher-scoring cell type (positive if row > column; negative if row < column)

Python packages scipy (v 1.9.3) and statsmodels (0.13.5) were used for statistical analysis.

**Survival prognosis analysis in SEQC neuroblastoma dataset**
Bulk gene expression data with the clinical characteristics per each patient were obtained from GEO database (GSE49711). The dataset was divided into different subcohorts based on MYCN amplification status and/or stage and/or high-risk status. Each subcohort was analyzed separately. To compare the survival outcomes, we used the survminer R package to generate survival curves for tumors with low (lower quartile) and high (upper quartile) expression levels of a specific gene signature. Signature expression level (or signature score) was calculated using the R package GSVA (v1.38.2, method = "ssgsea"). P values were calculated using the two-sided log-rank test, comparing the Kaplan-Meier curves of the two groups. To assess the robustness of the gene signatures, bootstrap resampling was done.

**Survival prognosis analysis in various neuroblastoma datasets**
All neuroblastoma cohort data were obtained from the R2 analysis and visualization platform (https://r2.amc.nl), except for the TARGET dataset, which was downloaded directly from the GDC data portal

(https://portal.gdc.cancer.gov/projects/TARGET-NBL) and assembled into metadata and count matrices via custom-made R/shell scripts. For all datasets, the untransformed counts (whose units varied in each dataset) were used directly. Harmonisation of all variables into homogeneous units was performed programmatically via R (v4.1.3).

Signature scores were computed with the GSVA package (v1.42.0, method = "ssgsea"). A subset of samples for survival analysis was then extracted, ranked, and stratified into percentile groups.

Survival analyses were conducted with the survival (v3.2–13) and survminer (v0.4.9) packages, provided via an R Markdown workflow. Covariate effects (age, gender) were estimated using Cox proportional hazards models, and Kaplan–Meier curves were adjusted via the survminer function *ggadjustedcurves* (method = "marginal").

Cox proportional hazards models were fitted using R package survival (v3.2–13). For each dataset, models included all Stage 4 patients and, separately, Stage 4 & non-MYCN-amplified patients, to obtain hazard ratios with 95% confidence intervals. Covariates included: ribosomal control and rRNA modifications signatures (high/low), MYCN status (amplified/not amplified), age (< 18 months/ ≥ 18 months), and gender (male/female). Risk status was excluded, as it was not consistently reported across datasets. Age categories were 1) less than 18 months and 2) greater than or equal to 18 months to make sure all the cohorts were comparable to Kocak et al.[96].

ROC curves for survival data were generated for each gene signature on full cohorts and subcohorts using the *roc* function (direction = "<") from the pROC package (v1.18.0), with either INSS stage = "Stage 4" or MYCN status = "Amplified" defined as cases and all other categories as controls.

## In vitro validation in neuroblastoma cell lines
**Cell cultures**. Six established neuroblastoma cell lines were included in this study: *MYCN*-amplified SK-N-BE(2) (cat. # 95011815) and non-*MYCN*-amplified SH-SY5Y (cat. # 94030304) purchased from ECACC, *MYC*-amplified CHLA-15 (CVCL_6594) and CHLA-20 (CVCL_6602) from the COG/ALSF Childhood Cancer Repository (kindly provided by Dr. Michael D. Hogarty, Children's Hospital of Philadelphia, PA, USA), and *MYCN*-amplified LAN-1 (CVCL_1827) and non-*MYCN*-amplified GIMEN (CVCL_1232) cells (a kind gift of Prof. Lumír Krejčí, Masaryk University, Brno, Czech Republic).

Additionally, 14 patient-derived neuroblastoma cell lines were included to complement the panel with models of both high-risk and non-high-risk tumors. These cell lines were established in our laboratory as previously described[103] from tumor tissues of neuroblastoma patients of the Department of Pediatric Oncology, University Hospital Brno, with written informed consent obtained under the research projects (IGA MZCR NR/9125-4, AZV MZCR 15-34621A, and MU 0829/ 2018) approved by Masaryk University's research ethics bodies (approval no. 23/2005, 26/08/2014, and EKV-2018-075, respectively). The patients did not receive any compensation; the tumor tissues were obtained during standard diagnostics and surgical treatment. The cohort description and related clinicopathological data are provided in Supplementary Fig. 18. Briefly, a revised Children's Oncology Group (COG) risk classifier (v2)[152] was used to determine the risk groups. Risk biomarkers were assessed according to standard protocols by certified laboratories of the University Hospital Brno and Masaryk Memorial Cancer Institute. Patients were examined using standard diagnostic procedures, including CT or MRI scans, MIBG scans, and bone marrow trephine biopsies. Tumor stage was assessed using image-defined risk factors according to the International Neuroblastoma Risk Group Staging System (INRGSS)[152]. All tumors were examined histopathologically and classified as favorable or unfavorable histology according to International Neuroblastoma Pathology Classification (INPC) criteria[153]. *MYCN* status was analyzed by fluorescence in situ hybridization (FISH). Ploidy was determined using flow cytometry (DNA index) or conventional karyotype analysis. Segmental chromosomal

aberrations (SCAs) with prognostic impact (loss of 1p, 11q, 3p, and 4p, and gain of 1q, 2p, and 17q)[152] were considered and examined using comparative genomic hybridization (CGH) or array CGH[154,155]. In some cases, locus-specific FISH probes were used to additionally infer SCAs: 1p (ZytoLight SPEC 1p36/1q25 DC Probe), 11q (MetaSystems XL MLL plus), or 17q (MetaSystems XL Iso(17q)). COG protocols were used to treat patients according to their risk.

Human neonatal dermal fibroblasts NDF-2 (#CC-2509; Lonza Bioscience, Durham, NC, USA) used in some experiments were kindly provided by Dr. Tomáš Bárta (Masaryk University). All cell lines were authenticated by STR profiling (Generi Biotech, Hradec Králové, Czech Republic) and routinely tested for mycoplasma by PCR.

Cell lines were cultured in a 1:1 mixture of DMEM/Ham's F-12 medium supplemented with 10% (CHLA-15 and CHLA-20) or 20% fetal calf serum, 2mM L-glutamine, 100 IU/mL penicillin and 100 µg/mL streptomycin (all purchased from Biosera, Nuaillé, France). For all neuroblastoma cell lines, media were further supplemented with 1% of nonessential amino acids (Biosera) and in case of CHLA-15 and CHLA-20 also with 1× ITS-X (Gibco). The cells were maintained under standard conditions at 37 °C in a humidified atmosphere containing 5% $CO_2$ and subcultured 1-2 times weekly.

Availability of newly generated neuroblastoma cell lines and any additional protocol details should be requested from the Dr. Jan Skoda lab (jan.skoda@sci.muni.cz).

**Western blotting and immunodetection.** Whole-cell extracts from all neuroblastoma cell lines were collected using RIPA lysis buffer (2 mM EDTA, 1% IGEPAL® CA-408 630, 0.1% SDS, 8.7 mg/ml sodium chloride, 5 mg/ml sodium deoxycholate, 50 mM Tris-HCl) supplemented with cOmplete™ Mini Protease Inhibitor Cocktail (#11836170001, Roche, Basel, Switzerland) and PhosSTOP (#4906837001, Roche). 15 µg of total proteins were resolved on 10% polyacrylamide gels and blotted onto PVDF membranes (Bio-Rad Laboratories, Munich, Germany). The membranes were then blocked with 5% non-fat dry milk dissolved in Tris-buffered saline with 0.05% Tween-20 (Sigma) and incubated overnight with the corresponding primary antibody. The following primary antibodies were used: mouse anti-WDR74 (1:300, Santa Cruz Biotechnology, #sc-393822), rabbit anti-NHP2 (1:1000, Abcam, #ab180498), rabbit anti-CD133 (1:1000, Abcam, #ab19898), rabbit anti-HTR3A (1:1000, Abcam, #ab13897), rabbit anti-c-MYC (1:1000, CST, #5605), rabbit anti-MCL-1 (1:1000, CST, #94296), mouse anti-cyclin D1 (1:1000, Exbio, #11-535-C100), rabbit anti-GADPH (1:5000, CST, #2118) and mouse anti-GAPDH (1:5000, Santa Cruz Biotechnology, #sc-365062). The next day, the membranes were incubated for 1 h with the corresponding HRP-linked secondary antibody. The following secondary antibodies were used: goat anti-rabbit IgG (1:5000, CST, #7074) and horse anti-mouse IgG (1:5000, CST, #7076). Proteins were visualized using chemiluminescent Amersham ECL Prime detection system (Cytiva, Marlborough, MA, USA) and Azure 600 imaging system (Azure Biosystems, Dublin, CA, USA). Band densities of proteins of interest were quantified using ImageJ (Fiji) software (version 2.1.0/1.53c; NIH, Bethesda, MD, USA) and normalized to that of GAPDH, which served as a loading control.

**Limiting dilution sphere formation assay.** Cells were harvested and dissociated into single-cell suspension by Accutase (Biosera), re-suspended in a defined DMEM/F12 based serum-free medium supplemented with 1× B-27 supplement w/o vitamin A (Gibco), 20 ng/mL FGF2 (Sigma-Aldrich), and 10 ng/mL EGF (Sigma-Aldrich). Cells were then serially diluted into ultra-low attachment 96-well plates (Corning, cat. #3474) to reach final cell densities of either 200, 100, 50, 25, 5, and 1 per well or 1000, 200, 100, 50, 25, 5 per well (based on previous screening of tested cell lines) with five technical replicates of each cell

density. Every three days, cells were replenished with defined medium with fresh growth factors. After a week, the fraction of wells containing neurospheres ≥50 µm in diameter was determined for each cell density using an IncuCyte® SX1 imaging software (Sartorius, Göttingen, Germany). Frequencies of sphere-forming cells among experimental groups were calculated and compared using ELDA software[156].

**Cell proliferation.** The proliferation activity of cells was assessed by analyzing cell growth using IncuCyte® SX1 live-cell imaging system (Sartorius) and by measuring cell viability by MTT assay. Cells were seeded in 96-well plates (SPL, #30096) at a density of 2000 cells per well in 100 µl of culture medium. To evaluate the effect of protein synthesis inhibition, cycloheximide (Tocris, #0970) treatment was performed 24 h after the seeding by adding 100 µl of fresh medium with cycloheximide to reach its final concentrations from 100 pM to 100 µM per well. The proliferation activity was analyzed after 72 h of treatment. Cell growth was determined by measuring confluence in each well using an IncuCyte® SX1 imaging software (Sartorius) relative to untreated controls. Metabolic activity of the cells was further assessed using 3-[4,5-dimethylthiazol-2-yl]-2,5-diphenyltetrazolium bromide (MTT) (Sigma) at a final concentration of 455 µg/ml. After a 3-h incubation under standard conditions, the medium with MTT was replaced by 200 µL of DMSO per well to solubilize the MTT product. Subsequently, the absorbance was measured at 570 nm with a reference absorbance at 620 nm wavelength using a Sunrise Absorbance Reader (Tecan). Half maximal inhibitory concentration (IC50) was determined from non-linear regression of the data with individual tested concentrations normalized to untreated control cells. Non-linear regression with variable slope was calculated using GraphPad Prism 8.1.1. software (GraphPad Software, San Diego, CA, US). Determined parameters were then used to calculate absolute IC50 values according to the following formula: relative IC50*(((50-top)/(bottom-50))^(−1/hill slope)).

**Protein synthesis rate.** Neuroblastoma cells of each tested cell line were seeded into Petri dishes (SPL, #20060). The next day, cells were treated with 30 µM cycloheximide for 8 h to inhibit translation and deplete short turnover proteins. After 8-h cycloheximide treatment, translation was restored by replacing the media with a fresh cycloheximide-free medium containing 5 µM of the proteasome inhibitor MG-132 (Tocris, #1748) to prevent degradation of analyzed proteins and the first whole-cell extracts were collected (time-point 0 min) together with respective cycloheximide untreated controls. Subsequently, additional whole-cell extracts were collected at the following time-points: 30, 60, 120, 240 and 480 min after medium replacement. Immunodetection of proteins (as described above) with known rapid turnover, c-MYC, MCL-1, and cyclin D1, was used to evaluate the rate of protein synthesis over this time course and protein band densities were normalized to untreated control cells (0 min).

**shRNA-mediated TSR3 knockdown analysis.** The stable TSR3-knockdown and control cell pool populations were prepared by lentiviral transduction of constructs encoding corresponding shRNA under a constitutive promoter (shTSR3: an equal titer mixture of lentiviral particles with three different target-specific constructs, vector IDs: VB250526-1389zdw, VB250526-1390jym, and VB250526-1391xqd; scramble control shCTRL: vector ID VB010000-9541pqu, all from VectorBuilder), according to the manufacturer's instructions. Cells were seeded in 12-well plates (SPL, #30012) at a density of 20,000 cells per well in 1 ml of culture medium. The next day, cells were pretreated with 10 µg/ml polybrene (#sc-134220, Santa Cruz Biotechnology) prior to transduction (MOI = 5). Stable transduced cell pools were selected by two rounds of 5-day treatment with 2 µg/ml puromycin (#sc −108071, Santa Cruz Biotechnology).

RNA extraction, reverse transcription and qPCR were performed as previously described[103]. All qPCRs were performed in technical triplicates. The expression of *TSR3* was determined using three biological replicates, with the heat shock protein gene HSP90AB1 serving as the endogenous reference control. Following primers (5′ → 3′) were used: *TSR3* (aminocarboxypropyl transferase) forward - AGTTGGGC-CACTGCGAC; *TSR3* reverse - AAGGTCTGGAAAGCCTACGATG; *HSP90AB1* (Heat shock protein 90 alpha family class B member 1) forward - CGCATGAAGGAGACACAGAA; *HSP90AB1* reverse - TCCCATCAAATTCCTTGAGC.

For growth analysis of shTSR3 and shCTRL cell pools, cells were seeded in 15 technical replicates into 96-well plates (SPL, #30096) at a density of 1500 cells/well (SH-SY5Y) or 1,000 cells/well (GIMEN) in 200 μl of culture medium. The day after seeding, cell growth was continuously analyzed by measuring cell confluence by the IncuCyte SX1 live-cell imaging system (Sartorius) every 6 h over a 96-h period.

**Statistical analysis.** All experiments with neuroblastoma cell lines were replicated three times, as detailed in the figure legends. Statistical analysis was done using GraphPad Prism 8.1.1 software. For comparisons between multiple groups for one variable, the one-way ANOVA test followed by Tukey's multiple comparison test was applied. Statistical significance of the comparisons between two groups (WDR74-high vs. WDR74-low, shTSR3 vs. shCTRL) was determined by unpaired two-tailed Welch's *t*-test. $p$ values $< 0.05$ were considered statistically significant; $*p < 0.05$, $**p < 0.01$, $***p < 0.001$. Linear correlation between 2 datasets was tested by Pearson correlation coefficient ($r$). For single-gene survival analysis, R2 Genomics Analysis and Visualization Platform (http://r2.amc.nl) was utilized and WDR74 expression (cutoff mode: first vs last quartile) was used to generate Kaplan-Meier survival curves from a publicly available Kocak dataset (GSE45547) of 405 annotated bulk non-*MYCN*-amplified neuroblastoma samples.

### Reporting summary

Further information on research design is available in the Nature Portfolio Reporting Summary linked to this article.

## Data availability

All new mouse sequencing data associated with this study have been deposited in the National Center for Biotechnology Information (NCBI) Gene Expression Omnibus (GEO) under the accession numbers GSE290341 (temporal *Polr1a* knockout experiment) and GSE308372 (in vitro overexpression of *Nhp2*, *Emg1*, and *Tsr3*). Sequencing data from neuroblastoma patients are available via the controlled-access European Genome-phenome Archive (EGA) under Study ID EGAS50000001103 and Dataset ID: EGAD50000001596. Access requests should be submitted to the Data Access Committee which will respond to requests within 4-6 weeks. Access will be granted upon completion of a Data Access Agreement. The previously published mouse cranial neural crest datasets (E8.5, E9.5, and E10.5) and trunk postotic neural crest datasets (E9.5 and E10.5) analyzed in this study are available in GEO under accession numbers GSE201257 and GSE129114, respectively. The processed data including mouse cranial neural crest dataset can be browsed from: https://adameykolab.hifo.meduniwien.ac.at/cellxgene_public/. Original data underlying this manuscript can be accessed from the Stowers Original Data Repository at https://www.stowers.org/research/publications/LIBPB-2601. Source data are provided with this paper in the Supplementary Information/Source Data Files. Any additional materials will be available from the corresponding author upon request. Source data are provided with this paper.

## Code availability

The code used for single-cell analysis can be found at the GitHub link: https://github.com/ipoverennaya/ribo_modification_paper.

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

## Acknowledgements

The Genotype-Tissue Expression (GTEx) Project was supported by the Common Fund of the Office of the Director of the National Institutes of Health, and by NCI, NHGRI, NHLBI, NIDA, NIMH, and NINDS. The data used for the analyses described in this manuscript were obtained from dbGaP (phs000424.v8.p2). The work of I.P. was supported by the European Union's Horizon 2020 Research and Innovation Program under Marie Sklodowska-Curie (grant agreement No. 860635, ITN NEUcrest). I.A. was supported by ERC Synergy grant KILL-OR-DIFFERENTIATE 856529, Knut and Alice Wallenberg Foundation, Swedish Research Council, Bertil Hallsten Foundation, Paradifference Foundation, Austrian Science Fund Project Grant. S.S. acknowledge support from ERC Synergy grant KILL-OR-DIFFERENTIATE 856529. R.G. was supported by ALSF (the Crazy 8 project grant). The work of L.S., A.G., K.P., and J.S. in neuroblastoma cell lines was supported by the Ministry of Health of the Czech Republic (No. NW24-07-00017). L.M. was supported by a postdoctoral fellowship from the American Association for Anatomy, and a National Institute for Dental and Craniofacial Research Kirschstein-NRSA F32 Post-doctoral Fellowship (DE033617). P.A.T. is supported by the Stowers Institute for Medical Research. M.K., T.Z. and J.K. acknowledge CzechNanoLab Research Infrastructure supported by MEYS CR (LM2023051). A.S.C. was supported by the Swedish state under the agreement between the Swedish government and the county councils, the ALF agreement in Gothenburg (ALFGBG-1006264). The generation of TSR3 mutant mice was supported by a 3 R grant from the Swedish Research Council to A.S.C. (#2023-02394). E.R.A. and B.S. acknowledge support from the Swedish Research Council (2024-03113, 2021-03537). M.F. was supported by the Deutsche Forschungsgemeinschaft (DFG) through CRC 1399 and CRC 1588 (grant ID 413326622 and 493872418, respectively) and by the CANTAR network (grant NW21-062B), an initiative of the Ministry of Science of the State of North-Rhine-Westphalia. Mass spectrometry experiments were carried out using the facilities of the Montpellier Proteomics Platform (PPM, Bio-Campus Montpellier), a member of the national Proteomics French Infrastructure (ProFI UAR 2048) supported by the French National Research Agency (ANR-24-INBS-0015, Investments for the future F2030). We thank Dr. Olga Kharchenko for the help with illustrations, and Dr. Vladimira Valova and Prof. Marie Jarosova for their assistance in retrieving the original comparative genomic hybridization profiles. We acknowledge the Core Facility Flow Cytometry, Medical University of Vienna and Biomedical Sequencing Facility at CeMM for access to their facilities and services during the project. We thank the Stowers Laboratory Animal Services Core for their care and maintenance of the mouse colonies, Kaitlyn Petentler and the Sequencing and Discovery Genomics Core for their sequencing support, and Karin Zueckert-Gaudenz for technical assistance with HCR. Figures 5a, e, and Supplementary Fig. 7a were created with BioRender.com. Specific publication licenses are available at: https://BioRender.com/pvtj8ya (Fig. 5a), https://BioRender.com/2cryor7 (Fig. 5e), and https://BioRender.com/smp65tq (Supplementary Fig. 7a).

## Author contributions

All authors participated in discussing and drafting the manuscript. I.P. performed most of the data analysis in this manuscript. A.M. and A.E. performed the experiments on neural crest. Y.G. performed data analysis on human samples. P.K. and S.S. provided tumor samples, sequenced them and performed the provisional analysis. M.A. generated Smart-seq2 data for neuroblastoma samples. V.P. and T.V. helped with embryonic mouse experiments. L.M. and P.T. designed and performed the experiments on transgenic *Polr1c* and *Polr1a* mouse lines. S.C. carried out the computational analysis of transcriptomic data generated from transgenic *Polr1a* mouse lines. M.K., T.Z., and J.K. performed the microCT scans. T.G. and P.V.K. analyzed the GTEx cohort. R.G., O.C., and T.S. performed in vitro overexpression of *Nhp2*, *Emg1* and *Tsr3*. L.L and A.S.C. created Tsr3-deficient mice and were involved in their characterization. A.A. and D.A. performed and analyzed mass spectrometry experiments. L.F.M.G., C.B., and M.F. helped with survival analysis of neuroblastoma patients in different cohorts. E.R.A and B.S. helped with in utero nano-injection experiments. L.S., A.G., and J.S. carried out the experiments on neuroblastoma cell lines. J.St. and K.P. provided tumor samples and relevant clinical data for the patient-derived neuroblastoma cell line cohort. I.A. participated in data analysis, secured funding and supervised the study. I.P., I.A., A.S.C., and P.V.K. jointly conceived the main idea of this study. All authors read and approved the manuscript.

## Funding

## Competing interests

P.V.K. serves on the scientific advisory board of Celsius Therapeutics and Biomage. P.V.K. is an employee of Altos Labs, Inc. The rest of the authors declare no competing interests.

## Additional information

[1]Department of Neuroimmunology, Center for Brain Research, Medical University of Vienna, Vienna, Austria. [2]Department of Physiology and Pharmacology, Karolinska Institutet, Stockholm, Sweden. [3]Department of Internal Medicine and Clinical Nutrition, Institute of Medicine, Centre for Bone and Arthritis

Research at the Sahlgrenska Academy, Gothenburg University, Gothenburg, Sweden. [4]Stowers Institute for Medical Research, Kansas City, MO, USA. [5]Department of Experimental Biology, Faculty of Science, Masaryk University, Brno, Czech Republic. [6]International Clinical Research Center, St. Anne's University Hospital, Brno, Czech Republic. [7]Central European Institute of Technology, Brno University of Technology, Brno, Czech Republic. [8]Department of Biomedical Informatics, Harvard Medical School, Boston, MA, USA. [9]IRCM, Université de Montpellier, ICM, INSERM, Montpellier, France. [10]Aix-Marseille University, CNRS, UMR 7288, IBDM, Marseille, France. [11]Department of Experimental Pediatric Oncology, University Children's Hospital of Cologne, Cologne, Germany. [12]Center for Molecular Medicine Cologne, Medical Faculty, University of Cologne, Cologne, Germany. [13]Department of Molecular Biosciences, the Wenner-Gren Institute, Stockholm University, Stockholm, Sweden. [14]Department of Pediatric Oncology, University Hospital Brno and Faculty of Medicine, Masaryk University, Brno, Czech Republic. [15]Department of Cell and Molecular Biology, Karolinska Institutet, Stockholm, Sweden. [16]Department of Microbiology, NYU Grossman School of Medicine, New York, NY, USA. [17]Division of Pathology, Department of Laboratory Medicine, Karolinska Institutet, Stockholm, Sweden. [18]Department of Oncology-Pathology, Karolinska Institutet, Stockholm, Sweden. [19]Childhood Cancer Research Unit, Department of Women's and Children's Health, Karolinska Institutet, Stockholm, Sweden. [20]Altos Labs, San Diego Institute of Science, San Diego, CA, USA. [21]IRMB-PPC, INM, CHU Montpellier, INSERM, Université de Montpellier, CNRS, Montpellier, France. [22]Department of Cell Biology and Physiology, University of Kansas Medical Center, Kansas City, KS, USA. [23]Science for Life Laboratory, Institute of Medicine, University of Gothenburg, Gothenburg, Sweden. [24]Present address: Division of Hematology/Oncology, Boston Children's Hospital, Boston, MA, USA. ✉e-mail: andrei.chagin@gu.se; igor.adameyko@meduniwien.ac.at

