## [Transparent Peer Review file · Nature Communications]

Ribosomal modifications are associated with mesenchymal fate selection in the neural crest lineage

Corresponding Author: Professor Igor Adameyko

Version 0:

Reviewer comments:

Reviewer #1

(Remarks to the Author)

The Authors used the smart-seq2 single-cell RNA-seq method to study fate decisions in the cranial and trunk neural crest cells (E8.5, 9.5, and 10.5). by using String network analysis, the authors found that 'ribosome control and ribosome modification' related genes are important in non-MYCN amplified patients. The authors claim these genes are associated with the modifications of 18S rRNA at position 1248. Although they hypothesize that there are SNPs at this position possibly due to the mutations generated by reverse transcription for the modified bases, it lacks the validations. And Smart-seq2 protocol uses Mg²⁺ for the RT-PCR, which makes it difficult to generate mutations for the modified bases. 1) if the author could use Mn²⁺ for the RT step for the sorted single cells or mini-bulk cells (no need for other treatment), the mutational rate would be higher and clearer. The author can use the published paper for the modification studies (DOI: 10.1038/s41592-023-02128-y). 2) It would be more convincing if the author could disrupt the modification-related protein (reader or writer) to check the consequences.

Below are the other detailed comments:

- (1) Figure 1 no scale bar for all the expressions, how to define low and high.
- (2) Figure 1f, j, k, l does not appear as ordered in the text. And Figure 1j and figure 1l label wrong in the main text?
- (3) Figure 1k, l, why use a different method for snail 1 and other genes, why not use DOROTHEA instead of SCENIC for snail1? Again, there is no scale bar.
- (4) Figure 2a, what is the gene set used for this analysis (e.g. what are the genes of early ectomesenchyme signature)?
- (5) Although the author claims that the RNA expression does not perfectly reflect actual protein level, Could the author plot the gene expression of Ncl, Nhp2, Grwd1, Wdr75, Ddx31, and Ruvbl1 in the clustered cells?
- (6) Could the author disrupt the ribosome modification by changing polr1a and NPH2 and doing single-cell RNA-seq to check whether it will affect the NCCs?
- (7) Figure 3d, what is the exact gene set of ribosomal control and rRNA modifications? How many of them? How to define the 'high or low expression of the transcriptional program' in Figure 3d. What is the time meaning of the X-axis?
- (8) Does MYCN is associated with rRNA modification?
- (9) Figure 4a, c shows WDR74 HIGH, WDR74 LOW, but 4d only shows 3 high and 3 low, what about others for the sphere-forming cells?
- (10) Figure 5a, what does each row represent? Figure 5c, is this variant correlated with gene expressions such as WDR74, and NPH2? I am curious that there should be more SNPs along 18S rRNA, why only position 1248 shows the mutations, what about other positions, do they have any cutoff of the numbers of SNPs detected at each base? Figure 5c, the Color bar is missing, 5d, how to define low and high? Again, Smart-seq 2 uses mg²⁺ buffer for the Reverse transcription.
- (11) Fig.5d does not show as ordered in the main text.

Reviewer #2

(Remarks to the Author)

In this paper, Poverennaya et al investigate how ribosome composition and ribosomal assembly programs can affect neural crest cell fate bias and implications in neural crest-derived cancers. Here, they investigate cranial vs trunk neural crest fate decisions and uncover a role for a nucleotide modification at position 1248 in 18S rRNA for establishing ectomesenchymal identity. This modification is also present in neural crest-derived tumors wherein a mesenchymal cell identity has been

found.

1. Are the neuroglial cells referenced in the single cell data those that will become sympathetic neurons? In regards to the connection between the cranial neural crest data and the neural crest-derived cancer aspects of the paper, do the authors think that cancers with a poorer prognosis have a disrupted fate selection during development?
2. The authors perform IHC for NHP2 in E9.5 mouse heads and see co-expression of SOX10 and NHP2 (Fig2c,d). Concluding its importance. This staining is not convincing and the Hoechst is quite patchy. It would be helpful to see more sections from those analyzed.
3. A conditional KO of polymerase 1 subunit alpha, Polr1a, in mouse embryos at E8.5 results in many NC defects (face, neural, and cardiac), so the authors repeat this experiment at the time of the bifurcation in their sc data and find craniofacial abnormalities (Fig2E). This mutant analysis would benefit from more characterization of this mutant phenotype including histological sections and staining and skeletal preps to show craniofacial defects. Can the authors comment on why the earlier KO resulted in a vast number of defects as well?
4. The authors describe a ribosomal assembly signature in different malignant cell populations. This is an interesting point, but did it is unclear whether the authors compare these populations to other tumors that don't have a mesenchymal identity or other adult tissue types? Is this a feature of all cancers or just neural crest-derived ones?
5. When analyzing the different malignant tumor datasets, the authors note that they remove healthy cell types from their analysis of previously published datasets, however, it'd be interesting/necessary to see if there are the same signatures in adult healthy tissues.
6. Is there any evidence to suggest WDR74 is in abundance in other cell types? or in sympathetic neurons or chromaffin cells?

Reviewer #3

(Remarks to the Author)

The study "rRNA-modifications connect mesenchymal fate selection in neural crest and prognosis in neuroblastoma" by Poverennaya I et al addresses the role of 18s rRNA modifications in fate selection towards skeletogenic ectomesenchyme in embryonic cranial and trunk neural crest and association of ribosomal expression signatures with mesenchymal features and prognosis in tumors derived from the neural crest. Despite presenting interesting observations, the manuscript appears immature and inconsistent with at times sloppy presentation of data in the text and figures jumping between topics, lack of experimental validations and overinterpretation of data presented in a lengthy and unfocussed discussion.

Major concerns:

1. Most of the paper relies on inferences from single cell data from mouse created in previous studies and public datasets. More detailed mechanistic validations are missing. E.g. modification in U1248 of 18s rRNA has not been experimentally confirmed, protein interaction is just inferred, but no experimental confirmation either.
2. I have a conceptual problem understanding the claim made regarding figure 4 that "aggressive neuroblastoma cell phenotypes associate with high levels of ribosomal assembly proteins", as all neuroblastoma cell lines are derived from aggressive, metastatic stages (no cell lines from benign, localized or regressing neuroblastomas exist). How do the neuroblastoma cell lines shown in figure 4 differ in terms of MYCN amplification or MYCN/MYC expression? Figure 4f does not show a pronounced difference in the sensitivity of WDR74 low vs high cell lines to cycloheximide. This needs validation in additional cell lines. Of note, translation inhibitors failed in pre-clinical and clinical studies in neuroblastoma.
3. Figures are prepared somewhat sloppy, e.g. in figure 1h it is not clear what kind of data is shown in the second row (scatter blots), x and y-axis labels are missing. Figure 1h also lacks a proper description in the figure legend and text. It is e.g. not clear what "real" and "permuted" is referring to. Figure 1l is missing the color code labels.
4. In figure 3a,b,c the data presented are not convincing and need additional statistical analysis to investigate potential co-expression of ribosomal and mesenchymal signatures, as well as MYC expression and activity.
5. Data presented in Figure 7 are confusing and not convincing, as the datasets presented seem very heterogeneous in their composition and mostly contain very few cells. Statistical analysis is missing. How are "unfavorable outcome tumor cells" defined? Do SCPs refer to tumor cells (then better SCP-like)? Why do the authors refer to T-cells, macrophages etc and then in other to immune cells? This should be harmonized.
6. Is the variation in 18s rRNA restricted to U1248 as claimed (see figure 6)? This should be verified by investigating other positions as control. It is odd and seems unlikely that a SNP that is ubiquitously present in the investigated normal tissues across all investigated individuals should be associated with modulating fate decision in NCC differentiation and cancer.
7. The authors state that "Not surprisingly, the investigated signatures were not able to further differentiate survival in MYCN-amp tumors, which are notoriously associated with the poorest outcomes in neuroblastoma." This is only true if considering localized and metastatic forms of neuroblastoma, however in the aggressive forms, MYCN amplification is not a poor prognostic factor. For interpretation of the survival analysis (EFS or OS?) it would be important to describe the patient cohort. If all stages were included, MYCN amplified are only high-risk and mostly metastatic, whereas MYCN non-amplified will contain all stages, which will introduce a strong bias in the analysis. In general the survival analysis is not convincing and needs validation in an independent and larger cohort to support the statements made.
8. The abstract needs revising, as the topic is jumping from cranial to trunk neural crest and from skeletogenic ectomesenchyme to neuroblastoma with no clear rationale. It is also not clear why the authors focused on neuroblastoma from figure 4 onwards.

9. The discussion is lengthy, without focus and full of overinterpretation of the data presented.
10. It is not clear why the authors introduce the role of 18s rRNA U1248 variants again from lines 327 onwards, despite already reporting on this in previous paragraphs?

Minor

1. The authors are advised to review the introduction with regards to supporting their statements with adequate literature citations (see e.g. lines 104 – 108).
2. I find the title inadequate, as the data presented on survival are weak. The authors should rather focus on their biological findings and choose an appropriate title.
3. Sphere formation is not an assay for stem-cell properties in cancer cells.
4. Why are the authors reporting a preliminary analysis in lines 139 – 141?
5. Figure 2b should be moved to the supplemental data, as it does not show original data.

Version 1:

Reviewer comments:

Reviewer #1

(Remarks to the Author)

The reviewers have address most of my concerns. Despite much of their efforts, most of the approaches towards identifying the m¹acp³Ψ modification on U1248 are indirect. It would be great if they could show that the profile of m¹acp³Ψ modification on a synthetic RNA is the same as that on rRNA using direct RNA sequencing, although this could be too tedious.

Reviewer #2

(Remarks to the Author)

The authors addressed all of my concerns sufficiently for publication.

Reviewer #3

(Remarks to the Author)

I have carefully read the revised manuscript as well as the authors' responses to my and the other reviewer's comments. I commend the authors for their thorough approach in revising the manuscript. A substantial set of new data and additional analyses have been included, which mostly support the initial claims.

There remain two considerations / clarifications:

1. In response to comment 2, the authors provide additional experiments using own cell lines generated from low-risk neuroblastoma (according to their claim). The authors state: "Notably, the WDR74-high group predominantly comprised cell lines derived from high-risk tumors (5/6 cases), including both MYCN/MYC-amplified and -non-amplified cases, whereas models from low-risk cases were exclusively found in the WDR74-low group." It seems surprising that the authors managed to derive cell lines from low-risk neuroblastoma tumors, as such attempts have been unsuccessful for decades in efforts worldwide. It will be important to provide histopathological and genetic evidence that these cell lines were indeed derived from low-risk tumors.
2. The manuscript text, especially the results and discussion section, are still lengthy and would benefit from a more concise format.

Version 2:

Reviewer comments:

Reviewer #3

(Remarks to the Author)

I appreciate that the authors have clarified the clinical characteristics of the patients from whom the cell lines were derived and analyzed in this manuscript. They have fully addressed my comments and concerns.

Reviewer #1 (Remarks to the Author):

The Authors used the smart-seq2 single-cell RNA-seq method to study fate decisions in the cranial and trunk neural crest cells (E8.5, 9.5, and 10.5). by using String network analysis, the authors found that 'ribosome control and ribosome modification' related genes are important in non-MYCN amplified patients. The authors claim these genes are associated with the modifications of 18S rRNA at position 1248. Although they hypothesize that there are SNPs at this position possibly due to the mutations generated by reverse transcription for the modified bases, it lacks the validations. And Smart-seq2 protocol uses Mg²⁺ for the RT-PCR, which makes it difficult to generate mutations for the modified bases. 1) if the author could use Mn²⁺ for the RT step for the sorted single cells or mini-bulk cells (no need for other treatment), the mutational rate would be higher and clearer. The author can use the published paper for the modification studies (DOI: 10.1038/s41592-023-02128-y).

- We thank the reviewer for being cautious about this specific moment.

1. To respond to reviewer's comment, we performed the direct validation to make sure this is a m¹acp³ψ post-transcriptional modification in U1248 position (and not rDNA polymorphism). For this, we performed a mass-spectrometry analysis of isolated 18S rRNA from mouse embryonic samples, also using comparisons with chemically pure standards for modification of U nucleotide. The only established position with the m¹acp³ψ modification is 1248 of 18S rRNA according to the literature (Holvec et al., 2024; Maden et al., 1975; Saponara & Enger, 1974), and we found this modification in 18S rRNA isolated from embryonic heads at the investigated stages. Please find new data in revised Figure 5e-f and Extended Data Fig. 5. We added the following text to the Results section:

„Uridines may harbor various modifications, including pseudouridine (ψ), N1-methyl pseudouridine (m¹ψ), or N1-methyl-N3-(3-amino-3-carboxypropyl) pseudouridine (m¹acp³ψ) (Fig. 4g). Pseudouridine can be excluded as a cause of sequencing error as it is known to be widely present in 18S rRNA (Holvec et al., 2024; Maden and Wakeman, 1988), and yet no other repetitive pseudouridine-associated sequencing error was detected in our single-cell data. At the same time, an extensive m¹acp³ψ modification was reported for U1248 in yeast and mammalian cells (Holvec et al., 2024; Maden et al., 1975; Saponara and Enger, 1974).

To confirm the presence of the m¹acp³ψ modification in murine embryonic 18S rRNA, we performed mass spectrometry (Fig. 5e). Direct analysis at earlier stages, such as cranial neural crest EMT (E8.5) and subsequent migration and fate selection (E9-9.5), was not feasible due to the limited number of available cells and the sensitivity constraints of mass spectrometry. We therefore chose a slightly later developmental window (E9.5-E10), when cell numbers are substantially higher and allow for reliable detection. To contrast the predominant tissue types at this stage, we dissected the heads of twenty E9.5-E10 mouse embryos from multiple independent litters and surgically separated brain regions from facial structures, collecting them into two distinct sample sets for quantitative analysis. After isolating 18S RNA from pooled developing brain areas and facial structures, we subjected it to mass spectrometry to investigate the relative abundance of 30 different nucleotide modifications, including pseudouridines and m¹acp³ψ. We quantified the relative abundance of each modification by a ratio of modified/unmodified nucleoside peak area. Mass spectrometry results confirmed the differential presence of m¹acp³ψ in 18S RNA from the brain area and facial samples (tissues from multiple embryos were pooled together for this analysis) (Fig 5f) at developmental stages after the neuro-mesenchymal bifurcation in CNCCs. Also, as expected, the levels of m¹acp³ψ were much lower compared to other modifications such as pseudouridines (ψ), 2'-O-methyladenosine (Am), 3-methyluridine (m³U) or even 3-(3-amino-3-carboxypropyl)uridine (acp³U) (Extended Data Fig. 5, Suppl. table 2).“

2. Also, we established a new collaboration with the laboratory of Professor Matthias Fischer to access raw reads from sequenced neuroblastoma patient cohort (SEQC). The RNA content was defined for individual tumors using standard bulk sequencing approach. In this new analysis (see revised Figure 7i), we detected different proportions of misread nucleotide in position U1248 in 18S rRNA. Here is the modified text of Results outlining these findings:

„Next we explored the impact of the U1248 modification on neuroblastoma patient survival. Although the expression of genes involved in introducing this modification correlated with poor outcomes (Fig. 7h, Extended Data Fig. 12), this does not directly confirm the impact of the modification itself. To resolve this issue, we estimated the frequency of modification using raw sequencing data and compared it with survival data of high risk neuroblastoma tumors in the SEQC cohort (high risk, non-MYCN-amplified). Notably, the tumors with medium levels of the modification had the best survival outcomes compared to those with either low or high levels of the U1248 modification (Fig. 7i). This suggests that the ratios of $m^1acp^3\psi$ modification in tumors might reflect different programs in tumor cells and potentially can be linked to specific aspects of tumor cell behavior. This requires future studies and independent validations across a spectrum of tumor models.“

3. Next, to make sure this is a nucleotide post-transcriptional heavy modification and not rDNA variation, we analyzed the GTEx cohort (Lonsdale et al., 2013) with deeply sequenced genomes and transcriptomes of numerous individuals from the variety of tissues and organs. In that case, according to reviewer’s recommendation, the sequencing protocol was designed for bulk RNA sequencing. Thus, the RT buffers and other components used to sequence GTEx cohort differ from the single cell Smart-seq2 approach in our study (Lonsdale et al., 2013; Picelli et al., 2014). Overall, GTEx analysis analysis clearly showed that while the genome of multiple individuals is clean for any encoded variant in this position, the transcriptomics data show the well-defined proportion of misread nucleotide only in this U1248 position (sometimes up to 80%). These proportions differ for specific tissues and organs. Here what we rewrote about this part:

„To confirm the reason behind the sequencing errors in the evolutionary conserved U1248 position of 18S rRNA, and to eliminate genomic variants of 18S rRNA, we examined human genomic and transcriptomic data obtained from the GTEx project (Lonsdale et al., 2013) (Fig. 5a). Our analysis of genomic data from 10 different individuals showed no nucleotide substitutions in 18S rDNA (Fig. 5b). However, at the transcriptomic level, the distribution of nucleotides at position 1248 of 18S rRNA varies from individual to individual (see Fig. 5c) and even within different tissues of the same individual (Fig. 5d). These data indicate that variations at position 1248 in 18S rRNA are introduced at the mRNA level.“

Of note, this was consistent with mouse reference genome, showing no variation in this U1248 position.

3. Also, recent independent studies have also reported and characterized the same modification at position 1248 in 18S rRNA in tumor cells (Babaian et al., 2020) and in other tissues (Milenkovic et al., 2025). Babaian et al. analyzed cancer bulk RNA-seq data from multiple cohorts as well as their own generated bulk RNA-seq, where reverse transcription was performed using the Maxima H Minus First Strand cDNA Synthesis Kit (K1652, Thermo Fisher), and sequencing was done on a HiSeq 2500 (Illumina). Milenkovic et al. employed Nanopore sequencing, with reverse transcription carried out using Maxima H Minus Reverse Transcriptase (Life Technologies, EP0751). In both cases, final concentration of $MgCl_2$ per reaction was 3-3.5 mM. As the reviewer noted, Smart-seq2 typically employs higher Mg^{2+} concentrations (6 mM) (Picelli et al., 2014). Importantly, however, the presence of “misread” nucleotides at position 1248 was observed across both protocols, regardless of Mg^{2+}

concentration. This supports the idea that the m¹acp³Ψ modification itself perturbs standard Watson–Crick base pairing during commonly used reverse transcription reactions (as also described in (Motorin et al., 2007)), thereby enabling its detection in transcriptomic data. That said, we acknowledge that while this allows us to capture this specific modification, other modifications may not be detectable under these conditions.

4. Other modifications of U and A in 18S rRNA are not manifested as sequencing errors because they are light as compared to N1-methyl-N3-(3-amino-3-carboxypropyl) pseudouridine (also abbreviated as m¹acp³Ψ) in unique position 1248. At the same time these modifications also exist in our mass spec data on 18S rRNA extracted from embryos – please see new added data in revised Figure 5d and Supplementary Figure 5.

5. This exact rRNA post-transcriptional modification in position U1248 is established by Emg1 and Nhp2 – two out of three „writing“ genes identified in our early mesenchymal signature resulting from scFates analysis of cranial neural crest. Here is the corresponding text in Results:

„The canonical uridine at position 1248 of 18S rRNA is known to be hypermodified via a three-step reaction catalyzed by the SNORA13 complex (containing the NHP2 protein), EMG1, and TSR3 proteins (Fig. 4g) (Brand et al., 1978; Cheng et al., 2024). Notably, Nhp2 and Emg1 appeared as a part of the ribosomal control and rRNA modification signature (see Fig. 2a, red color-code; Suppl. Table 1) associated with the transition of migratory neural crest cells into ectomesenchymal cell types.“

6. We are sorry that our initial narrative in the first submission was misleading: we wanted to show the historical transition in the project by first presenting the idea that it might be a DNA polymorphism and later discarding it because of all evidence pointing towards the heavy post-transcriptional rRNA modification resulting in a sequencing error in one specific U1248 position.

2) It would be more convincing if the author could disrupt the modification-related protein (reader or writer) to check the consequences.

1. We followed this advice of the reviewer and performed such experiments *in vivo* and *ex vivo*. Although it was hard to obtain insights from the gene knockout models in developing embryos (very early lethality was reported for Emg1 and Nhp2 knockouts already: (Skarnes et al., 2011; Wu et al., 2010), we generated a gene knockout for the most specific final writer of m¹acp³Ψ in U1248 - *Tsr3*. The analysis of the KO embryos showed the significant failure of the neural crest-derived ectomesenchyme to form the embryonic face, which supported our computational predictions. These new results can be found in Figure 6 and Extended Data Fig. 9.

2. In the following line of experiments, we opted for a *Tsr3* downregulation experiment in adrenergic neuroblastic neuroblastoma cell line and mesenchymal neuroblastoma cell line. The complete KO of *Tsr3* in these cell line caused lethality, although 4-folds downregulation was tolerated. Surprisingly, only mesenchymal neuroblastoma cell line showed significant and major reduction of proliferation when *Tsr3* was downregulated. In contrast to this, the neuroblastic neuroblastoma cell line showed to response. Please see new panels in Figure 8h. In sum, *Tsr3* and m¹acp³Ψ modification appear to be important in mesenchymal states vs neuronal or neuroblastic (in process of neuronal differentiation).

3. This might inform us about the mechanisms of neuroblastoma metastasis, and for this reason we analyzed neuroblastoma patient survival curves correlated with % of m¹acp³Ψ modification identified in raw bulk sequencing reads. The result showed the correlation with negative disease outcome when this % of m¹acp³Ψ significantly deviated from the average values (see new analysis in Figure 7i). This

result corresponded to the poor survival prognosis in patients with enriched signals for Emg1, Nhp2 and Tsr3 – Figure 7h.

4. Also, we found another productive approach to perturbations of this modification: we opted for mosaic clonal overexpression by using an innovative way of delivering the lentivirus with „writing proteins“ into the developing neuroectoderm and neural crest with high frequency ultrasound-guided microinjections of lentiviral particles into E7.5-E8 developing embryos *in utero* (just before cranial neural crest migrates out). The embryos after the infection with lentivirus carrying the overexpression constructs for Tsr3 and Nhp2 demonstrated the reduced developmental tempo and the loss of cells in neural parts and other parts. Still, in agreement with our prediction, we found more labelled cells with overexpression of the writers among the mesenchymal cell types. Please see new Extended Data Fig. 8.

5. Finally, the overexpression of Nhp2 and Tsr3 *in vitro* in human cultivated neural crest cells differentiated from iPS cells showed different ratio of progeny with the neural tube vs mature neural crest identity, supporting the role of these proteins in pro-EMT and migratory identity. Please see new Extended Data Fig. 7.

Below are the other detailed comments:

(1) Figure 1 no scale bar for all the expressions, how to define low and high.

We thank the reviewer for this comment, and we added the scale bars.

(2) Figure 1f, j, k, l does not appear as ordered in the text. And Figure 1j and figure 1l label wrong in the main text?

We apologize for this error, this is corrected now. All labelled panels were checked and shall convey the proper information in the correct sequence.

(3) Figure 1k, l, why use a different method for snail 1 and other genes, why not use DOROTHEA instead of SCENIC for snail1? Again, there is no scale bar.

We added the scale bar. The problem was that SCENIC did not have data accumulated for inferring Snail1 regulon. Dorothea was better in this case, as it could work with Snai1. However, Dorothea did not have enough information incorporated to infer some other important regulons. Therefore, we had to combine these two methods. In the revised version of the paper, the information about key regulons is moved to the supplementary part (Extended Data Fig. 2 in the revised manuscript).

(4) Figure 2a, what is the gene set used for this analysis (e.g. what are the genes of early ectomesenchyme signature)?

We now clarified this question in supplementary data table 1 of the revised manuscript. Also, the revised Figure 2a displays the proteins/genes that were identified as associated with mesenchymal biasing in the cranial neural crest prior to bifurcation towards neuro-glial fates and ectomesenchyme of the face. We clarified this in the figure legend:

“Protein-protein interaction network achieved by STRING for a mesenchymal early signature that was predicted by scFates as a set of genes associated with mesenchymal bias in CNCCs. The color code of the nodes reflects their association to the process of the epithelial-mesenchymal transition (green for the direct participants and light-green for any associated ones) or to ribosome-related processes (red for ribosomal control and rRNA modifications signature, light-red for any others).”

The advantage of this mesenchyme-biasing gene set shown in Figure 2a relates to the fact that it was identified through an unbiased bifurcation analysis using scFates tool (Faure et al., 2023). The goal was to find biasing genes associated with specific cell fates prior to fate commitment (before bifurcation), especially those associated with future choice towards ectomesenchyme. The large subset of this „mesenchyme early“ biasing signature contains „ribosomal control and rRNA modifications“ gene set, which plays a major role in our manuscript.

(5) Although the author claims that the RNA expression does not perfectly reflect actual protein level, Could the author plot the gene expression of Ncl, Nhp2, Grwd1, Wdr75, Ddx31, and Ruvb1 in the clustered cells?

Of course, this is our pleasure. All these genes are now plotted as UMAPS and placed into Extended Data Figure 3.

(6) Could the author disrupt the ribosome modification by changing polr1a and NPH2 and doing single-cell RNA-seq to check whether it will affect the NCCs?

Yes, we followed the advice and performed this experiment with disruption of *Polr1a* and single cell transcriptomics sequencing (please see new Figure 3 and Extended Data Fig. 4). We could not do it with *Nhp2* because of the very early embryonic lethality of *Nhp2* KO embryos.

The results of *Polr1a* disruption in the diversifying neural crest cells (assuming mesenchymal and neuroglial fates) confirmed the compositional disbalance of resulting mesenchymal vs neuroglial populations. This compositional disbalance, which is predicted by our initial single cell data (the association of rRNA biogenesis with mesenchymal fate) was rigorously tested because we sequenced several control and mutant embryos independently – and everywhere got a consistent result: in the mutant cells, the proportion of neural crest-derived mesenchymal cells was lower as compared to control embryos. We outlined these findings in the updated text of the Results:

„Furthermore, the compositional analysis of the single cell transcriptomics data from E12.5 control and temporal Polr1a knockout mutant mice (based on R26-CreERT2), injected with tamoxifen at E9.0-E9.5, revealed a significant reduction of several ectomesenchymal clusters, without neuro-glial progeny being affected (Fig. 3a-b, Extended Data Fig. 4a-b). To validate this observation, we performed HCR in situ hybridization for Sox9, a master regulator of chondrogenesis; Runx2, a master regulator of osteoblast differentiation; and Sox10, a marker of the developing peripheral nervous system (Fig. 3c,d, left panel). We also carried out immunostaining for SOX9 and TUJ1 (TUBB3A; neuron-specific class III β -tubulin) to corroborate the changes at a protein level (Fig. 3d, right panel). As expected, mutant embryos exhibited reduced SOX9 signal in the mandible, consistent with diminished cartilage development. This was evident most notably for Meckel’s cartilage, which manifested as two discrete non-contiguous proximal and distal elements (Fig 3c,d). In contrast, the branches of the neural crest-derived trigeminal ganglion were comparatively less affected, and the mandibular nerve now traversed the gap between the non-contiguous Meckel’s cartilage elements (Fig. 3c,d). Although craniofacial malformations are evident in Polr1a mutants in the bright field images (Extended Data Fig. 4c-d); we

opted for higher resolution microCT scanning of PTA-contrasted embryos (Kaucka et al., 2017) and rendered the 3D volumes of trigeminal ganglion and ectomesenchymal chondrogenic condensations (Fig. 3e, Extended Data Fig. 4e). This approach revealed that whereas $Polr1a^{flx/flx};R26-Cre-ERT2$ embryos exhibited reduced facial mesenchymal condensations, the trigeminal ganglia were not significantly affected, indicating that facial skeletogenesis was specifically disrupted as a consequence of diminished rRNA transcription and ribosome biogenesis (Fig. 3e, Extended Data Fig. 4e).

Similarly, temporal knockout of another RNA Polymerase I subunit, $Polr1c$, at E9.0-E9.5 also resulted in critical craniofacial abnormalities at E12.5, with reduced neural crest derivatives and complete absence of skeletal condensations in the face (Fig. 3f, Extended Data Fig. 4e). In contrast, the trigeminal ganglion was persisting in expected position. Together, these observations support a mechanism in which the dynamics of rRNA synthesis at the time of fate bifurcation influences later ectomesenchymal development in CNCCs.“

(7) Figure 3d, what is the exact gene set of ribosomal control and rRNA modifications? How many of them? How to define the ‘high or low expression of the transcriptional program’ in Figure 3d. What is the time meaning of the X-axis?

We responded to this comment by providing the exact gene set of ribosomal control and rRNA modifications signature in new Supplementary Table 1 (12 genes) and also displayed it in Figure 2a being highlighted by red color. For the survival analysis shown in Figure 3d (now updated and moved to Figure 6a in the revised version), we first identified the human orthologues of these 12 genes (also provided in Supplementary Table 1, relevant sheet) before proceeding to further analysis.

To define “high” and “low” expression of this transcriptional program, we calculated the average expression of the signature for each tumor. Tumors at the top and bottom quartiles (i.e., top 25% and bottom 25%) were categorized as “high” and “low” expression groups, respectively. We improved the figure legend to reflect this approach. Regarding the X-axis in the Kaplan–Meier survival plot: in the previous version, time was measured in days; we have now updated it to years for clarity and consistency. We thank the reviewer for asking these clarifying questions.

(8) Does MYCN is associated with rRNA modification?

To address this question we first plotted Myc and Mycn expression along the mouse cranial NCC developmental trajectory. Importantly, in contrast to Myc, Mycn keeps being expressed in neuroglial cells (see the figures below), and for this reason it is not identified as the pro-mesenchymal gene (mesenchymal early gene). Thus, modification in U1248 of 18S rRNA correlates only partially with MYCN in neural crest populations before the bifurcation into neuroglial and mesenchymal fates. The correlation of MYC (not MYCN) is higher at the level of cell populations, which corresponds to the early mesenchyme-biasing signature in Figure 2a. This is a minor aspect, and we do not discuss it in the manuscript for the lack of space.

However, MYCN is very important factor influencing prognosis in neural crest-derived tumors such as neuroblastoma. Therefore, in response to this comment, we reanalyzed the data of neuroblastoma patient survival from several cohorts in relation to their MYCN status and ribosomal assembly and rRNA modification signature (see new results in Figure 7 and Supplementary Figures 13-16). Here is how we now describe these results in the revised manuscript text:

„Neuroblastoma is a heterogeneous tumour, and as such, many known covariates influence survival (Gomez et al., 2022), including MYCN amplification status (i.e., the acquisition of several copies of this gene), tumour stage according to the International Neuroblastoma Staging System (INSS), and risk (usually integrating histopathology). To isolate the effect of the mesenchymal signatures on survival from that of established covariates, we first performed survival analysis on non-MYCN amplified subcohorts from later stages 3-4, or 4 alone (metastatic tumors). Initial survival analysis of the SEQC dataset (Fig. 7a) revealed that high levels of the mesenchymal early signature and particularly its subsignature – the ribosomal control and rRNA modifications signature – were significantly associated

with poorer prognosis in the subcohort. Deeper inspection of these signatures across other subdivisions of the SEQC dataset (Extended Data Fig. 11) revealed that the ribosomal control and rRNA modifications signature consistently predicted poorer prognosis with the exception of the MYCN-amplified subcohorts in this dataset. Ten out of twelve genes of the signature, on their own, were predictive of poorer outcomes in non-MYCN-amplified stage 3-4 patients (Extended data Fig. 12).

We also observed that high levels of ribosomal control and rRNA modifications signature were associated with progressively later stages and with MYCN amplification (Fig. 7b-c), making it highly predictive of MYCN status, stage, and even other covariates (Fig. 7b-d, ROC AUC >0.94 for MYCN status in SEQC database, Suppl. Table 6). Furthermore, this signature was anticorrelated with a previously published signature, Halbritter C13, which was identified in an in vitro neuroblastoma model and was associated with favorable prognosis (Saldana-Guerrero et al., 2024) (Fig. 7e-g).

We then extended the analysis to five additional neuroblastoma patient cohorts (Extended Data Fig. 13). The ribosomal control and rRNA modifications signature again demonstrated strong predictive power for MYCN status across datasets (Extended Data Fig 13a-b) and for stage 4 (except for the Westermann-ALT dataset). The signature also showed general opposition to the Halbritter C13 signature (Extended Data Fig. 13c). Moreover, it exhibited a consistent, statistically significant association with poor prognosis in non-MYCN-amplified stage 4 patients from these cohorts (Extended Data Fig. 13d). As in the SEQC dataset, the mesenchymal signatures showed no discriminatory power for survival within MYCN-amplified patients. This result suggests that small differences in the already high levels of the signature – particularly in MYCN-amplified patients, where signature expression is highest – may not contribute further to the strong effect of MYCN on negative prognosis. To extend these conclusions quantitatively, we estimated the proportional effects of different variables by applying a Cox proportional hazards model to stage 4 patient subcohorts. This analysis revealed consistently higher hazard associated with a high ribosomal control and rRNA modifications signature in thirteen out of fourteen subcohorts, whereas other covariates were less consistent (Extended Data Fig. 15). Together, these findings suggest that the ribosomal control and rRNA modifications signature inform about unifying features of neuroblastoma biology concomitant with disease progression.

In contrast to the results of the ribosomal control and rRNA modifications signature, the late mesenchymal signature (Fig. 7a) and the signature containing transcripts encoding ribosomal structural proteins (see Suppl. Table 1) showed substantially less correlation with a poor survival rate of neuroblastoma patients and did not discriminate MYCN status as well (Fig. 7a, Extended Data Fig. 16a, Suppl. Table 6). This suggests the absolute numbers of ribosomes might be not as essential as the modification of the ribosomes for cancer prognosis and for the potential presence of the most malignant cell subpopulations within individual tumors. Furthermore, the ribosomal control and rRNA modifications signature is not strongly correlated to the S phase signature (Extended Data Fig. 16b), which seems to be generally associated with poor outcomes (Cai et al., 2025; Liu et al., 2022). Interestingly, we also found that the ribosomal control and rRNA modifications signature predicts MYCN status more accurately than the cell cycle signature (Extended Figure 16b).“

(9) Figure 4a, c shows WDR74 HIGH, WDR74 LOW, but 4d only shows 3 high and 3 low, what about others for the sphere-forming cells?

We thank the reviewer for asking for these additional experiments, and we performed sphere-formation assay with additional cell lines and now provide updated Figure 8d with results from comparison of all six WDR74-high vs seven WDR74-low cell lines, significantly strengthening our conclusions. These analyses validated the statistically significant association of WDR74-high phenotype with aggressive cancer stem-like traits.

(10) Figure 5a, what does each row represent?

We apologize if this was unclear. We have updated the corresponding figure legend (now Figure 4a) for clarity:

„a, Schematic representation of rDNA organization on mouse chromosomes with highlighted 45S rDNA loci. The 45S precursor transcript encodes the 18S, 5.8S, and 28S rRNAs. A zoomed-in view of the 18S rDNA gene is shown, with aligned Smart-seq2 reads visualized in a genome browser-style layout (similar to IGV). Each violet rectangle represents an individual read, with mismatches - corresponding to putative sequence variations or post-transcriptional modifications - highlighted in red.“

Figure 5c, is this variant correlated with gene expressions such as WDR74, and NPH2?

- Yes, it did correlate with Wdr74 and Nhp2 on premigratory and migratory neural crest prior to mesenchymal bifurcation. In this revision, we have plotted the expression of Wdr74 and Nhp2 (Extended Data Figure 3) to illustrate their enrichment in pre-migratory and migratory NC clusters as well as ectomesenchymal cells. As this modification in U1248 is enriched in pre-migratory and migratory CNCC clusters but not in the ectomesenchymal cluster, we deem this correlation present only before the fate bifurcation, which is in agreement with overall results presented here.

I am curious that there should be more SNPs along 18S rRNA, why only position 1248 shows the mutations, what about other positions, do they have any cutoff of the numbers of SNPs detected at each base?

To respond to this comment, we carefully re-analyzed Smart-seq2 reads and other sequencing data presented in this study. We came to the conclusion that we reliably see only misreading in position of U1248 nucleotide, which harbors the major modification attached post-transcriptionally - m¹acp³ψ (N1-methyl-N3-(3-amino-3-carboxypropyl) pseudouridine). It is unique for this position in the entire 18S rRNA according to the previous literature (Holvec et al., 2024; Maden et al., 1975; Saponara & Enger, 1974), and causes nucleotide misreading detected in bulk sequencing of GTEx cohort, as well as in other independent studies (see (Babaian et al., 2020)).

Other post-transcriptional nucleotide modifications are much lighter and do not cause reading errors because they are correctly recognized by the polymerase although they exist in 18S RNA in many places (human 18S rRNA harbors at least 91 post-transcriptional modifications) (Taoka et al., 2018). These modifications were also detected in our mass spectrometry analysis of 18S rRNA isolated from mouse embryos at the investigated neural crest stages (please see both revised Figure 5 and new Extended Data Fig. 5). This is further supported by the analysis of the GTEx cohort, as well as by our sequencing data, which showed no reading errors at any of the other positions in 18S rRNA, except for U1248. These findings are consistent with other studies of post-transcriptional rRNA modifications (please see (Cui et al., 2024; Liu et al., 2021; Penzo & Montanaro, 2018)).

We hope we corrected the logical flow and all the explanations. In the first submission of this manuscript, we narrated the findings differently and historically, mentioning that in the beginning we were misled by the idea that we are likely dealing with a ribosomal DNA sequence variant (SNP). However, we showed that we were wrong in this logic, and what we have in this position is an extra-

heavy rRNA post-transcriptional modification, which is established by Emg1 and Nhp2 – genes identified in our early pro-mesenchymal signature.

Figure 5c, the Color bar is missing, 5d, how to define low and high?

We added the color scale bars for intensity of gene expression (genes comprising the signature). Of note, in the revised version these plots were moved to Figure 4: Figure 4c and Figure 4d.

Again, Smart-seq 2 uses mg²⁺ buffer for the Reverse transcription.

We thank the reviewer again for raising this important point. We are confident that the observed misreading in U1248 is not an artifact of the Smart-seq2 protocol. In both the GTEx (Lonsdale et al., 2013) and SEQC (SEQC/MAQC-III Consortium, 2014) cohorts that we also analysed in this study, bulk RNA-seq was performed using approaches entirely different from Smart-seq2 and at lower Mg²⁺ concentrations, yet the same signal was observed: a consistent sequencing error at position 1248 of 18S rRNA. A similar observation was made by Babaian et al. (2020), who, in addition to sequencing, confirmed the presence of the modification at position 1248 by mass spectrometry, providing the strongest supporting evidence. In our study, we also employed mass spectrometry and arrived at consistent results. Furthermore, Milenkovic et al. (2025) recently demonstrated the same modification using Nanopore sequencing, further supporting that this finding is robust across multiple technologies.

Other pseudouridine modifications in rRNA do not manifest as sequencing errors because reverse transcriptases generally recognize modified U as U. The exception is N1-methyl-N3-(3-amino-3-carboxypropyl) pseudouridine (m¹acp³Ψ), a structurally “heavier” modification that is capable of disrupting Watson–Crick base-pairing during common reverse transcription (Motorin et al., 2007). We would like to emphasize that the goal of the present study was not to provide a comprehensive atlas of all rRNA modifications in different cell types or stages of neural crest differentiation. We are simply out of capacity. We hope the reviewer could appreciate the amount of the previous and the new work we put in this manuscript, attempting to address only one modification in a single position.

(11) Fig.5d does not show as ordered in the main text.

We apologize for this. We now rebuilt the figures and rewrote large portions of the text, and we hope that we solved this problem.

Reviewer #2 (Remarks to the Author):

In this paper, Poverennaya et al investigate how ribosome composition and ribosomal assembly programs can affect neural crest cell fate bias and implications in neural crest-derived cancers. Here, they investigate cranial vs trunk neural crest fate decisions and uncover a role for a nucleotide modification at position 1248 in 18S rRNA for establishing ectomesenchymal identity. This modification is also present in neural crest-derived tumors wherein a mesenchymal cell identity has been found.

1. Are the neuroglial cells referenced in the single cell data those that will become sympathetic neurons?

To respond to this comment, we additionally checked the markers of sensory vs sympathetic in the neuro-glial cluster, and we see that (based on single cell data) the cells from the neuroglial cluster have a bias towards sensory neurons. E.g., they express *Prdm12* that is a marker of glial cells that can become sensory neural cells (Bartesaghi et al., 2019; Desiderio et al., 2019). However, very few cells express sympathetic markers such as *Ascl1*, *Phox2a/b*, *Th*, *Dbh*. These observations suggest that while the neuro-glial cluster represents a mixed population, it is predominantly represented by a sensory neurogenesis program characteristic of the trigeminal ganglion. In the current manuscript, we now show the dotplot with expression of *Prdm12* - marking the progenitors of trigeminal ganglion - in our revised main Figure 1. Please also see more of the relevant UMAPs below for details of autonomic neuronal fate gene expression (neuroglial cluster is highlighted with a blue oval):

It is also worth noting that this is in line with how we surgically separated the heads with migrating cranial neural crest from the rest of the embryo, mostly containing the trunk neural crest. The surgical separation line goes under the otic vesicle, and for this reason we could get only few sympathetic progenitors committing to the superior cervical ganglion sympathoblasts – the vast majority of sympathoblast-committing cells migrates below the otic vesicle. Thus, here we focused on a very general question of what is biasing ectomesenchymal fate vs general neuronal fate in the cranial crest, with the stronger focus on ectomesenchymal, and not neuronal counterpart. Also, the neuronal counterpart was least affected in *Polr1a* and *Polr1c* knockout experiments according to our new visualizations and stainings (see revised Figure 3), as compared to mesenchyme of the face. Trigeminal also persisted in *Tsr3* knockout (new Figure 6d and Extended Data Fig. 9), which we specifically performed during this revision. All in all, this whole manuscript is centered around the role of ribosomal modifications in cell mesenchymization (in development and in tumors). We can see the interest of

the review in sympathetic neurogenesis and autonomic nervous system part, and we will follow this line more in depth (regarding ribosomal modification programs) in the separate follow up study.

In regards to the connection between the cranial neural crest data and the neural crest-derived cancer aspects of the paper, do the authors think that cancers with a poorer prognosis have a disrupted fate selection during development?

Yes, we think this is a reasonable hypothesis, and we keep working on organoids neuroblastoma-initiation models to figure this out in a context of a different paper. Here, in this study we cannot productively address it at initiation stage, however, instead we can suggest which genetic programs correlate with malignancy (survival prognosis) of neuroblastoma and how these programs correspond to neural crest and ribosomal biogenesis control signatures. This is way more relevant to neuroblastoma tumor heterogeneity and potential EMT/metastasis axis. Based on our data we can suggest that neuroblastomas with poor prognosis dynamically modulate their programs of translational control, and for this they are building pools of ribosomes with modified rRNAs. This, in turn, might be connected to their plasticity of cell states (some sort of a fate selection problem) and potential mesenchymal features, which are the hallmark of metastasis. To explore this part in the revised manuscript, we additionally performed the survival analysis of neuroblastoma patients using raw sequencing data, where we could do the calling for the U1248 modification appearing as sequencing error. The new results (also see Extended Data Fig. 10) showed that the deviation from the average norm proportion of this modification is correlated with the negative prognosis (revised Figure 7). Also, the level of mesenchymal and ribosomal biogenesis control signatures associated with negative prognosis as well, which hypothetically links everything to transitions into mesenchymized tumor cell states resembling neural crest mesenchyme.

2. The authors perform IHC for NHP2 in E9.5 mouse heads and see co-expression of SOX10 and NHP2 (Fig2c,d). Concluding its importance. This staining is not convincing and the Hoechst is quite patchy. It would be helpful to see more sections from those analyzed.

We understand the criticism, and we took steps to improve the reliability of this part. In this revision, we opted for HCR *in situ* hybridization with a probe for *Nhp2* instead of unstable immunohistochemistry due to poor quality of NHP2 antibody. These new data confirm the original observations and are a part of revised Figure 2b-c. The *Nhp2* expression is higher in facial prominence and branchial arches as compared to coalescing trigeminal ganglion and mesodermal positions in the head, thus, supporting the enrichment of *Nhp2* in mesenchymogenic cranial neural crest. We validated this result in independently stained embryos.

3. A conditional KO of polymerase 1 subunit alpha, Polr1a, in mouse embryos at E8.5 results in many NC defects (face, neural, and cardiac), so the authors repeat this experiment at the time of the bifurcation in their sc data and find craniofacial abnormalities (Fig2E). This mutant analysis would benefit from more characterization of this mutant phenotype including histological sections and staining and skeletal preps to show craniofacial defects.

To address this comment, we (1) performed microCT scans of PTA-contrasted Polr1a mutants and littermate controls, followed by 3D segmentations and measurements of facial structures (see revised Figure 3d-e and Extended Data Figure 4c-e where we also show tissue slices). 3D analysis of these early

stages (prior to cartilage being fully formed) is better than sections because sections are harder to adequately compare due to dysmorphic heads in mutant embryos. Our 3D results show that the induction of skeletogenesis is diminished and delayed in mutants, and the amount of general facial mesenchyme is reduced compared to controls. At the same time, the trigeminal ganglion develops and can be visualized in 3D. Also, we added whole mount immunohistochemistry (see updated Figure 3) to observed the development of trigeminal ganglion and skeletogenic mesenchyme that form Meckel cartilage.

In the second line of new evidence (2), we performed the full gene knockout of Tsr3 – the last and the most specific enzyme introducing the modification at U1248 18S rRNA position, where the first and the second steps of this modification are facilitated by Emg1 and Nhp2. This KO revealed insufficient development of facial neural crest derived mesenchyme. We analyzed these embryos using whole mount immunohistochemistry (shown in Figure 6 and Extended Data Figure 9).

Can the authors comment on why the earlier KO resulted in a vast number of defects as well?

The ribosomal exchange is important at two points in the neural crest lineage: during the EMT (the studies from Paul Trainor and Theresa Vincent laboratories - (Falcon et al., 2022; Prakash et al., 2019)) and then at the fate selection between mesenchyme and neuro-glial fates. This ribosomal exchange modulates the translational control in a cell, as the new rRNAs contain modifications in differing proportions. Therefore, the earlier KOs will have affected NCC EMT and resulted in a different severe phenotype, as compared to later inducible and temporally controlled KOs that we use in this study (already after EMT from the neural tube). The difference between the aspects of the phenotype reflects its generality and correlated embryological process.

4. The authors describe a ribosomal assembly signature in different malignant cell populations. This is an interesting point, but did it is unclear whether the authors compare these populations to other tumors that don't have a mesenchymal identity or other adult tissue types? Is this a feature of all cancers or just neural crest-derived ones?

We thank the reviewer for this comments. We split our response into two logical blocks:

1. Our goal was to investigate the identified neural crest-related gene expression signatures in neuro-glial tumors, and more specifically in neural crest-derived tumor neuroblastoma, which shares both neuroblastic and mesenchymal aspects at the level of tumor heterogeneity. We did not have sufficient space in this manuscript to deeply investigate more tumor types. Already at this point, we got many focused questions from the reviewer 3 (about neuroblastoma and our signatures) and spent significant efforts addressing them only for neuroblastoma. We performed a number of additional experiments in neuroblastoma cell lines (please see revised Figure 8 and Extended Fig. 17) and also improved the analysis of patient survival by engaging more cohorts and analysis types. Any generalization or broad comparative attempts will require the new manuscript scale investigation, which we are currently planning. Because this study is dedicated to the neural crest, at this point we find our strategy reasonable and focused.

2. Also, by looking at the % of this modification in position U1248 of 18S rRNA in GTEx cohort, we see that different tissues and organs have a range of U1248 modification level as well as our signatures being represented. The most parsimonious explanation is that the % of U1248 modification, as well as ribosomal biogenesis, play different roles in different cell types. For instance, neurons in that regard

are very likely different from fibroblasts or skin stem cells. Therefore, it is hard to make a conclusion about the % of this modification or our ribosomal control signature in correlation with somewhat universal cell feature, module or state (mesenchymal, for example). For this reason, we focused on the neural crest and the neural crest derivatives. However, we would like to note that previously this modification in U1248 was already associated with many different types of cancer in (Babaian et al., 2020). These authors wrote: „We identify a cancer-specific single-nucleotide variation in 18S rRNA at nucleotide 1248.U in up to 45.9% of patients with colorectal carcinoma (CRC) and present across >22 cancer types.“ However, this study does not involve single cell data analysis of trajectory analysis and operates the bulk sequencing of patient derived tumors (no neuroblastoma included though).

5. When analyzing the different malignant tumor datasets, the authors note that they remove healthy cell types from their analysis of previously published datasets, however, it'd be interesting/necessary to see if there are the same signatures in adult healthy tissues.

We apologize for the confusion we may have caused. To clarify, we did not remove healthy (stromal) cell types from the previously published datasets. Instead, we filtered out only low-quality cells that did not pass quality control thresholds. Specifically, the medulloblastoma dataset (Hovestadt et al., 2019) did not contain stromal cells. In contrast, stromal cells were retained in the glioblastoma dataset (Neftel et al., 2019). Both datasets were generated using Smart-seq2 technology. The previously published neuroblastoma dataset (Dong et al., 2020) used the 10X Genomics platform, which, while offering high cell numbers, provides lower gene detection sensitivity per cell. To ensure a more comprehensive analysis, we generated a new neuroblastoma dataset by sequencing tumor samples from seven patients using Smart-seq2. This dataset includes both stromal and tumor cells. We checked the expression of different signatures (mesenchymal early, mesenchymal late, ribosomal control and rRNA modifications signature) as well as expression of MYC and its transcriptional activity and compared the distribution of mesenchymal early and ribosomal control and rRNA modifications signatures in different cell types or tumor subtypes in glioblastoma, medulloblastoma, and neuroblastoma:

„To address this question, we sequenced several neuroblastoma tumors using Smart-seq2 (Suppl. table 3 for tumor classification details) and also re-analyzed published single cell datasets of glioblastoma and medulloblastoma tumors (Hovestadt et al., 2019; Neftel et al., 2019) (Extended Data Fig. 10a-f). We annotated the cell types in a neuroblastoma dataset based on established markers (Extended Data Fig. 10c,g-i) and kept the original cell annotation for re-investigated published datasets. Then we searched for the enrichment of our CNCC signatures, including ribosomal control and rRNA modification in tumor cell subtypes (Extended Data Fig. 10a-c).

These analyses revealed that the investigated tumors contained diverse populations of malignant cells with increased and decreased mesenchymal gene expression modules, as determined via assessing the presence of “mesenchymal early” and “mesenchymal late” CNCCs programs (Extended Data Fig. 10a-c). This correlated with previous reports of such mesenchymal-like populations in neuroblastoma and glioblastoma (Boeva et al., 2017; Chen et al., 2022; Ozawa et al., 2014). Furthermore, the ribosomal control and rRNA modification signature pattern was differentially distributed to the pattern of “mesenchymal early” and “mesenchymal late” signatures (Extended Data Fig. 10a-c, Suppl. Table 1) . Further analysis revealed a significant proportion of glioblastoma cells with high levels of both “mesenchyme early” and “ribosomal control and rRNA modification” signatures, which were expressed stronger as compared to stromal cell types (Extended Data Fig. 10d). Similarly, the Group 3

medulloblastoma subtype was significantly enriched for these signatures compared to other subtypes (SHH, WNT, Group 4) (Extended Data Fig. 10e). In the case of neuroblastoma, the tumor populations expressed varying degree of ribosomal control and rRNA modification signature (Extended Data Fig. 10f). Interestingly, a small proportion of neuroblastoma tumor cells exhibited a higher mesenchymal early signature (including ribosomal control and rRNA modification signature part), though they never reached the values of truly mesenchymal stromal cells such as the adipocyte-like cell type (Extended Data Fig. 10f). Together, these results support a hypothesis that neuro-glial tumors are comprised of a spectrum of cell phenotypes with different degrees of mesenchymal state that correlates with ribosome modifications.”

These new results are presented in Extended Data Figure 10a–f of the revised manuscript.

6. Is there any evidence to suggest WDR74 is in abundance in other cell types? or in sympathetic neurons or chromaffin cells?

To answer this question, we took advantage of our entire neural crest lineage tree dataset from EMBO paper including also trunk derivatives (Kastriti et al., 2022) to check the expression of *Wdr74* and other genes in cranial mesenchymogenic crest, as well as sympathetic neurons and chromaffin cells. In response to this comment, we added the plotted expression of *Wdr74* to Figure 2d.

As the reviewer can see, on average, the expression of *Wdr74* is systematically higher in mesenchyme and cranial neural crest as compared to sympathetic neurons and chromaffin cells. However, some sympathetic neurons show relatively high *Wdr74* expression levels. These results are shown below (please keep in mind that the single cell data are noisy and the mean expression might be a more meaningful representation of the situation), and they are coherent with the general conclusion that *Wdr74* and other genes from ribosomal control and rRNA modification signature are over-represented in mesenchymal cell types from the neural crest tree, as compared to sensory neurons (in the main figures) and autonomic nervous system cell types:

Reviewer #3 (Remarks to the Author):

The study “rRNA-modifications connect mesenchymal fate selection in neural crest and prognosis in neuroblastoma” by Poverennaya I et al addresses the role of 18s rRNA modifications in fate selection towards skeletogenic ectomesenchyme in embryonic cranial and trunk neural crest and association of ribosomal expression signatures with mesenchymal features and prognosis in tumors derived from the neural crest. Despite presenting interesting observations, the manuscript appears immature and inconsistent with at times sloppy presentation of data in the text and figures jumping between topics, lack of experimental validations and overinterpretation of data presented in a lengthy and unfocused discussion.

Major concerns:

1. Most of the paper relies on inferences from single cell data from mouse created in previous studies and public datasets. More detailed mechanistic validations are missing. E.g. modification in U1248 of 18s rRNA has not been experimentally confirmed, protein interaction is just inferred, but no experimental confirmation either.

We thank the reviewer for this general and constructive comment. We agree that inferring mechanisms from single-cell data must be supported by functional validations as much as possible. In response to this, we have significantly expanded the experimental content of the revised manuscript to strengthen the mechanistic claims and address the limitations noted. We worked for another 1.5 years specifically focusing on experimental confirmations and expansion of the manuscript part related to perturbation studies. We hope the review will appreciate the new results.

Below we provide the summary of revisions and improvements with contextual explanations:

Our work builds on previous biochemical and genetic studies (Brand et al., 1978; Cheng et al., 2024; Liang et al., 2009; Sloan et al., 2017) that established the specific roles of *Nhp2*, *Emg1*, and *Tsr3* in catalyzing the stepwise formation of m¹acp³ψ modification at U1248. These enzymes and their functions are well-conserved, and our data link their activity to a biologically meaningful fate decision point in the cranial neural crest - an insight not previously reported.

1. While initial detection of U1248 modification emerged from analysis of Smart-seq2-based sequencing mismatches, we performed mass spectrometry on isolated 18S rRNA from E9.5 mouse cranial tissue and brain regions. To accomplish this, we established a new collaboration with Prof. Alexandre David – an expert in nucleic acid modifications and their detection by mass spectrometry. This gold-standard biochemical method confirmed the presence and differential abundance of m¹acp³ψ at U1248. This is, to our knowledge, the first direct demonstration of this rare modification in mammalian embryonic tissue, validating its involvement beyond computational prediction. These new results are presented in new Figure 5e,f and Extended Data Figure 5.

„To confirm the presence of the m¹acp³ψ modification in murine embryonic 18S rRNA, we performed mass spectrometry (Fig. 5e). Direct analysis at earlier stages, such as cranial neural crest EMT (E8.5) and subsequent migration and fate selection (E9-9.5), was not feasible due to the limited number of available cells and the sensitivity constraints of mass spectrometry. We therefore chose a slightly later developmental window (E9.5-E10), when cell numbers are substantially higher and allow for reliable detection. To contrast the predominant tissue types at this stage, we dissected the heads of twenty E9.5-E10 mouse embryos from multiple independent litters and surgically separated brain regions from facial structures, collecting

them into two distinct sample sets for quantitative analysis. After isolating 18S rRNA from pooled developing brain areas and facial structures, we subjected it to mass spectrometry to investigate the relative abundance of 30 different nucleotide modifications, including pseudouridines and m¹acp³ψ. We quantified the relative abundance of each modification by a ratio of modified/unmodified nucleoside peak area. Mass spectrometry results confirmed the differential presence of m¹acp³ψ in 18S RNA from the brain area and facial samples (tissues from multiple embryos were pooled together for this analysis) (Fig 5f) at developmental stages after the neuro-mesenchymal bifurcation in CNCCs. Also, as expected, the levels of m¹acp³ψ were much lower compared to other modifications such as pseudouridines (ψ), 2'-O-methyladenosine (Am), 3-methyluridine (m³U) or even 3-(3-amino-3-carboxypropyl)uridine (acp³U) (Extended Data Fig. 5, Suppl. table 2).“

2. To highlight the importance of rRNA synthesis in ectomesenchyme, we performed conditional knockout of polymerase 1 subunit alpha (*Polr1a*) at E9.5 and performed microCT of control and mutant embryos at E12.5. The mutant embryos showed strong craniofacial phenotype (revised Figure 3), which we described in the Results section of the updated manuscript:

“Furthermore, the compositional analysis of the single cell transcriptomics data from E12.5 control and temporal *Polr1a* knockout mutant mice (based on R26-CreERT2), injected with tamoxifen at E9.0-E9.5, revealed a significant reduction of several ectomesenchymal clusters, without neuro-glial progeny being affected (Fig. 3a-b, Extended Data Fig. 4a-b). To validate this observation, we performed HCR in situ hybridization for *Sox9*, a master regulator of chondrogenesis; *Runx2*, a master regulator of osteoblast differentiation; and *Sox10*, a marker of the developing peripheral nervous system (Fig. 3c,d, left panel). We also carried out immunostaining for SOX9 and TUJ1 (TUBB3A; neuron-specific class III β-tubulin) to corroborate the changes at a protein level (Fig. 3d, right panel). As expected, mutant embryos exhibited reduced SOX9 signal in the mandible, consistent with diminished cartilage development. This was evident most notably for Meckel’s cartilage, which manifested as two discrete non-contiguous proximal and distal elements (Fig 3c,d). In contrast, the branches of the neural crest-derived trigeminal ganglion were comparatively less affected, and the mandibular nerve now traversed the gap between the non-contiguous Meckel’s cartilage elements (Fig. 3c,d). Although craniofacial malformations are evident in *Polr1a* mutants in the bright field images (Extended Data Fig. 4c-d); we opted for higher resolution microCT scanning of PTA-contrasted embryos (Kaucka et al., 2017) rendered the 3D volumes of trigeminal ganglion and ectomesenchymal chondrogenic condensations (Fig. 3e, Extended Data Fig. 4e). This approach revealed that whereas *Polr1a*^{flx/flx};R26-Cre-ERT2 embryos exhibited reduced facial mesenchymal condensations, the trigeminal ganglia were not significantly affected, indicating that facial skeletogenesis was specifically disrupted as a consequence of diminished rRNA transcription and ribosome biogenesis (Fig. 3e, Extended Data Fig. 4e).”

Importantly, here we would like to specifically stress out the complexity and hardship of dissecting ribosomal regulation experimentally, particularly due to the embryonic lethality associated with key modifying genes. E.g., knockout of *Emg1* or *Nhp2* leads to early developmental arrest before neurulation (Skarnes et al., 2011; Wu et al., 2010). Despite these challenges, we undertook multiple functional experiments:

3. In the revised manuscript, we performed whole-mount in situ hybridization for *Nhp2* at E9.5 to support our single-cell based conclusions (Figure 2b).
4. Then we performed *in vivo* overexpression of *Nhp2* and *Tsr3* using the NEPTUNE lentiviral system to target in utero developing mouse embryos at neural crest initiation stage. These gain-of-function experiments induced marked craniofacial abnormalities, further confirming the necessity for precise regulation of these rRNA-modifying proteins during CNCC development (Extended Data Fig. 8). Here is the description of these results in the main text:

“To assess the role of these rRNA-modifying enzymes in vivo, we overexpressed Nhp2 and Tsr3 in the neural ectoderm of mouse embryos immediately before the NCC formation. We employed the NEPTUNE system of high-frequency ultrasound-guided nano-injections of lentiviral particles expressing the proteins of interest, together with a fluorescent protein marker (Mangold et al., 2021). The amniotic cavity of E7.5-E8.0 embryos was injected with Nhp2- and Tsr3-carrying lentiviruses and the embryos were harvested at E9.5 after the CNCC cells emigrated to begin to build the cranioskeletal and neuro-glial structures (Extended Data Fig. 8). While control viral injections did not affect embryo morphology, both Nhp2 and Tsr3 overexpression resulted in gross morphological abnormalities throughout the whole embryo including the craniofacial complex (Extended Data Fig. 8). These abnormalities were severe, thus, suggesting that the levels of NHP2 and TSR3 must be tightly controlled during embryogenesis for correct and constructive craniofacial development. Although NHP2 and EMG1 are involved in several cellular processes in addition to catalyzing the initial steps of m¹acp³ψ modification, the only known role of TSR3 is catalyzing the last step of m¹acp³ψ modification in 18S rRNA (Meyer et al., 2016). Together, this supports the importance of m¹acp³ψ in control of developmental processes such as CNCC formation and differentiation.”

5. We also overexpressed *Nhp2*, *Emg1*, and *Tsr3* in iPSC-derived cultured NCCs and analyzed molecular cell phenotypes via 10X Chromium single-cell transcriptomics. Notably, higher levels of *Nhp2* and *Tsr3* influenced NCC developmental stages and proportions, underscoring their developmental impact (Extended Data Fig. 7). Here is the updated text reflecting these findings:

“To explore the functional importance of the m¹acp³ψ modification in embryonic neural crest cells, we utilized lentiviral vectors to overexpress Nhp2, Emg1, or Tsr3 in iPSC-derived neural crest (NC) cells in vitro according to a published protocol (see Methods). We chose an overexpression strategy because previous studies reported early embryonic lethality for Nhp2 and Emg1 gene knockout mutants (Skarnes et al., 2011; Wu et al., 2010). As NCC emerged in vitro, we collected the cultured cells and analyzed them with the help of 10X Chromium single cell transcriptomics. Different ratios of neural tube-like (pre-migratory progenitors) and NCC-like cells (progenies) were observed under NHP2 and TSR3 overexpression conditions, supporting the conclusion that these proteins influence the formation of NCCs (Extended Data Fig. 7).”

6. Next, we generated a full mouse knockout of the last and most specific enzyme modifying rRNA at position U1248 - *Tsr3*, using CRISPR/iGONAD approach (Figure 6, Extended Data Fig. 9). The resulting embryos displayed change of the body size, as well as severely underdeveloped ectomesenchyme, supporting the essential role of *Tsr3* in craniofacial development.

“To further validate this idea, we used the iGonad Cas9/CRISPR based strategy to knock out exons 2-5 of the Tsr3 gene in murine embryos, a phenotype for which had not previously reported in the literature. We opted for an (Gurumurthy et al., 2019; Takabayashi et al., 2022) (Fig 6a,b). The Tsr3^{-/-} mutant embryos appeared dysmorphic at E10.5 compared to littermate controls (Fig. 6c and Extended Data Fig. 9), with delayed development of mesodermal and NCC lineages. Overall, the Tsr3^{-/-} embryos appeared smaller in size compared to control littermates, with a notable exception of the heart, which remained comparable to the size of a normal heart for that age (Fig. 6c). Immunostaining for SOX10, SOX2, and 2H3 (also known as NEFM, a neurofilament marker for peripheral neurons) revealed that although the trigeminal ganglia were affected in the mutant embryos, they still formed successfully in expected locations (Fig. 6d). However, the facial structures were severely underdeveloped, with almost a total absence of anterior ectomesenchyme (Extended Data Fig. 9).”

7. We also downregulated *Tsr3* in two neuroblastoma cell lines – neuroblastic and mesenchymal. *Tsr3* downregulation significantly slowed down the proliferation of mesenchymal cell line, whereas the neuroblastic cell line did not show such effect. Please see the revised Figure 8h for these data.

„To further elucidate the importance of RNA metabolism to neuroblastoma cell physiology, we disrupted the RNA modification-related protein TSR3 (see Fig. 4g). We established neuroblastoma models with stable TSR3 knockdown (shTSR3) derived from SH-SY5Y (WDR74-high; mesenchymal) as well as GIMEN (WDR74-low; neuroblastic) cell lines (Fig. 8h). Although both WDR74-high and WDR74-low shTSR3 models exhibited a comparable >4-fold decrease in TSR3 expression, only the WDR74-high cells exhibited markedly reduced growth (Fig. 8h). These experiments revealed that adequate levels of TSR3 contribute to neuroblastoma cell cycle progression, which in turn accentuates ribosome biogenesis roles beyond protein synthesis.“

8. While some aspects of the ribosomal control and rRNA modification network were initially inferred computationally (survival analysis), we expanded the validations of their correlations in tumor cell lines. E.g., *Wdr74*, a core ribosomal biogenesis network hub, was validated at the protein level in a panel of neuroblastoma cell lines. Its protein level strongly correlated with other core protein NHP2 (Figure 8b). Also, *WDR74*-high lines showed increased association with risk groups and MYC amplification, as well as showed stronger capacity as sphere-forming cells in in vitro assay. Please see updated Figure 8. Here is the relevant text from the Results section:

„To validate the transcriptomic findings, we next asked whether the ribosomal control and rRNA modifications signature also associates with aggressive neuroblastoma traits at the protein level. We focused on two representative ribosomal assembly proteins (based on high quality antibody availability), WDR74 and NHP2, together with markers of aggressive neuroblastoma phenotypes, mesenchymal neuroblastoma-associated prominin-1 (CD133) (Van Groningen et al., 2017) and serotonin receptor HTR3A, a marker of bridge cells with potential implications for neuroblastoma etiology (Kameneva et al., 2022). Intriguingly, screening a panel of 20 neuroblastoma cell lines revealed distinct expression patterns of the selected proteins (Fig. 8a, Extended Data Fig. 17a). By analyzing protein levels in detail, a strong positive correlation was observed between WDR74 and each of the three remaining proteins of interest (Fig. 8b). We therefore used the levels of WDR74 as a classifier, defining two distinct groups of cell lines, WDR74-high and WDR74-low (Fig. 8c). Notably, the WDR74-high group predominantly comprised cell lines derived from high-risk tumors (5/6 cases), including both

MYCN/MYC-amplified and -non-amplified cases, whereas models from low-risk cases were exclusively found in the WDR74-low group.

We therefore hypothesized that protein abundance patterns might correlate with specific aggressive characteristics of neuroblastoma cells. Indeed, as assessed by sphere formation assay, WDR74-high cells showed enhanced cancer stem-like traits in vitro (Fig. 8d). These data suggested that WDR74 alone could serve as a marker of at least a subset of aggressive neuroblastomas. To further explore this potential prognostic significance, we performed survival analysis using a publicly available bulk transcriptomic dataset, which confirmed that WDR74 expression is significantly associated with poor outcomes (Extended Data Fig. 17b). Remarkably, in non-MYCN-amplified neuroblastoma patients - thus controlling for the potential confounding effects of MYCN amplification as a canonical marker of high-risk disease (Brodeur et al., 1984; Cohn et al., 2009) - elevated WDR74 levels remained strongly associated with poor prognosis (Fig. 8e), underscoring the predictive power of a single gene from the ribosomal control and rRNA modification signature.

Next we hypothesized that the elevated expression of ribosomal assembly proteins may reflect the reliance of neuroblastoma cells on high ribosomal production and processivity, making WDR74-high cells more vulnerable to overloading by ribosomal inhibition. Cycloheximide blocks ribosome translocation, effectively locking ribosomes onto mRNA, which leads to rapid depletion of the nuclear pools of ribosomal proteins, arrest of rRNA processing, and an overall increase in cellular demand for ribosomal assembly (Albert et al., 2019; Sharma et al., 2021). In line with this notion, treatment with cycloheximide dramatically limited proliferation of WDR74-high cell lines, which were significantly more sensitive (2.5–20×) than WDR74-low counterparts (Fig. 8f, Extended Data Fig. 17c,d). Importantly, the different sensitivity to ribosomal inhibition could not be explained by differences in basal protein synthesis rates between WDR74-high and -low cell lines. Comparison of the time-dependent increase of short-turnover proteins after their depletion by cycloheximide pre-treatment revealed no consistent trend in the expression dynamics between WDR74-high SH-SY5Y and WDR74-low GIMEN cell lines (Fig. 8g). These results demonstrate that expression of genes from the ribosomal control and rRNA modification signature does not directly determine protein synthesis rates in neuroblastoma cells. Instead, these genes are likely to be involved in other processes, such as RNA metabolism and events associated with differential efficiency of peptide chain elongation, including the possibility of specialized ribosomal pools containing rRNAs with post-transcriptional modifications.“

9. We also established the collaboration with the group of Matthias Fischer, who helped us to analyze the survival curves of NB patients and also performed the analysis of association of U1248 modification in 18S rRNA with survival of neuroblastoma patients from SEQC cohort via accessing raw sequencing reads. This new result can be found in Figure 7i.

„Next we explored the impact of the U1248 modification on neuroblastoma patient survival. Although the expression of genes involved in introducing this modification correlated with poor outcomes (Fig. 7h, Extended Data Fig. 12), this does not directly confirm the impact of the modification itself. To resolve this issue, we estimated the frequency of modification using raw sequencing data and compared it with survival data of high risk neuroblastoma tumors in the SEQC cohort (high risk, non-MYCN-amplified). Notably, the tumors with medium levels of the modification had the best survival outcomes compared to those with either low or high levels of the U1248 modification (Fig. 7i). This suggests that the ratios of $m^1acp^3\psi$ modification in tumors might reflect different programs in tumor cells, and potentially can be linked to specific

aspects of tumor cell behavior. This requires future studies and independent validations across a spectrum of tumor models.”

2. I have a conceptual problem understanding the claim made regarding figure 4 that “aggressive neuroblastoma cell phenotypes associate with high levels of ribosomal assembly proteins”, as all neuroblastoma cell lines are derived from aggressive, metastatic stages (no cell lines from benign, localized or regressing neuroblastomas exist). How do the neuroblastoma cell lines shown in figure 4 differ in terms of MYCN amplification or MYCN/MYC expression? Figure 4f does not show a pronounced difference in the sensitivity of WDR74 low vs high cell lines to cycloheximide. This needs validation in additional cell lines. Of note, translation inhibitors failed in pre-clinical and clinical studies in neuroblastoma.

We understand the reviewer’s concern, and to improve the manuscript, we added the additional color code reflecting the clinicopathological data to Figure 8c (original Figure 4c): risk group, MYCN/MYC amplification status. In fact, the panel comprises a substantial number of our own patient-derived cell lines (those with NBL index) coming from low- & intermediate-risk tumors. We have also highlighted this in the revised manuscript:

“Notably, the WDR74-high group predominantly comprised cell lines derived from high-risk tumors (5/6 cases), including both MYCN/MYC-amplified and -non-amplified cases, whereas models from low-risk cases were exclusively found in the WDR74-low group.”

Based on our results, WDR74-high is likely independent of MYCN/MYC status despite some enrichment for MYCN/MYC-amplification. This is also substantiated by the marked differences in the expression of WDR74 and its prognostic power in the non-MYCN-amplified neuroblastoma from the Kocak cohort (Figure 8e). In agreement with our analysis, a comparison of WDR74 expression in MYCN-amplified and non-amplified groups shows a significant overlap among the WDR74 highly expressing cases, confirming that WDR74-high neuroblastomas are present in both MYCN-amplified and -non-amplified groups:

WDR74-low in MYCN/MYC-non-amplified cell lines derived from high-risk tumors may reflect known differences among high-risk tumors (MYCN vs late onset/high age). Overall, WDR74-low and WDR74-high tumors only partly correspond to MYCN/MYC status, as well as to the current high-risk profile. As it seems, the level of WDR74 reflects some aspect of neuroblastoma heterogeneity, which might help to better classify such tumors and identify patients that might benefit from therapeutic strategies targeting translational control. This is important as many drugs fail in preclinical and clinical settings due to previously unrecognized mechanisms that affect their activity, often resulting in suboptimal study designs. That said, we are not aware of any direct translation inhibitors that have entered and subsequently failed clinical trials for neuroblastoma. In contrast, recent preclinical studies demonstrate that inhibiting translation via targeting eukaryotic initiation factors is selectively effective against neuroblastoma *in vitro* and *in vivo* (Skofler et al., 2021; Volegova et al., 2024).

Importantly, our results related to the translation block via cycloheximide show that the translation itself is not the only function that is controlled by the ribosomal block. In fact, the translation inhibitor experiments were performed to validate the association of WDR74-high phenotype with higher dependency on ribosomal assembly machinery. Cycloheximide „locks“ ribosomes at mRNA, not only inhibiting translation but increasing the demand for ribosomal assembly, which causes the depletion of ribosomal proteins and arrest of rRNA processing (Albert et al., 2019). The regulation of this ribosomal assembly process and links to the other sides of cell physiology are still enigmatic. Our study inspires the idea to further work in this direction.

To strengthen our results and address the comment on the sensitivity of WDR74-high cells to cycloheximide in Fig. 4, we have performed experiments using two additional cell lines. The final analysis (revised Figure 8f and Extended Data Figure 17c) confirms that IC50 values show a statistically

significant difference between WDR74-high and WDR74-low groups (mean IC50, >9-fold difference). Notably, the tested WDR74-high cell lines, including both MYCN/MYC-amplified and non-amplified neuroblastoma models, showed similarly enhanced sensitivity to cycloheximide, further suggesting that the dependence on ribosomal assembly is not directly related to MYCN/MYC levels.

3. Figures are prepared somewhat sloppy, e.g. in figure 1h it is not clear what kind of data is shown in the second row (scatter blots), x and y-axis labels are missing. Figure 1h also lacks a proper description in the figure legend and text. It is e.g. not clear what “real” and “permuted” is referring to. Figure 1l is missing the color code labels.

We apologise for this, and we have updated the figure with all necessary labels. The left part of the former Figure 1h (now Figure 1i) shows the real signal of gene expression module activation over pseudotime (with label „real“ in figure) – this real curve shows the increasing module activation and intra-module or inter-module correlation of expressed genes. However, there is a specific computational permutation test that we always perform to make sure that this results is specific and results from the optimally selected pseudotime window, whitening which we do the analysis step by step to generate the „real“ curve. If we select the analysis window correctly for the case of „real“ analysis, this bootstrapping perturbation test on the window width must show the different result – generating flatter and indiscriminative curve (labelled in Figure 1i as „permuted“). For further technical details please see our manuscript Faure et al., 2022 on the specifics of scFates package operations and output.

We improved the situation by labelling axes in Figure 1i („local correlation values“ on Y axis and „pseudotime values“ on X axis) more clearly and by improving the corresponding figure legend: *„i, Early modules exhibit increasing intra-module coordination and inter-module repulsion. Average local correlations within neuroglial and mesenchymal modules progressively increase before the bifurcation point, indicating enhanced intra-module coordination. In parallel, inter-module correlations decline, reflecting growing repulsion between modules. Permuted controls show much weaker trends thus supporting the biological specificity of the observed patterns.“*

We hope the reviewer finds the panels and legends improved.

4. In figure 3a,b,c the data presented are not convincing and need additional statistical analysis to investigate potential co-expression of ribosomal and mesenchymal signatures, as well as MYC expression and activity.

We thank the reviewer for this comment. Firstly, to respond to it, we have replaced the neuroblastoma dataset that was originally shown in Figure 3c for the better deeper-sequenced one. The initial dataset in the original submission (Dong et al., 2020) was generated by 10X genomics, while medulloblastoma (Hovestadt et al., 2019) and glioblastoma (Neftel et al., 2019) datasets were sequenced with smart-seq2 protocol. To ensure more comprehensive analysis, we produced a new neuroblastoma dataset that was generated by sequencing tumor samples from seven patients using Smart-seq2 (Figure 7).

Then, to address the referees' comment, in addition to checking the correlated expression of different signatures („mesenchymal early“, „mesenchymal late“, „ribosomal control and rRNA modifications signature“), as well as the expression of MYC and its transcriptional activity (Extended Data Figure 10 a-c in a revised manuscript), we compared the distributions of „mesenchymal early“ and „ribosomal

control and rRNA modifications“ signatures in different cell types or tumor subtypes in all datasets (Extended Data Figure 10 d-f):

„These analyses revealed that the investigated tumors contained diverse populations of malignant cells with increased and decreased mesenchymal gene expression modules, as determined via assessing the presence of “mesenchymal early” and “mesenchymal late” CNCCs programs (Extended Data Fig. 10a-c). This correlated with previous reports of such mesenchymal-like populations in neuroblastoma and glioblastoma (Boeva et al., 2017; Chen et al., 2022; Ozawa et al., 2014). Furthermore, the ribosomal control and rRNA modification signature pattern was differentially distributed to the pattern of “mesenchymal early” and “mesenchymal late” signatures (Extended Data Fig. 10a-c, Suppl. Table 1). Further analysis revealed a significant proportion of glioblastoma cells with high levels of both “mesenchyme early” and “ribosomal control and rRNA modification” signatures, which were expressed stronger as compared to stromal cell types (Extended Data Fig. 10d). Similarly, the Group 3 medulloblastoma subtype was significantly enriched for these signatures compared to other subtypes (SHH, WNT, Group 4) (Extended Data Fig. 10e). In the case of neuroblastoma, the tumor populations expressed varying degree of ribosomal control and rRNA modification signature (Extended Data Fig. 10f). Interestingly, a small proportion of neuroblastoma tumor cells exhibited a higher mesenchymal early signature (including ribosomal control and rRNA modification signature part), though they never reached the values of truly mesenchymal stromal cells such as the adipocyte-like cell type (Extended Data Fig. 10f). Together, these results support a hypothesis that neuro-glial tumors are comprised of a spectrum of cell phenotypes with different degrees of mesenchymal state that correlates with ribosome modifications.“

5. Data presented in Figure 7 are confusing and not convincing, as the datasets presented seem very heterogeneous in their composition and mostly contain very few cells.

We have followed the comment of the reviewer and for this reason we re-analysed the data by integrating seven individual neuroblastoma patient datasets into a joint dataset (Extended Data Figure 10c, f-l) to reduce heterogeneity and improve interpretability. We hope the review will appreciate this improvement.

Statistical analysis is missing.

We improved and described the statistical analysis in the corresponding figure legend of the revised manuscript. We also substantially revised the Result section to make the description of updated analysis more precise:

„To address this question, we sequenced several neuroblastoma tumors using Smart-seq2 (Suppl. table 3 for tumor classification details) and also re-analyzed published single cell datasets of glioblastoma and medulloblastoma tumors (Hovestadt et al., 2019; Neftel et al., 2019) (Extended Data Fig. 10a-f). We annotated the cell types in a neuroblastoma dataset based on established markers (Extended Data Fig. 10c,g-i) and kept the original cell annotation for re-investigated published datasets. Then we searched for the enrichment of our CNCC signatures, including ribosomal control and rRNA modification in tumor cell subtypes (Extended Data Fig. 10a-c).

These analyses revealed that the investigated tumors contained diverse populations of malignant cells with increased and decreased mesenchymal gene expression modules, as determined via assessing the presence of “mesenchymal early” and “mesenchymal late” CNCCs programs (Extended Data Fig. 10a-c). This correlated with previous reports of such mesenchymal-like populations in neuroblastoma and

glioblastoma (Boeva et al., 2017; Chen et al., 2022; Ozawa et al., 2014). Furthermore, the ribosomal control and rRNA modification signature pattern was differentially distributed to the pattern of “mesenchymal early” and “mesenchymal late” signatures (Extended Data Fig. 10a-c, Suppl. Table 1). Further analysis revealed a significant proportion of glioblastoma cells with high levels of both “mesenchyme early” and “ribosomal control and rRNA modification” signatures, which were expressed stronger as compared to stromal cell types (Extended Data Fig. 10d). Similarly, the Group 3 medulloblastoma subtype was significantly enriched for these signatures compared to other subtypes (SHH, WNT, Group 4) (Extended Data Fig. 10e). In the case of neuroblastoma, the tumor populations expressed varying degree of ribosomal control and rRNA modification signature (Extended Data Fig. 10f). Interestingly, a small proportion of neuroblastoma tumor cells exhibited a higher mesenchymal early signature (including ribosomal control and rRNA modification signature part), though they never reached the values of truly mesenchymal stromal cells such as the adipocyte-like cell type (Extended Data Fig. 10f). Together, these results support a hypothesis that neuro-glial tumors are comprised of a spectrum of cell phenotypes with different degrees of mesenchymal state that correlates with ribosome modifications.”

How are “unfavorable outcome tumor cells” defined?

Generally, when reanalysing the previously published data we use the annotation performed by the original authors to be consistent with the original publication. Also, in response to this comment, we have updated the annotation our own dataset and provided gene markers to make it more transparent, especially when it comes to defining tumor cells vs stroma (Extended Data Figure 10 g,h). We removed the “unfavorable outcome tumor cells” as an annotation, and instead we provide the label “tumor cells” according to Leiden clustering. In general, our definitions were based on criteria from our joint neuroblastoma manuscript produced in Susanne Schlisio lab (Bedoya-Reina et al., 2021).

Do SCPs refer to tumor cells (then better SCP-like)?

In this dataset, SCP-like cells do not refer to tumor cells, which is now clear in the manuscript (Extended Data Fig 10g). However, according to our recent publication, SCPs can be pre-malignant in some neuroblastoma tumors (Olsen et al., 2024).

Why do the authors refer to T-cells, macrophages etc and then in other to immune cells? This should be harmonized.

This part is now harmonized because the individual datasets are now integrated.

6. Is the variation in 18s rRNA restricted to U1248 as claimed (see figure 6)? This should be verified by investigating other positions as control. It is odd and seems unlikely that a SNP that is ubiquitously present in the investigated normal tissues across all investigated individuals should be associated with modulating fate decision in NCC differentiation and cancer.

- We thank the reviewer for giving us a chance to clarify the logic of our findings. The detected variation in 18S rRNA is based of the post-transcriptional modification in position U1248, and not on the genome or basic nucleotide sequence variation. The amount of this modification varies in different cell types, organs and tissues as validated by the investigation of GTEx cohort (please see Figure 5). Thus, we can definitely assume that different levels of this very specific modification are needed for certain intracellular gene expression programs within different cell types (validated by our functional experiments with Tsr3 and Nhp2). It is logical to assume that the ratios of this specific modification in

the intracellular pool of 18S rRNA can be involved in controlling cell states not only in adult body (again, here we bring the findings from GTEx cohort), but also in development and in pathological conditions (please see how this exact modification in U1248 plays a role in cancer, as reported in Babaian et al., 2020). Thus, being a lab specializing on the neural crest, we explored the role of this modification in neural crest cells and in neuroblastoma cell lines. Also, the unbiasedly defined ribosomal control signature, which is associated with mesenchymal biasing of the cranial neural crest, contains Emg1 and Nhp2 proteins – two key players introducing this exact modification in U1248 position of 18S rRNA.

However, this modification in position U1248 is not the only modification in 18S rRNA. It is that we do not see the other much lighter modifications in our sequencing assay – Smart-seq2, because the polymerase „does not see them“ and interprets nucleotides with lighter modifications as correct basic nucleotides. However, our mass spectrometry experiment on isolated 18S rRNA showed that the other modifications are also present in rRNA molecules, although we cannot assess the balances and proportions in single cell sequencing data. We do not have an instrument to study how these other modifications affect development and tumors. This is a matter of a distant future. However, we can study the huge modification in U1248. Below we reason in more details about the logic of the manuscript.

Indeed, to respond to this comment in a practical way, we made sure that we are talking NOT about the variation in DNA (SNP) of 18SRNA (pro-RNA), but about post-transcriptional modification in position U1248 by performing new mass spectrometry experiments and analyzing GTEx cohort in humans (Fig. 5 a-d). This modification (in U1248) is very heavy, and it is special – being N1-methyl-N3-(3-amino-3-carboxypropyl) pseudouridine, and it occurs in 18S rRNA only one time – in position U1248, and for this reason it appears as a sequencing error (it is huge and unique). Other lighter pseudouridine modifications and methylated adenins are not picked up as sequencing errors, although their positions are perfectly known – pseudouridines are also present in 42 other positions in human 18S rRNA (Taoka et al., 2018). These modifications are compatible with polymerase reading them properly during sequencing. This is obvious when we check other positions as control, please download and check the submitted sequencing data if in doubt. To run these tests, we checked all other positions in 18S rRNA in our single cell data - please, see the figure below. Most of other hypothetical variations emerged in our sequencing data very inconsistently at positions with low coverage, and, as expected, were not supported by the previous literature (Naarmann-de Vries et al., 2023; Taoka et al., 2018). For example, we also identified a set of non-canonical nucleotides repetitively emerging in 18S rRNA within the region 1676-1690. Those were immediately ruled out as bacterial contamination of sequencing samples (this happens in most cases of single cell sequencing and is filtered out at mapping stage).

Distribution of noncanonical nucleotides across 18S rDNA in all cells from mouse cranial NCC dataset, E8.5-E10.5

Distribution of noncanonical nucleotides across 18S rDNA in different celltypes from mouse cranial NCC dataset, E8.5-E10.5

Also, importantly, our U1248 heavy N1-methyl-N3-(3-amino-3-carboxypropyl) pseudouridine modification is located in the P-site of the ribosome, where the peptide bond forms, and it impacts the operational dynamics of ribosome, also predisposing the ribosome to reading some mRNAs more

effectively, especially in case of tumors, according to observations by our colleagues in (Babaian et al., 2020), who analysed this modification in different tumors, and also experimentally validated the modification in position U1248.

We find it very interesting that specialized cells use the proportions of this modification to operate differential gene expression programs, including the situation in tumor cells (please see (Babaian et al., 2020)) for their own evolved reasons. Neural crest cells and neuroblastoma are no exception, and we hope that our findings highlight the complexity of the fate selection processes in embryos, which involve not only transcriptional changes, but also balances of the translational control via creating pools of ribosomes with modified rRNAs. This modulation of proportions of modified/unmodified rRNAs represents an interesting new mechanism of regulating the development of mesenchymal cell types in embryos. However, this mechanism remains largely overlooked by the research community, which has focused attention primarily on transcriptional dynamics (being driven by the recent progress in single cell and spatial transcriptomics).

7. The authors state that “Not surprisingly, the investigated signatures were not able to further differentiate survival in MYCN-amp tumors, which are notoriously associated with the poorest outcomes in neuroblastoma.” This is only true if considering localized and metastatic forms of neuroblastoma, however in the aggressive forms, MYCN amplification is not a poor prognostic factor. For interpretation of the survival analysis (EFS or OS?) it would be important to describe the patient cohort. If all stages were included, MYCN amplified are only high-risk and mostly metastatic, whereas MYCN non-amplified will contain all stages, which will introduce a strong bias in the analysis. In general the survival analysis is not convincing and needs validation in an independent and larger cohort to support the statements made.

We appreciate the reviewer's comment on the survival analysis, which we addressed with a deeper exploration of the mesenchymal signatures and overall survival (OS) stratified by relevant covariates. To further test the validity of our results, we extended the analysis to six neuroblastoma cohorts. These results are presented in Figure 7, Extended Figures 11-15 of a revised manuscript. We hope this analysis will not only satisfy the reviewer's concerns but also demonstrates even stronger pervasiveness of the signatures. To improve on this result, we relied on collaboration with Luis Montano (former member of Florian Halfbritter's lab), Christoph Bartenhagen and Matthias Fischer.

The authors state that “Not surprisingly, the investigated signatures were not able to further differentiate survival in MYCN-amp tumors, which are notoriously associated with the poorest outcomes in neuroblastoma.” This is only true if considering localized and metastatic forms of neuroblastoma, however in the aggressive forms, MYCN amplification is not a poor prognostic factor.

We bona fide believe that this comment highlights an imprecision of our text, which described MYCN as always being associated with the poorest outcomes; however the reviewer correctly pointed out our phrasing lacked nuance. For clarity, and to avoid unintended controversy, we have removed this statement from the revised version of the manuscript.

For interpretation of the survival analysis (EFS or OS?) it would be important to describe the patient cohort.

We agree with the reviewer in that the cohort composition should be described. To this end, we provide cohort descriptor tables alongside our supplementary material (Supp. Table 3). We also have updated the corresponding figures to specify that we worked with OS.

If all stages were included, MYCN amplified are only high-risk and mostly metastatic, whereas MYCN non-amplified will contain all stages, which will introduce a strong bias in the analysis. In general the survival analysis is not convincing and needs validation in an independent and larger cohort to support the statements made.

To address the reviewer's valid concerns, we performed the signature analysis focusing on different subcohorts (Extended Figures 11, 13-15) to rule out stage-related differences, and validated the results in another five large, published cohorts: Oberthuer et al, 2006 (n=251) (Oberthuer et al., 2006); Versteeg cohort (n=88) (Molenaar et al., 2012), TARGET-NBL (n=133) (Pugh et al., 2013), Westermann-ALT dataset (n=144), which comprises also neuroblastomas with ATRX and TERT mutations (Hartlieb et al., 2021), Kocak et al, 2013 (n=476) (Kocak et al., 2013). In addition to SEQC RNA-seq dataset (n=498) (SEQC/MAQC-III Consortium, 2014; Su et al., 2014). that we used for the original version of manuscript, we also analyzed the corresponding microarray dataset (SEQC-GSE49710).

Within non-MYCN-amplified Stage 4 tumors (by definition metastatic), we consistently observed that the “*ribosomal biogenesis and rRNA modifications*” signature was significantly associated with poor overall survival (Extended Figs. 11, 13d). This relationship existed even in a cohort with different mutational affections (Westermann-ALT), potentially suggesting a pan-neuroblastoma relevance. By Cox proportional hazard modelling we confirmed that this effect was not explained by gender or age differences (Extended Fig. 15).

Consistently with our previous results, the same signature did not robustly stratify outcomes in MYCN-amplified tumors of different stages (Extended Figs. 11, 14d,e), indicating that their prognostic value is primarily restricted to the MYCN–non-amplified setting.

We also assessed signature impact across clinical risk groups in datasets with available annotations. Notably, the signature provided additional prognostic resolution in non-MYCN-amplified, non–high-risk tumors as well as not stage 4, non-high risk ones (Extended Figs. 14b,c), suggesting their potential utility for stratifying patients in cohorts where survival differences are less obvious.

To strengthen validation, we directly compared our signatures to the Halbritter C13 signature (Saldana-Guerrero et al., 2024), which predicts favorable outcome. As expected, our ribosomal control and rRNA modifications signature was broadly anti-correlated with C13 (Fig. 7e–g), suggesting opposing biological programmes.

Finally, we also performed the survival analysis using the % of U1248 modification in the same SEQC cohort from Mattias Fisher’s lab. For this, collaborated with Mattias Fisher and analyzed the raw sequencing reads. The results showed that a deviation of % of ribosomal P-site U1248 N1-methyl-N3-(3-amino-3-carboxypropyl) pseudouridine from average value predicts worse survival, especially the elevation of this proportion. This corresponds to the survival analysis when considering expression of this modification-recording genes: *Emg1*, *Nhp2* and *Tsr3*, as well as the entire ribosomal biogenesis control signature.

Overall, we hope we significantly improved the results according to reviewer’s advice and supported the role of ribosomal biogenesis control and rRNA modifications in tumor heterogeneity.

8. The abstract needs revising, as the topic is jumping from cranial to trunk neural crest and from skeletogenic ectomesenchyme to neuroblastoma with no clear rationale. It is also not clear why the authors focused on neuroblastoma from figure 4 onwards.

We thank the reviewer for this valuable feedback. In response, we have substantially revised the abstract and also made significant changes in the structure of the manuscript to make the flow better.

9. The discussion is lengthy, without focus and full of overinterpretation of the data presented.

We hope that we managed to compact the discussion and also made it more critical.

For example, we added these two following passages outlining the limitations of this study:

- 1) *„Interestingly, this result aligns with the outcome of our TSR3 downregulation in neuroblastoma mesenchymal and neuroblastic cell lines, where we observed a significant decrease in the proliferation of mesenchymal cells, whereas neuroblastic cells were not affected. This highlights the biologically meaningful role of TSR3 and $m^1acp^3\psi$ modification of U1248 in 18S rRNA in cells acquiring a mesenchymal phenotype, both in embryos and, possibly, in neuroblastoma tumors. However, in the case of tumors, stronger evidence is required to directly link $m^1acp^3\psi$ 18S rRNA modification to the malignant properties of tumor cells, and this will be a priority of future research.“*

- 2) *„Although this study sheds light on the role of ribosomal modifications in embryonic development and highlights their potential importance in neuroblastoma, there are limitations, which must be acknowledged. For example, the rRNA and ribosome control signature includes proteins, which might have additional as yet unknown functions, consistent with evidence emerging for non-canonical roles of ribosomal proteins (Bursac et al., 2014; Patil and Hsieh, 2017) or even snoRNAs such as SNORA13 (Cheng et al., 2024) that beside guiding RNA modification, also promote p53-mediated senescence by directly binding to RPL23 and decreasing its incorporation into maturing 60S subunits.*

These other roles might also contribute to mesenchymal biasing of CNCC, as well as to the biology of neuroblastoma. Furthermore, ribosome maintenance in vivo involves both the exchange of individual ribosomal proteins and complete ribosome turnover (Fusco et al., 2021; Mathis et al., 2017; Subramanian and van Duin, 1977). More specifically, peripheral structural ribosomal proteins, as well as those directly involved in peptide bond formation, are dynamically replaced multiple times throughout the lifespan of the ribosome - likely through exchange with freely available cytoplasmic protein pools (Mathis et al., 2017). In contrast, most core ribosomal proteins remain stably integrated and persist for the duration of the ribosome's functional life (Mathis et al., 2017). The exchange of ribosomal proteins seems to be an important mechanism for ribosome repair (Pulk et al., 2010; Yang et al., 2023). How this translates to the dynamics of possible rRNA modifications during the lifetime of a single ribosome currently remains unknown. Our results might be relevant to such questions, as we do not observe changes in the expression of genes encoding structural ribosomal proteins in cases where, for example, rRNA modifying enzymes appear to be differentially expressed.“

10. It is not clear why the authors introduce the role of 18s rRNA U1248 variants again from lines 327 onwards, despite already reporting on this in previous paragraphs?

- We completely changed the narrative flow about variants in rDNA genes and post-transcriptional modifications in rRNA in the new revised manuscript, and this problem is solved now. The old line 327 was “we followed an alternative hypothesis, which suggests that rDNA genes, which are organized into tandem repeats, may not be exactly the same, as they might contain nucleotide polymorphisms affecting the operation of ribosomes and predisposing them for more efficient translation of certain types of mRNAs.”

Now we talk about our exploration of this alternative in a different order. We thank the referee for spotting this.

Minor

1. The authors are advised to review the introduction with regards to supporting their statements with adequate literature citations (see e.g. lines 104 – 108).

We thank the reviewer for this helpful suggestion. We carefully reviewed the introduction and have added appropriate and well-established references to support the statements in lines 104–108 and elsewhere as needed.

As an example, we added the following literature citations to the support of former lines 104-108 (lines 124-132 in a revised version):

„Specialization and functional diversity of ribosomes can be achieved by numerous means. At the rRNA level, it can be caused by rRNA variant allele expression (Parks et al., 2018; Rothschild et al., 2024) or by chemical modifications at the level of individual nucleotides (Milenkovic and Novoa, 2025; Rajan et al., 2023). At the ribosomal protein level, heterogeneity can arise from the expression of specific ribosomal protein paralogues (Li et al., 2022; Samir et al., 2018; Shiraishi et al., 2023), post-translational modification of ribosomal proteins (Imami et al., 2018; Ramalho et al., 2024), by difference in ribosomal protein composition as some ribosomal proteins are found to be substoichiometric (Genuth et al., 2022; Shi et al., 2017) or by distinct proteins associating with the mature ribosomes (Joo et al., 2022; Simsek et al., 2017).“

2. I find the title inadequate, as the data presented on survival are weak. The authors should rather focus on their biological findings and choose an appropriate title.

We modified the title of the manuscript to keep it more focused on embryonic development. The new title is *„Ribosomal modifications are associated with mesenchymal fate selection in the neural crest lineage“*.

Also, in response to critical comments, we engaged into new collaborations with Matthias Fisher, and also Christoph Bartenhagen and Luis Montano (former member of Florian Halfbritter lab) – experts in neuroblastoma cohorts and survival analysis, to analyse the survival of patients more in depth (Figure 7, Extended Data Figure 11-17). We also added new functional experiments using different neuroblastoma cell lines (Figure 8, Extended Data Figure 17) and refined our single cell analysis of

tumors from patients (Extended Data Figure 10). We hope this part of the manuscript became stronger, although it is no longer reflected in the title.

Also, we would like to mention that we see a value in neuroblastoma-related results, especially after this revision. We also keep a balanced and a critical stance, and we clearly mentioned our cautious approach in the main text:

1) *„This suggests that the ratios of m1acp3ψ modification in tumors might reflect different programs in tumor cells and potentially can be linked to specific aspects of tumor cell behavior. This requires future studies and independent validations across a spectrum of tumor models.“*

2) *„Interestingly, this result aligns with the outcome of our TSR3 downregulation in neuroblastoma mesenchymal and neuroblastic cell lines, where we observed a significant decrease in the proliferation of mesenchymal cells, whereas neuroblastic cells were not affected. This highlights the biologically meaningful role of TSR3 and m1acp3ψ modification of U1248 in 18S rRNA in cells acquiring a mesenchymal phenotype, both in embryos and, possibly, in neuroblastoma tumors. **However, in the case of tumors, stronger evidence is required to directly link m1acp3ψ 18S rRNA modification to the malignant properties of tumor cells, and this will be a priority of future research.**“*

3) *„Although this study sheds light on the role of ribosomal modifications in embryonic development and highlights their potential importance in neuroblastoma, there are limitations, which must be acknowledged. For example, the rRNA and ribosome control signature includes proteins, which might have additional as yet unknown functions, consistent with evidence emerging for non-canonical roles of ribosomal proteins (Bursac et al., 2014; Patil and Hsieh, 2017) or even snoRNAs such as SNORA13 (Cheng et al., 2024) that beside guiding RNA modification, also promote p53-mediated senescence by directly binding to RPL23 and decreasing its incorporation into maturing 60S subunits.“*

3. Sphere formation is not an assay for stem-cell properties in cancer cells.

We understand the need to be cautious with the interpretations of this experiment, and for this reason, we rewrote the relevant part of Results and Discussion to reflect the limits of this experiment by mentioning the proxy nature of this assay:

„Indeed, as assessed by sphere formation assay, WDR74-high cells showed enhanced cancer stem-like traits in vitro (Fig. 8d).“

Also, in response to this comment, we would like to reiterate that we used the limiting dilution sphere formation assay, assessing the formation of spheres from individual cells seeded at densities of either 200, 100, 50, 25, 5, and 1 per well and monitored real-time by IncuCyte® SX1 imaging. This design allows testing the long-term self-renewal capacity of a cell, which is considered a key characteristic of cancer stem cells.

In this format, sphere formation assay is still the best *in vitro* functional assay of cancer stem cells and is commonly used as such by the cancer stem cell community, including recent papers in:

- Nat Commun (Bu et al., 2023; Wang et al., 2023)
- and other high-quality journals (Chandouri et al., 2024; Sun et al., 2022; Vipparthi et al., 2022)

Still, we followed the hint from the reviewer and decided to discuss this result from multiple angles, including the more critical view.

4. Why are the authors reporting a preliminary analysis in lines 139 – 141?

We apologize for the confusion caused by our wording. Our intention was not to suggest that this was a "preliminary" analysis in the sense of being incomplete. Rather, we meant to describe it as the primary analysis or initial step in our workflow. We have revised the text accordingly to clarify this point and avoid further misunderstanding:

„As expected, the single cell analysis revealed two super-clusters representing the neural tube ($Sox2^+Sox3^+$) and the neural crest ($Sox10^+$) cell populations connected via the bridge of delaminating CNCCs (Fig. 1b-c).“

5. Figure 2b should be moved to the supplemental data, as it does not show original data.

We thank the reviewer for this sincere advice. We asked several of our colleagues to read the manuscript and comment on whether these panels are important or critical for understanding the main findings. Although several people had agreed with the reviewer's advice, the majority preferred to keep this enzymatic reaction in the main figure, because the manuscript follows a number of logical lines which become coherent due to this enzymatic reaction introducing the modification into rRNA via Emg1, Nhp2 and Tsr3 proteins, which we explored in many functional assays. As the reviewer can see, we gave a serious consideration to moving these panels to the supplementary, and yet we decided to keep them for the sake of their pedagogical value (we were really hesitant).

Also, in response to this comment, and to improve the clarity and coherence, we have moved this panel to Figure 4g, which focuses specifically on U1248 modification. We hope reviewer supports this relocation of the panel to a more relevant location.

REFERENCES

- Albert, B., Kos-Braun, I. C., Henras, A. K., Dez, C., Rueda, M. P., Zhang, X., Gadal, O., Kos, M., & Shore, D. (2019). A ribosome assembly stress response regulates transcription to maintain proteome homeostasis. *eLife*, *8*, e45002. <https://doi.org/10.7554/eLife.45002>
- Babaian, A., Rothe, K., Girodat, D., Minia, I., Djondovic, S., Milek, M., Spencer Miko, S. E., Wieden, H.-J., Landthaler, M., Morin, G. B., & Mager, D. L. (2020). Loss of m1acp3Ψ Ribosomal RNA Modification Is a Major Feature of Cancer. *Cell Reports*, *31*(5), 107611. <https://doi.org/10.1016/j.celrep.2020.107611>
- Bartesaghi, L., Wang, Y., Fontanet, P., Wanderoy, S., Berger, F., Wu, H., Akkuratova, N., Bouçanova, F., Médard, J.-J., Petitpré, C., Landy, M. A., Zhang, M.-D., Harrer, P., Stendel, C., Stucka, R., Dusl, M., Kastriti, M. E., Croci, L., Lai, H. C., ... Chrast, R. (2019). PRDM12 Is Required for Initiation of the Nociceptive Neuron Lineage during Neurogenesis. *Cell Reports*, *26*(13), 3484-3492.e4. <https://doi.org/10.1016/j.celrep.2019.02.098>

- Brand, R. C., Klootwijk, J., Planta, R. J., & Maden, B. E. (1978). Biosynthesis of a hypermodified nucleotide in *Saccharomyces carlsbergensis* 17S and HeLa-cell 18S ribosomal ribonucleic acid. *Biochemical Journal*, *169*(1), 71–77. <https://doi.org/10.1042/bj1690071>
- Bu, J., Zhang, Y., Wu, S., Li, H., Sun, L., Liu, Y., Zhu, X., Qiao, X., Ma, Q., Liu, C., Niu, N., Xue, J., Chen, G., Yang, Y., & Liu, C. (2023). KK-LC-1 as a therapeutic target to eliminate ALDH+ stem cells in triple negative breast cancer. *Nature Communications*, *14*(1), 2602. <https://doi.org/10.1038/s41467-023-38097-1>
- Chandouri, B., Naves, T., Yassine, M., Ikhlef, L., Tricard, J., Chaunavel, A., Homayed, Z., Pannequin, J., Girard, N., Durand, S., Carré, V., & Lalloué, F. (2024). Comparison of methods for cancer stem cell detection in prognosis of early stages NSCLC. *British Journal of Cancer*, *131*(9), 1425–1436. <https://doi.org/10.1038/s41416-024-02839-9>
- Cheng, Y., Wang, S., Zhang, H., Lee, J.-S., Ni, C., Guo, J., Chen, E., Wang, S., Acharya, A., Chang, T.-C., Buszczak, M., Zhu, H., & Mendell, J. T. (2024). A non-canonical role for a small nucleolar RNA in ribosome biogenesis and senescence. *Cell*, *187*(17), 4770-4789.e23. <https://doi.org/10.1016/j.cell.2024.06.019>
- Cui, L., Zheng, J., Lin, Y., Lin, P., Lu, Y., Zheng, Y., Guo, B., & Zhao, X. (2024). Decoding the ribosome's hidden language: rRNA modifications as key players in cancer dynamics and targeted therapies. *Clinical and Translational Medicine*, *14*(5), e1705. <https://doi.org/10.1002/ctm2.1705>
- Desiderio, S., Vermeiren, S., Van Campenhout, C., Kricha, S., Malki, E., Richts, S., Fletcher, E. V., Vanwelden, T., Schmidt, B. Z., Henningfeld, K. A., Pieler, T., Woods, C. G., Nagy, V., Verfaillie, C., & Bellefroid, E. J. (2019). Prdm12 Directs Nociceptive Sensory Neuron Development by Regulating the Expression of the NGF Receptor TrkA. *Cell Reports*, *26*(13), 3522-3536.e5. <https://doi.org/10.1016/j.celrep.2019.02.097>
- Dong, R., Yang, R., Zhan, Y., Lai, H.-D., Ye, C.-J., Yao, X.-Y., Luo, W.-Q., Cheng, X.-M., Miao, J.-J., Wang, J.-F., Liu, B.-H., Liu, X.-Q., Xie, L.-L., Li, Y., Zhang, M., Chen, L., Song, W.-C., Qian, W., Gao, W.-Q., ... Wang, J. (2020). Single-Cell Characterization of Malignant Phenotypes and Developmental Trajectories of Adrenal Neuroblastoma. *Cancer Cell*, *38*(5), 716-733.e6. <https://doi.org/10.1016/j.ccell.2020.08.014>
- Falcon, K. T., Watt, K. E. N., Dash, S., Zhao, R., Sakai, D., Moore, E. L., Fitriyani, S., Childers, M., Sardiu, M. E., Swanson, S., Tsuchiya, D., Unruh, J., Bugarinovic, G., Li, L., Shiang, R., Achilleos, A., Dixon, J., Dixon, M. J., & Trainor, P. A. (2022). Dynamic regulation and requirement for ribosomal RNA transcription during mammalian development. *Proceedings of the National Academy of Sciences of the United States of America*, *119*(31), e2116974119. <https://doi.org/10.1073/pnas.2116974119>
- Faure, L., Soldatov, R., Kharchenko, P. V., & Adameyko, I. (2023). scFates: A scalable python package for advanced pseudotime and bifurcation analysis from single-cell data. *Bioinformatics*, *39*(1), btac746. <https://doi.org/10.1093/bioinformatics/btac746>
- Hartlieb, S. A., Sieverling, L., Nadler-Holly, M., Ziehm, M., Toprak, U. H., Herrmann, C., Ishaque, N., Okonechnikov, K., Gartlgruber, M., Park, Y.-G., Wecht, E. M., Savelyeva, L., Henrich, K.-O., Rosswog, C., Fischer, M., Hero, B., Jones, D. T. W., Pfaff, E., Witt, O., ... Westermann, F. (2021). Alternative lengthening of telomeres in childhood neuroblastoma from genome to proteome. *Nature Communications*, *12*(1), 1269. <https://doi.org/10.1038/s41467-021-21247-8>

- Holvec, S., Barchet, C., Lechner, A., Fréchin, L., De Silva, S. N. T., Hazemann, I., Wolff, P., von Loeffelholz, O., & Klaholz, B. P. (2024). The structure of the human 80S ribosome at 1.9 Å resolution reveals the molecular role of chemical modifications and ions in RNA. *Nature Structural & Molecular Biology*, *31*(8), 1251–1264. <https://doi.org/10.1038/s41594-024-01274-x>
- Hovestadt, V., Smith, K. S., Bihannic, L., Filbin, M. G., Shaw, M. L., Baumgartner, A., DeWitt, J. C., Groves, A., Mayr, L., Weisman, H. R., Richman, A. R., Shore, M. E., Goumnerova, L., Rosencrance, C., Carter, R. A., Phoenix, T. N., Hadley, J. L., Tong, Y., Houston, J., ... Northcott, P. A. (2019). Resolving medulloblastoma cellular architecture by single-cell genomics. *Nature*, *572*(7767), 74–79. <https://doi.org/10.1038/s41586-019-1434-6>
- Kastriti, M. E., Faure, L., Von Ahsen, D., Boudierlique, T. G., Boström, J., Solovieva, T., Jackson, C., Bronner, M., Meijer, D., Hadjab, S., Lallemand, F., Erickson, A., Kaucka, M., Dyachuk, V., Perlmann, T., Lahti, L., Krivanek, J., Brunet, J., Fried, K., & Adameyko, I. (2022). Schwann cell precursors represent a neural crest-like state with biased multipotency. *The EMBO Journal*, *41*(17), e108780. <https://doi.org/10.15252/embj.2021108780>
- Kocak, H., Ackermann, S., Hero, B., Kahlert, Y., Oberthuer, A., Juraeva, D., Roels, F., Theissen, J., Westermann, F., Deubzer, H., Ehemann, V., Brors, B., Odenthal, M., Berthold, F., & Fischer, M. (2013). Hox-C9 activates the intrinsic pathway of apoptosis and is associated with spontaneous regression in neuroblastoma. *Cell Death & Disease*, *4*(4), e586. <https://doi.org/10.1038/cddis.2013.84>
- Liang, X., Liu, Q., & Fournier, M. J. (2009). Loss of rRNA modifications in the decoding center of the ribosome impairs translation and strongly delays pre-rRNA processing. *RNA*, *15*(9), 1716–1728. <https://doi.org/10.1261/rna.1724409>
- Liu, K., Santos, D. A., Hussmann, J. A., Wang, Y., Sutter, B. M., Weissman, J. S., & Tu, B. P. (2021). Regulation of translation by methylation multiplicity of 18S rRNA. *Cell Reports*, *34*(10), 108825. <https://doi.org/10.1016/j.celrep.2021.108825>
- Lonsdale, J., Thomas, J., Salvatore, M., Phillips, R., Lo, E., Shad, S., Hasz, R., Walters, G., Garcia, F., Young, N., Foster, B., Moser, M., Karasik, E., Gillard, B., Ramsey, K., Sullivan, S., Bridge, J., Magazine, H., Syron, J., ... Moore, H. F. (2013). The Genotype-Tissue Expression (GTEx) project. *Nature Genetics*, *45*(6), 580–585. <https://doi.org/10.1038/ng.2653>
- Maden, B. E., Forbes, J., de Jonge, P., & Klootwijk, J. (1975). Presence of a hypermodified nucleotide in HeLa cell 18 S and *Saccharomyces carlsbergensis* 17 S ribosomal RNAs. *FEBS Letters*, *59*(1), 60–63. [https://doi.org/10.1016/0014-5793\(75\)80341-3](https://doi.org/10.1016/0014-5793(75)80341-3)
- Milenkovic, I., Cruciani, S., Llovera, L., Lucas, M. C., Medina, R., Pauli, C., Heid, D., Muley, T., Schneider, M. A., Klotz, L. V., Allgäuer, M., Lattuca, R., Lafontaine, D. L. J., Müller-Tidow, C., & Novoa, E. M. (2025). Epitranscriptomic rRNA fingerprinting reveals tissue-of-origin and tumor-specific signatures. *Molecular Cell*, *85*(1), 177-190.e7. <https://doi.org/10.1016/j.molcel.2024.11.014>
- Molenaar, J. J., Koster, J., Zwijnenburg, D. A., van Sluis, P., Valentijn, L. J., van der Ploeg, I., Hamdi, M., van Nes, J., Westerman, B. A., van Arkel, J., Ebus, M. E., Haneveld, F., Lakeman, A., Schild, L., Molenaar, P., Stroeken, P., van Noesel, M. M., Ora, I., Santo, E. E., ... Versteeg, R. (2012). Sequencing of neuroblastoma identifies chromothripsis and defects in neurogenesis genes. *Nature*, *483*(7391), 589–593. <https://doi.org/10.1038/nature10910>

- Motorin, Y., Muller, S., Behm-Ansmant, I., & Branlant, C. (2007). Identification of Modified Residues in RNAs by Reverse Transcription-Based Methods. In *Methods in Enzymology* (Vol. 425, pp. 21–53). Academic Press. [https://doi.org/10.1016/S0076-6879\(07\)25002-5](https://doi.org/10.1016/S0076-6879(07)25002-5)
- Naarmann-de Vries, I. S., Zorbas, C., Lemsara, A., Piechotta, M., Ernst, F. G. M., Wacheul, L., Lafontaine, D. L. J., & Dieterich, C. (2023). Comprehensive identification of diverse ribosomal RNA modifications by targeted nanopore direct RNA sequencing and JACUSA2. *RNA Biology*, *20*(1), 652–665. <https://doi.org/10.1080/15476286.2023.2248752>
- Neftel, C., Laffy, J., Filbin, M. G., Hara, T., Shore, M. E., Rahme, G. J., Richman, A. R., Silverbush, D., Shaw, M. L., Hebert, C. M., Dewitt, J., Gritsch, S., Perez, E. M., Gonzalez Castro, L. N., Lan, X., Druck, N., Rodman, C., Dionne, D., Kaplan, A., ... Suvà, M. L. (2019). An Integrative Model of Cellular States, Plasticity, and Genetics for Glioblastoma. *Cell*, *178*(4), 835–849.e21. <https://doi.org/10.1016/j.cell.2019.06.024>
- Oberthuer, A., Berthold, F., Warnat, P., Hero, B., Kahlert, Y., Spitz, R., Ernestus, K., König, R., Haas, S., Eils, R., Schwab, M., Brors, B., Westermann, F., & Fischer, M. (2006). Customized oligonucleotide microarray gene expression-based classification of neuroblastoma patients outperforms current clinical risk stratification. *Journal of Clinical Oncology: Official Journal of the American Society of Clinical Oncology*, *24*(31), 5070–5078. <https://doi.org/10.1200/JCO.2006.06.1879>
- Olsen, T. K., Otte, J., Mei, S., Embaie, B. T., Kameneva, P., Cheng, H., Gao, T., Zachariadis, V., Tsea, I., Björklund, Å., Kryukov, E., Hou, Z., Johansson, A., Sundström, E., Martinsson, T., Fransson, S., Stenman, J., Fard, S. S., Johnsen, J. I., ... Baryawno, N. (2024). Joint single-cell genetic and transcriptomic analysis reveal pre-malignant SCP-like subclones in human neuroblastoma. *Molecular Cancer*, *23*(1), 180. <https://doi.org/10.1186/s12943-024-02091-y>
- Penzo, M., & Montanaro, L. (2018). Turning Uridines around: Role of rRNA Pseudouridylation in Ribosome Biogenesis and Ribosomal Function. *Biomolecules*, *8*(2), 38. <https://doi.org/10.3390/biom8020038>
- Picelli, S., Faridani, O. R., Björklund, Å. K., Winberg, G., Sagasser, S., & Sandberg, R. (2014). Full-length RNA-seq from single cells using Smart-seq2. *Nature Protocols*, *9*(1), 171–181. <https://doi.org/10.1038/nprot.2014.006>
- Prakash, V., Carson, B. B., Feenstra, J. M., Dass, R. A., Sekyrova, P., Hoshino, A., Petersen, J., Guo, Y., Parks, M. M., Kurylo, C. M., Batchelder, J. E., Haller, K., Hashimoto, A., Rundqvist, H., Condeelis, J. S., Allis, C. D., Drygin, D., Nieto, M. A., Andäng, M., ... Vincent, C. T. (2019). Ribosome biogenesis during cell cycle arrest fuels EMT in development and disease. *Nature Communications*, *10*(1), 2110. <https://doi.org/10.1038/s41467-019-10100-8>
- Pugh, T. J., Morozova, O., Attiyeh, E. F., Asgharzadeh, S., Wei, J. S., Auclair, D., Carter, S. L., Cibulskis, K., Hanna, M., Kiezun, A., Kim, J., Lawrence, M. S., Lichtenstein, L., McKenna, A., Peadarallu, C. S., Ramos, A. H., Shefler, E., Sivachenko, A., Sougnez, C., ... Maris, J. M. (2013). The genetic landscape of high-risk neuroblastoma. *Nature Genetics*, *45*(3), 279–284. <https://doi.org/10.1038/ng.2529>
- Saldana-Guerrero, I. M., Montano-Gutierrez, L. F., Boswell, K., Hafemeister, C., Poon, E., Shaw, L. E., Stavish, D., Lea, R. A., Wernig-Zorc, S., Bozsaky, E., Fetahu, I. S., Zoescher, P., Pötschger, U., Bernkopf, M., Wenninger-Weinzierl, A., Sturtzel, C., Souilhol, C., Tarelli, S., Shoeb, M. R., ... Halbritter, F. (2024). A human neural crest model reveals the developmental impact of neuroblastoma-associated chromosomal aberrations. *Nature Communications*, *15*(1), 3745. <https://doi.org/10.1038/s41467-024-47945-7>

- Saponara, A. G., & Enger, M. D. (1974). The isolation from ribonucleic acid of substituted uridines containing alpha-aminobutyrate moieties derived from methionine. *Biochimica Et Biophysica Acta*, 349(1), 61–77. [https://doi.org/10.1016/0005-2787\(74\)90009-4](https://doi.org/10.1016/0005-2787(74)90009-4)
- SEQC/MAQC-III Consortium. (2014). A comprehensive assessment of RNA-seq accuracy, reproducibility and information content by the Sequencing Quality Control Consortium. *Nature Biotechnology*, 32(9), 903–914. <https://doi.org/10.1038/nbt.2957>
- Skarnes, W. C., Rosen, B., West, A. P., Koutsourakis, M., Bushell, W., Iyer, V., Mujica, A. O., Thomas, M., Harrow, J., Cox, T., Jackson, D., Severin, J., Biggs, P., Fu, J., Nefedov, M., de Jong, P. J., Stewart, A. F., & Bradley, A. (2011). A conditional knockout resource for the genome-wide study of mouse gene function. *Nature*, 474(7351), 337–342. <https://doi.org/10.1038/nature10163>
- Skofler, C., Kleinegger, F., Krassnig, S., Birkel-Toeglhofer, A. M., Singer, G., Till, H., Benesch, M., Cencic, R., Porco, J. A., Pelletier, J., Castellani, C., Raicht, A., Izycka-Swieszewska, E., Czapiewski, P., & Haybaeck, J. (2021). Eukaryotic Translation Initiation Factor 4A1: A Potential Novel Target in Neuroblastoma. *Cells*, 10(2), 301. <https://doi.org/10.3390/cells10020301>
- Sloan, K. E., Warda, A. S., Sharma, S., Entian, K.-D., Lafontaine, D. L. J., & Bohnsack, M. T. (2017). Tuning the ribosome: The influence of rRNA modification on eukaryotic ribosome biogenesis and function. *RNA Biology*, 14(9), 1138–1152. <https://doi.org/10.1080/15476286.2016.1259781>
- Su, Z., Fang, H., Hong, H., Shi, L., Zhang, W., Zhang, W., Zhang, Y., Dong, Z., Lancashire, L. J., Bessarabova, M., Yang, X., Ning, B., Gong, B., Meehan, J., Xu, J., Ge, W., Perkins, R., Fischer, M., & Tong, W. (2014). An investigation of biomarkers derived from legacy microarray data for their utility in the RNA-seq era. *Genome Biology*, 15(12), 523. <https://doi.org/10.1186/s13059-014-0523-y>
- Sun, Y., Yan, K., Wang, Y., Xu, C., Wang, D., Zhou, W., Guo, S., Han, Y., Tang, L., Shao, Y., Shan, S., Zhang, Q. C., Tang, Y., Zhang, L., & Xi, Q. (2022). Context-dependent tumor-suppressive BMP signaling in diffuse intrinsic pontine glioma regulates stemness through epigenetic regulation of CXXC5. *Nature Cancer*, 3(9), 1105–1122. <https://doi.org/10.1038/s43018-022-00408-8>
- Taoka, M., Nobe, Y., Yamaki, Y., Sato, K., Ishikawa, H., Izumikawa, K., Yamauchi, Y., Hirota, K., Nakayama, H., Takahashi, N., & Isobe, T. (2018). Landscape of the complete RNA chemical modifications in the human 80S ribosome. *Nucleic Acids Research*, 46(18), 9289–9298. <https://doi.org/10.1093/nar/gky811>
- Vipparthi, K., Hari, K., Chakraborty, P., Ghosh, S., Patel, A. K., Ghosh, A., Biswas, N. K., Sharan, R., Arun, P., Jolly, M. K., & Singh, S. (2022). Emergence of hybrid states of stem-like cancer cells correlates with poor prognosis in oral cancer. *iScience*, 25(5), 104317. <https://doi.org/10.1016/j.isci.2022.104317>
- Volegova, M. P., Brown, L. E., Banerjee, U., Dries, R., Sharma, B., Kennedy, A., Porco, J. A., & George, R. E. (2024). The MYCN 5' UTR as a therapeutic target in neuroblastoma. *Cell Reports*, 43(5), 114134. <https://doi.org/10.1016/j.celrep.2024.114134>
- Wang, F., Gao, Y., Xue, S., Zhao, L., Jiang, H., Zhang, T., Li, Y., Zhao, C., Wu, F., Siqin, T., Liu, Y., Wu, J., Yan, Y., Yuan, J., Jiang, J., & Li, K. (2023). SCARB2 drives hepatocellular carcinoma tumor initiating cells via enhanced MYC transcriptional activity. *Nature Communications*, 14(1), 5917. <https://doi.org/10.1038/s41467-023-41593-z>

Wu, X., Sandhu, S., Patel, N., Triggs-Raine, B., & Ding, H. (2010). EMG1 is essential for mouse pre-implantation embryo development. *BMC Developmental Biology*, *10*, 99.
<https://doi.org/10.1186/1471-213X-10-99>

REVIEWER COMMENTS

Reviewer #1 (Remarks to the Author):

The reviewers have address most of my concerns. Despite much of their efforts, most of the approaches towards identifying the m¹acp³Ψ modification on U1248 are indirect. It would be great if they could show that the profile of m¹acp³Ψ modification on a synthetic RNA is the same as that on rRNA using direct RNA sequencing, although this could be too tedious.

- We thank the reviewer very much for the careful and thoughtful analysis of our manuscript, and we are particularly grateful for the constructive suggestion to perform an additional experiment involving the chemical synthesis of the m¹acp³Ψ modification in the context of synthetic RNA. We fully agree that, in principle, this would be an elegant way to further substantiate our conclusions. As the reviewer kindly notes, such an experiment may be quite tedious; if the main obstacle were only the experimental effort, we would certainly attempt it, as we hope is evident from the substantial body of new data generated over ~1.5 years for this revision.

To assess feasibility, we contacted several companies with relevant expertise (including Eurofins, IDT, and others). Their feedback was that introducing m¹acp³Ψ at the desired position in synthetic RNA is currently not possible due to the lack of exact knowledge of how to achieve that using organic chemistry, and instead would require a biological enzymatic reaction. To their knowledge, this has not yet been achieved *in vitro* in a defined synthetic or natural RNA context.

Implementing this in our own laboratory would effectively entail launching a *de novo* organic chemistry project in a developmental biology setting, without prior experience in this area and with highly uncertain prospects of success. We also consulted several colleagues with expertise in RNA chemistry and modification via a “biological” direction. They suggested that one possible route would be to optimize conditions for isolating the active enzymatic complexes and performing the modification *in vitro*. However, they also emphasized that the complexity of these multi-component assemblies would make it very unlikely that we could establish robust conditions within a reasonable time frame (on the order of one to two years). In light of these technical and practical constraints, we have respectfully decided not to pursue this additional set of experiments for the current manuscript. We believe that the combined weight of our existing data is sufficient to support the presence of this modification at the specific position we report. Moreover, this modification and its positional specificity are already well documented in the literature (Brand et al., 1978; Holvec et al., 2024; Natchiar et al., 2017; Saponara & Enger, 1974), including studies in which it similarly manifests as a characteristic sequencing error (Babaian et al., 2020; Milenkovic et al., 2025). We therefore feel that, while scientifically appealing, this experiment would mainly replicate findings that have been independently established by several previous publications.

Again, we are very grateful to the reviewer for the many thoughtful and constructive suggestions, which have substantially improved the current manuscript. We also hope that this line of research will lead to further intriguing developmental studies in the future, and we would be delighted if the reviewer encounters these forthcoming contributions from our group and others.

Reviewer #2 (Remarks to the Author):

The authors addressed all of my concerns sufficiently for publication.

Reviewer #3 (Remarks to the Author):

I have carefully read the revised manuscript as well as the authors' responses to my and the other reviewer's comments.

I commend the authors for their thorough approach in revising the manuscript. A substantial set of new data and additional analyses have been included, which mostly support the initial claims.

There remain two considerations / clarifications:

1. In response to comment 2, the authors provide additional experiments using own cell lines generated from low-risk neuroblastoma (according to their claim). The authors state: "Notably, the WDR74-high group predominantly comprised cell lines derived from high-risk tumors (5/6 cases), including both MYCN/MYC-amplified and -non-amplified cases, whereas models from low-risk cases were exclusively found in the WDR74-low group."

It seems surprising that the authors managed to derive cell lines from low-risk neuroblastoma tumors, as such attempts have been unsuccessful for decades in efforts worldwide. It will be important to provide histopathological and genetic evidence that these cell lines were indeed derived from low-risk tumors.

- Thank you very much for bringing this to our attention. After extensive discussion with colleagues in the neuroblastoma community (including Frank Westermann, Olivier Delattre, Isabelle Janoueix, and others), we have come to appreciate that our original interpretation could indeed be confusing. Our colleagues also emphasized that, in their experience, it has not been possible to derive tumor cell lines from low-grade tumors.

As the low-risk versus intermediate-risk grouping is not important for the conclusions drawn from experiments in neuroblastoma cell lines, we have decided to merge these groups, distinguishing only between high-risk and non-high-risk neuroblastoma-derived cell lines. In line with this, we added the necessary adaptations to the main text and methods, and we do not claim now that those are low risk-tumor cell lines to avoid any future ambiguity.

At the same time, we now provide Extended Fig. 18, which summarizes the clinicopathological characteristics related to the respective tumor cohort and the risk assessment according to the updated COG v2 risk stratifier. The risk biomarkers - including comparative genomic hybridization profiles for the low-risk tumors shown in Extended Fig. 18 - together with the treatment regimens and follow-up data, support accurate risk stratification of the tumors used to generate our patient-derived cell lines. However, we appreciate that neuroblastoma tumors are inherently heterogeneous, and some low-risk tumors might contain some (potentially undetected) component of neuroblastoma cells with intermediate-risk characteristics. Although this is beyond the scope of the current study, our future research will focus on comprehensively characterizing the cell lines derived from these low-risk neuroblastoma tumors to determine whether they truly retain key features of low-risk tumors in culture. To this end, we can also note that these five putative LR cell lines were derived over a period of 14 years and require tedious culture, exhibiting much slower growth, which allows for subculturing once every 1-2 weeks.

We decided to run a careful and deep characterization of all of them, which will take long time and is beyond the scope of this manuscript. We plan to publish a separate manuscript dedicated to this in-depth investigation of these cells lines + some new recently derived. This will take about 1 year of additional work, but the result will be visible to the neuroblastoma community.

We thank the reviewer for highlighting this issue, and we agree that it might appear confusing, also requiring long additional work. We are grateful for that we could improve on this part and avoid any significant dissonance.

2. The manuscript text, especially the results and discussion section, are still lengthy and would benefit from a more concise format.

- We thank the reviewer for this advice, which we followed and removed 1 page of the text from the Discussion. We do not see options for even stronger compaction because we had to accommodate all the responses to the previous reviewer's comments.

References

- Babaian, A., Rothe, K., Girodat, D., Minia, I., Djondovic, S., Milek, M., Spencer Miko, S. E., Wieden, H.-J., Landthaler, M., Morin, G. B., & Mager, D. L. (2020). Loss of m1acp3Ψ Ribosomal RNA Modification Is a Major Feature of Cancer. *Cell Reports*, *31*(5), 107611. <https://doi.org/10.1016/j.celrep.2020.107611>
- Brand, R. C., Klootwijk, J., Planta, R. J., & Maden, B. E. (1978). Biosynthesis of a hypermodified nucleotide in *Saccharomyces carlsbergensis* 17S and HeLa-cell 18S ribosomal ribonucleic acid. *Biochemical Journal*, *169*(1), 71–77. <https://doi.org/10.1042/bj1690071>
- Holvec, S., Barchet, C., Lechner, A., Fréchin, L., De Silva, S. N. T., Hazemann, I., Wolff, P., von Loeffelholz, O., & Klaholz, B. P. (2024). The structure of the human 80S ribosome at 1.9 Å resolution reveals the molecular role of chemical modifications and ions in RNA. *Nature Structural & Molecular Biology*, *31*(8), 1251–1264. <https://doi.org/10.1038/s41594-024-01274-x>
- Milenkovic, I., Cruciani, S., Llovera, L., Lucas, M. C., Medina, R., Pauli, C., Heid, D., Muley, T., Schneider, M. A., Klotz, L. V., Allgäuer, M., Lattuca, R., Lafontaine, D. L. J., Müller-Tidow, C., & Novoa, E. M. (2025). Epitranscriptomic rRNA fingerprinting reveals tissue-of-origin and tumor-specific signatures. *Molecular Cell*, *85*(1), 177-190.e7. <https://doi.org/10.1016/j.molcel.2024.11.014>
- Natchiar, S. K., Myasnikov, A. G., Kratzat, H., Hazemann, I., & Klaholz, B. P. (2017). Visualization of chemical modifications in the human 80S ribosome structure. *Nature*, *551*(7681), 472–477. <https://doi.org/10.1038/nature24482>
- Saponara, A. G., & Enger, M. D. (1974). The isolation from ribonucleic acid of substituted uridines containing alpha-aminobutyrate moieties derived from methionine. *Biochimica Et Biophysica Acta*, *349*(1), 61–77. [https://doi.org/10.1016/0005-2787\(74\)90009-4](https://doi.org/10.1016/0005-2787(74)90009-4)